# A hippocampus-accumbens code guides goal-directed appetitive behavior

Oliver Barnstedt [1,2,6] ✉, Petra Mocellin[1,3,7] & Stefan Remy [1,2,4,5] ✉

The dorsal hippocampus (dHPC) is a key brain region for the expression of spatial memories, such as navigating towards a learned reward location. The nucleus accumbens (NAc) is a prominent projection target of dHPC and implicated in value-based action selection. Yet, the contents of the dHPC→NAc information stream and their acute role in behavior remain largely unknown. Here, we found that optogenetic stimulation of the dHPC→NAc pathway while mice navigated towards a learned reward location was both necessary and sufficient for spatial memory-related appetitive behaviors. To understand the task-relevant coding properties of individual NAc-projecting hippocampal neurons (dHPC[→NAc]), we used in vivo dual-color two-photon imaging. In contrast to other dHPC neurons, the dHPC[→NAc] subpopulation contained more place cells, with enriched spatial tuning properties. This subpopulation also showed enhanced coding of non-spatial task-relevant behaviors such as deceleration and appetitive licking. A generalized linear model revealed enhanced conjunctive coding in dHPC[→NAc] neurons which improved the identification of the reward zone. We propose that dHPC routes specific reward-related spatial and behavioral state information to guide NAc action selection.

Memories allow an organism to use past experience to optimize current and future behaviors[1]. The hippocampus (HPC) is one of the main sites of memory-related plasticity[2–4]. Yet, surprisingly little is known about the translation of this information into behavioral action: which hippocampal output pathways send which types of information to guide memory-driven behavior? This gap in our understanding stems from technical challenges associated with recording from large numbers of identified hippocampal projection neurons while animals perform memory-dependent behaviors.

The HPC is heavily implicated in the processing of mnemonic and spatial information. In particular, episodic and spatial memories require an intact hippocampus[2,5]. In line with this, a large proportion of dorsal HPC (dHPC) principal neurons, referred to as "place cells", are tuned to specific spatial locations in one-, two-, or three-dimensional environments[6–8]. They are acutely required to find a hidden reward zone on a linear track[9], and are hypothesized to form a "cognitive map"[10] that supports goal-directed navigation[10]. This cognitive map is further supported by neuronal coding of navigationally relevant features such as borders, speed, reward/goal locations[11–14], but also non-spatial information such as that about future decisions or behavioral tasks[15,16]. The conjunction of such features may provide a "scaffold" that supports the formation of episodic memories[17]. This hippocampal information is routed via hippocampal Cornu Ammonis field 1 (CA1) and subiculum to various cortical and subcortical brain regions to influence memory-guided behaviors[5].

One prominent output region is the nucleus accumbens (NAc), a region within the basal ganglia that is crucial for value-based action selection, e.g., choosing to press a lever in expectation of reward[18,19].

[1]Department of Cellular Neuroscience, Leibniz Institute for Neurobiology, 39118 Magdeburg, Germany. [2]German Center for Neurodegenerative Diseases (DZNE), 39120 Magdeburg, Germany. [3]International Max Planck Research, School for Brain & Behavior (IMPRS), 53175 Bonn, Germany. [4]Center for Behavioral Brain Sciences (CBBS), 39106 Magdeburg, Germany. [5]German Center for Mental Health (DZGP), partner site Halle-Jena-Magdeburg, 39118 Magdeburg, Germany. [6]Present address: Institute for Biology, Otto-von-Guericke University, 39120 Magdeburg, Germany. [7]Present address: Department of Molecular and Cell Biology and Helen Wills Neuroscience Institute, University of California, Berkeley, CA 94720-3370, USA. ✉e-mail: oliver.barnstedt@ovgu.de; stefan.remy@lin-magdeburg.de

The NAc receives projections from both dorsal and ventral CA1 and subiculum[20–22] and is hypothesized to represent an "interface between limbic and motor circuitry"[23]. Indeed, the NAc is considered a main site for transforming a hippocampal spatial code into motivation-driven action[24]. Accordingly, NAc neurons show spatial tuning, are required for spatial memory acquisition and consolidation, and display task-dependent synchrony with dHPC neurons[25–28]. Disabling HPC→NAc projections diminishes conditioned place preference (CPP)− a form of spatial reward learning[22,29] – while optogenetic stimulation is sufficient to artificially induce CPP[30,31]. While these findings cement the NAc's role as an indispensable hippocampus output node, surprisingly little is known about which spatial, contextual, and behavioral patterns of hippocampal information the NAc receives. Here, we set out to understand the specific contents of this information stream during goal-directed navigation and its acute role in behavior. We hypothesized that the HPC may selectively route both spatial as well as other task-relevant information to engage the NAc in reward-seeking behaviors upon approaching a learned reward zone.

To test the role of this pathway for head-fixed spatial reward learning and navigation, we used optogenetic silencing and activation tools, and found that dHPC→NAc activity is both necessary for spatially precise memory-guided appetitive behaviors as well as sufficient to elicit such behaviors. To understand the coding properties of NAc-projecting neurons, we employed dual-color two-photon imaging, capturing in vivo calcium signals of large populations of neurons in dHPC, while using a projection-specific red fluorophore to allow identification of NAc-projecting neurons (dHPC→NAc). Directly comparing dHPC→NAc activity with the rest of the dHPC population (dHPC⁻), we found enhanced spatial tuning in dHPC→NAc neurons, with strong modulation by local cue boundaries and the reward zone. We also found elevated coding of low velocities and appetitive licking. Lastly, we show stronger conjunctive coding of space, velocity, and appetitive licking, which improves classification of the reward zone with a linear decoder. We thus propose that dHPC routes reward context-enriched spatial and behavioral state information to bias NAc action selection.

## Results

### The hippocampus-accumbens pathway is both necessary and sufficient for spatial reward memory-related appetitive behavior

To investigate coding of spatial information and goal-directed behavior in dHPC→NAc neurons, we trained food-restricted mice ($n = 18$) on a head-fixed spatial reward learning task. Mice had to run on a self-propelled treadmill, traversing a 360 cm long textile belt with six differently textured zones, including one otherwise unmarked 30 cm long fixed reward zone (located in the "center" of the track for one cohort, at the "end" for the others)[32,33]; licking a spout in this zone causes a liquid reward to be dispensed once per lap (Fig. 1a and Supplementary Fig. 1a–e). Mice underwent five days of training in which they learned to obtain more rewards by progressively increasing their licking in both reward zone and a 30 cm "anticipation zone" preceding the reward zone (Fig. 1b–d and Supplementary Fig. 1f–n).

Previous studies suggest that the dHPC→NAc pathway is necessary for learned associations of spatial context with reward expectations[22,29]. To test if this pathway is also involved in the retrieval of spatial reward memories in our head-fixed goal navigation task, we injected C57Bl/6 mice with retrograde-targeted CaMKIIa-driven ArchT ($n = 5$ mice) or EGFP ($n = 5$ mice) into the NAc, while implanting optic fibers above dHPC (Fig. 1e, f). We then trained habituated and food-restricted mice to associate a hidden reward zone on a linear track for four days as described before. On the fifth day, mice received light stimulation (15 mW of 561 nm continuous light) for up to ten seconds upon entering a stimulation zone starting 90 cm before the reward zone up until its end (Fig. 1g). While the number of rewarded laps did not significantly change (Supplementary Fig. 2a), we observed an ArchT-specific

significant reduction in appetitive licking right before entry into the reward zone, suggesting a lack of spatial precision or confidence to engage in appetitive licking (Fig. 1h, i; Supplementary Fig. 2b–d; Supplementary Table 1). We observed no effect on consummatory licking (Supplementary Fig. 2e, f). To understand if this pathway also contributes to learning novel spatial reward zones, we next shifted both reward zone and light stimulation zone by 180 cm and observed the learning performance of the animals over the course of three days. While control mice continuously improved their performance to lick at the novel reward zone, ArchT-expressing mice showed a significantly worse learning performance (Fig. 1j; Supplementary Table 2). These results suggest that the dHPC→NAc pathway is necessary for both the expression of spatial reward memories as well as its acquisition in head-fixed spatial navigation tasks.

Previous work suggests that the activation of specific hippocampal pyramidal neurons can drive appetitive licking[9]. To test if such a causal role may be mediated by dHPC→NAc projections, we injected animals with either CaMKIIa-driven ChR2 ($n = 4$ mice) or EYFP ($n = 3$ mice) into dHPC and implanted light fibers in the NAc (Fig. 2a, b). After habituating mice to run on the treadmill and receive rewards upon licking the lick spout, mice were given 5 mW of 473 nm 20 Hz (5 ms duration) pulsed laser light for up to 10 seconds[31,34,35] upon entry into an unmarked light stimulation zone. We found that, shortly after stimulation onset, ChR2-expressing mice reliably showed increased mouth movement for up to two seconds after stimulation, while we observed no effects in mice expressing EYFP ($P_{ChR2} = 0.0099$, $P_{EYFP} = 0.617$, paired $t$-tests; Fig. 2c–f; Supplementary Table 3; Supplementary Movie 1). In line with this, we also found a significant deceleration of running on the treadmill upon light delivery in ChR2 animals but not EYFP animals ($P_{ChR2} = 0.0381$, $P_{EYFP} = 0.334$, paired $t$-tests; Fig. 2g, h). We also find evidence that the stimulation of this pathway enhances spatial reward learning (Supplementary Fig. 2g; Supplementary Tables 5–7). Altogether, these findings support the idea that the dHPC→NAc projection may represent one pathway through which hippocampal information is transformed into spatial goal-directed appetitive action.

### Enhanced spatial coding by dHPC→NAc neurons

Recent work on the role of the hippocampus in spatial memory has uncovered large-scale functional heterogeneity across anatomical axes, transcriptional differences, as well as projection patterns[33,36,37]. To understand if the observed role of the dHPC→NAc projection for spatial reward memory is subserved by a set of functionally distinct projection neurons, we used dual-color two-photon imaging in a subset of mice ($n = 6$). For this, we injected AAVrg-Cre in NAc and the static red fluorophore DIO-mCherry in dHPC (CA1/subiculum border region) of Thy1-GCaMP6s mice[38]. This approach allowed us to obtain dynamic calcium signals both in a large majority of mCherry-negative hippocampal neurons (dHPC⁻) and specifically mCherry co-expressing NAc-projecting neurons (dHPC→NAc), simultaneously within the same field of view using the same calcium indicator (Fig. 3a, Supplementary Fig. 3, and Supplementary Movie 2). It allowed us to overcome the technical challenge of directly recording from large populations of projection-specific neurons using electrophysiological methods[36,39–41]. Optical access to dorsal CA1 and prosubiculum (also known as proximal subiculum[42,43]) was established by implanting a chronic hippocampal window after virus injections[44,45] (Supplementary Fig. 1c). Imaging data from CA1 and prosubiculum were acquired after 5 days of behavioral training, were motion-corrected using NormCorre, and spatio-temporal components were extracted using constrained non-negative matrix factorization (CNMF)[46]. We thus obtained calcium signals for further analysis from a total of 5372 GCaMP-expressing neurons including 444 putative dHPC→NAc neurons in 6 mice across 19 imaging sessions (Fig. 3b and Supplementary Figs. 3 and 4). These numbers approximate previously established proportions of NAc-projecting neurons in dorsal prosubiculum and distal CA1[36,47,48]. To

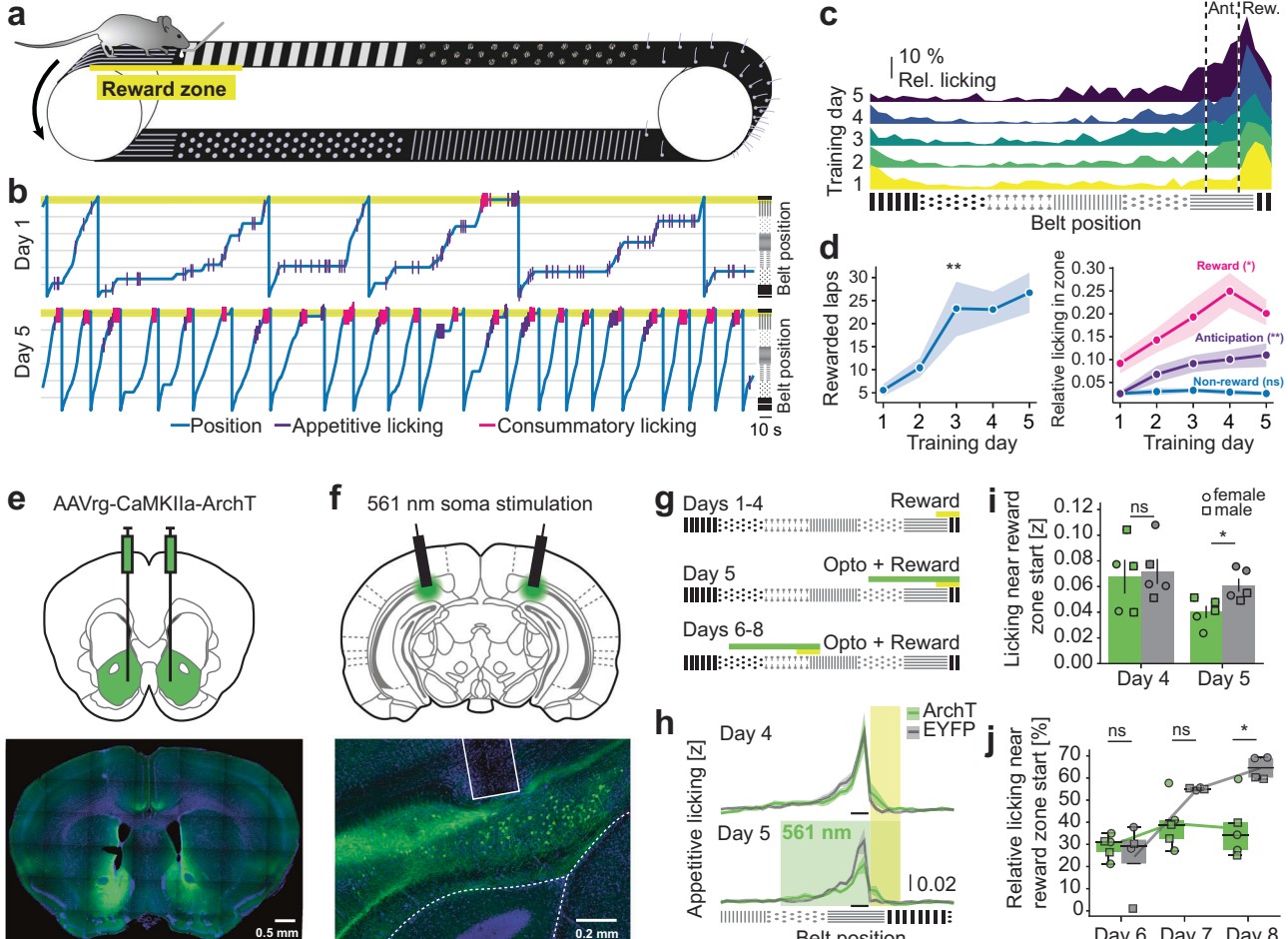

**Fig. 1 | The hippocampus-accumbens pathway is necessary for spatial reward memories. a–d** Training mice on a spatial reward learning task. **a** Schematic of behavioral task. **b** Representative traces from training days 1 and 5. **c** Licking in anticipation (Ant.) and reward (Rew.) zone increases over the course of training. **d** Number of rewarded laps per training session ($F(4) = 6.344$, $P = 0.0043$, GG-correction), anticipatory ($F(4) = 3.803$, $P = 0.0080$) and reward licking ($F(4) = 4.276$, $P = 0.0145$, GG-correction) significantly increased over the course of five training days ($n = 18$ mice, repeated-measures ANOVA). **e** Injection schematic (top) and axonal expression of ArchT-GFP in NAc (bottom). **f** Optic fiber placement (top) and somatic expression of ArchT-GFP in the dorsal subiculum; light fiber tracts indicated by white solid lines (bottom). **g** Mice were trained to lick in a reward zone for four days. On day 5, optogenetic stimulation was provided around the reward zone. On days 6–8, optogenetic stimulation and reward zones were shifted by 180 cm. **h** Position-binned appetitive lick traces around reward zone on day 4 without stimulation (top) and day 5 with stimulation (bottom) for ArchT-injected mice

(green lines) and control mice (gray lines). Yellow bar: reward zone; green rectangle: light stimulation; black horizontal bars: time points compared in (**i**). **i** Appetitive licking near reward zone start is not statistically significant between groups on day 4 without light stimulation ($t(8) = 0.2611$; $P = 1$) but on day 5 with light stimulation ($t(8) = 2.874$; $P = 0.041$). Mixed ANOVA, main effect group $F(1, 8) = 7.316$, $P = 0.0269$, main effect day $F(1, 8) = 13.59$, $P = 0.0062$, interaction $F(1, 8) = 0.646$, $P = 0.445$, $n = 10$ mice. **j** Learning of a novel reward zone is impaired with optogenetic inhibition. Relative licking near reward zone start is significantly reduced on day 8 ($t(5.349) = 4.076$, $P = 0.0250$). Mixed ANOVA, main effect group $F(1, 7) = 5.909$, $P = 0.0454$, main effect day $F(1, 14) = 19.062$, $P = 0.0093$, interaction $F(1, 14) = 8.314$, $P = 0.0042$, $n = 9$ mice. All post-hoc tests two-tailed Bonferroni-corrected Welch's $t$-tests. All data are presented as mean ± SEM. ns: not significant, $*P < 0.05$, $**P < 0.01$, circular data points: female, square data points: male. Source data are provided as a Source Data file.

circumvent potential issues correlating time-sensitive behaviors with slowly decaying calcium transients, we used deconvolved calcium events ("S") for further analysis.

Given the dHPC's well-established role in representing spatial information and the role of dHPC→NAc projections in spatial memory tasks[22,29], we tested if and how spatial information may be encoded by dHPC→NAc neurons compared to dHPC⁻ neurons. For this, we extracted calcium events for each neuron and compared this activity across the linear spatial environment of the treadmill belt as mice traversed it lap by lap. Indeed, we found large numbers of both dHPC⁻ and dHPC→NAc neurons with repeatedly elevated calcium levels at the same positions on the belt (Fig. 3c–e and Supplementary Fig. 4). Comparing each neuron's spatial information content[49] with that of a randomly shuffled distribution, we identified a significantly higher proportion of such place cells in dHPC→NAc neurons (169/444 neurons; 38%) compared to

dHPC⁻ neurons (1581/4928 neurons; 32%; $\chi^2$, $P = 0.012$; Fig. 3f and Supplementary Fig. 5a). These proportions were not different between mice ($\chi^2(4) = 0.24$, $P = 0.99$). Within this population of place cells, we further analyzed how specifically space is encoded, and found that dHPC→NAc place cells had a higher spatial information rate[49] (median 1.46 vs 1.04, $P < 0.001$; Fig. 3g and Supplementary Fig. 5b). They also had significantly higher levels of sparsity (median 0.29 vs 0.23, $P < 0.001$; Fig. 3h and Supplementary Fig. 5c), a measure for how diffuse a neuron is firing in the spatial domain[50]. Furthermore, the reliability of in-place field activity per lap (median 0.57 vs 0.50, $P = 0.011$), place field stability (lap-to-lap correlation; median 0.33 vs. 0.23, $P < 0.001$), and the place field's relative calcium activity (activity inside the place field – activity outside the place field; median 9.29 vs 7.64, $P = 0.0066$) were significantly higher for dHPC→NAc place cells compared to dHPC⁻ neurons (Fig. 3i–k and Supplementary Fig. 5d–f). In line

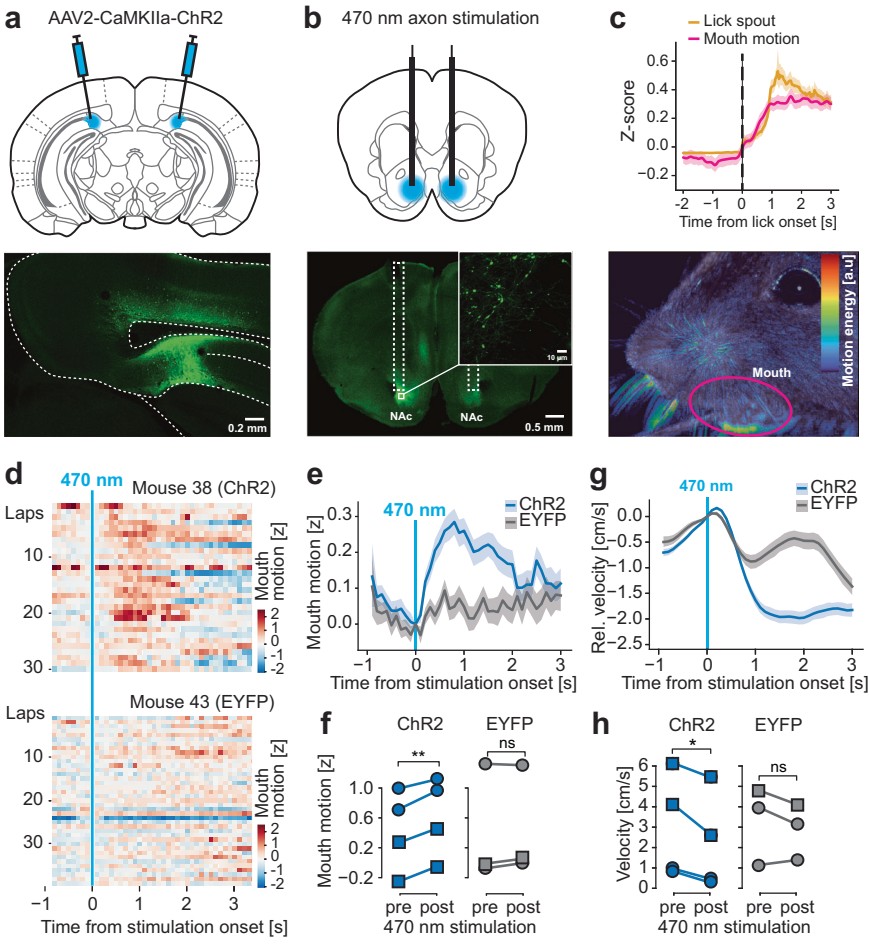

**Fig. 2 | Stimulation of the dorsal hippocampus-accumbens pathway induces lick-related mouth motion and deceleration. a** Injection schematic (top) and somatic expression of ChR2-EYFP in dorsal prosubiculum (bottom). 200 nl of AAV2-CaMKIIa-ChR2/EYFP were injected bilaterally. **b** Optic fiber placement (top) and axonal expression of ChR2-EYFP in the NAc, where light fibers are placed bilaterally (tracts indicated by white dotted lines; bottom). **c** Average motion energy in mouth region (magenta line) increases with licking. Golden line shows analog lick spout signal (top). Tracking of orofacial movements via infrared camera recordings. False-color coded motion energy (pixel-by-pixel intensity difference to previous image). Automatically segmented mouth region is indicated by a magenta line (bottom). **d** Representative examples of mouth motion around onset of optogenetic stimulation in an animal expressing ChR2 (top) or EYFP control (bottom). **e** Trial-averaged mouth motion activity around time of optogenetic stimulation in ChR2 (blue) and EYFP (gray)-expressing mice. **f** Mouth motion is

significantly increased with optogenetic stimulation in ChR2 animals ($t(3) = 7.485$; $P = 0.00494$) but not EFYP animals ($t(2) = 1.353$; $P = 0.309$). Mixed ANOVA, main effect group $F(1, 5) = 0.036$, $P = 0.856$, main effect stimulation $F(1, 5) = 42.47$, $P = 0.0013$, interaction $F(1, 5) = 14.72$, $P = 0.0122$, followed by two-tailed post-hoc Bonferroni-corrected $t$-tests ($n = 7$ mice). **g** Trial-averaged relative velocity around time of optogenetic stimulation in ChR2 (blue) and EYFP (gray)-expressing mice. **h** Velocity is significantly decreased with optogenetic stimulation in ChR2 animals ($t(3) = -3.551$; $P = 0.0381$) but not EFYP animals $t(2) = -1.263$; $P = 0.334$. Mixed ANOVA, main effect group $F(1, 5) = 0.112$, $P = 0.752$, main effect stimulation $F(1, 5) = 12.31$, $P = 0.017$, interaction $F(1, 5) = 2.80$, $P = 0.155$, followed by two-tailed post-hoc Bonferroni-corrected $t$-tests ($n = 7$ mice). All data are presented as mean ± SEM. ns: not significant, *$P < 0.05$, **$P < 0.01$, circular data points refer to female mice, square data points to male mice. Source data are provided as a Source Data file.

with this, we also found smaller overdispersion in dHPC→NAc place cells ($\sigma^2 = 2.04$ vs. 2.42, $P < 0.001$; Supplementary Fig. 5g; Supplementary Table 4). These results suggest that the dHPC routes enhanced and more reliable spatial information to NAc compared to the general dHPC population, in line with previous results pointing towards the necessity of the dHPC→NAc projection for spatial memory expression[22,29].

## Place fields are modulated by local cue boundaries and are overrepresented near the reward zone

Previous studies found that place fields are often not homogeneously distributed across the environment but can be modulated by salient environmental features such as textures, borders, or reward/goal zones[14,51,52]. We hypothesized that information about such spatial features may be preferentially routed to NAc, given the relevance of NAc for spatial memory[27,28] and particularly for reward-related behaviors[53]. Upon inspection of place cells' spatially binned calcium profiles

(Fig. 4a, b and Supplementary Figs. 4, 5h–i), we noticed an overabundance of place fields that seem to cover entire texture zones, starting at the beginning of a texture zone and ending before the next texture zone begins. To quantify this observation, we plotted the densities of place field start and end positions across the belt and compared the observed densities with those of randomly shuffled place fields. We found a significant overrepresentation of place field start and end positions (but not centers) near texture boundaries for both dHPC- and dHPC→NAc populations (ratios >99.9th percentile, permutation test). This effect was more pronounced in dHPC→NAc neurons (PF start: $P = 0.033$, PF end: $P = 0.036$, $\chi^2$; Fig. 4c, d and Supplementary Fig. 5j–m). This suggests that dHPC→NAc neurons are more strongly modulated by local cue boundaries.

Previous studies have shown that reward zones and their preceding vicinity are often over-represented by place cells[33,54,55]—we hypothesized that this effect might involve dHPC→NAc neurons, given the NAc's role in reward-related behaviors[53]. When pooled across all

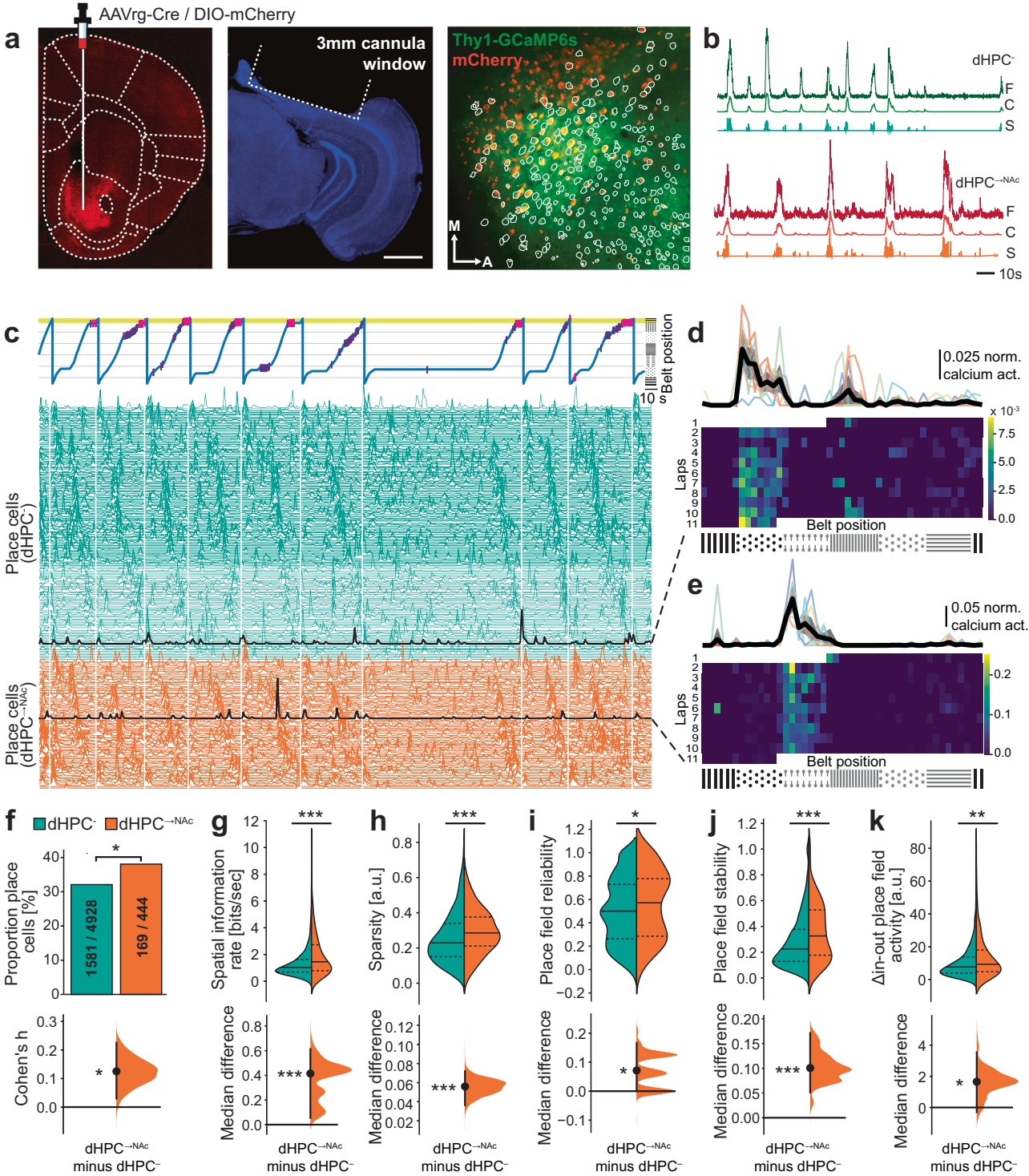

Nature Communications | (2024)15:3196                                                                                  5

sessions, we found little evidence of such an overrepresentation (Supplementary Fig. 5n, o). However, mouse behavior may show high variability, and previous work has demonstrated the importance of behavioral engagement for spatial representations[56] and the dependence of spatial reward overrepresentations on individual task success and reward expectation[33,54,57]. We thus divided sessions into high- and low-success based on lick performance (lick precision and reward dispensation, see Methods) and found significantly more dHPC→NAc place cells near the reward zone (reward and anticipation zones) in high-success sessions compared to low-success sessions ($P = 0.0036$, Welch's $t$-test). At the same time, dHPC⁻ neurons showed only a trend towards significance using $t$-statistics ($P = 0.051$, Welch's $t$-test;

Fig. 4e, f) that reached significance using estimation statistics (Supplementary Fig. 5p). Based on this, we conclude that both populations overrepresent the reward zone during high-success trials, but this effect is particularly pronounced in dHPC→NAc neurons. In line with previous studies, both neuronal populations also showed a strong correlation between the success rate (percentage of rewarded laps) and the proportion of place fields near the reward zone across sessions ($r = 0.62$; Supplementary Fig. 5q).

If reward zone information is preferentially encoded in dHPC→NAc populations and NAc neurons play an integral part in reward-related behaviors, we would expect linear decoders to identify reward or anticipation zones more accurately based on dHPC→NAc activity

**Fig. 3 | dHPC$^{\rightarrow NAc}$ neurons carry enhanced and more reliable spatial information. a** Schematic of dual-color projection neuron imaging method. Thy1-GCaMP6s mice were injected with AAVrg-Cre in the medial NAc and DIO-mCherry in dHPC. Representative coronal brain slice showing axonal mCherry expression in NAc (AP −1.3; left). Representative coronal brain slice stained with DAPI (blue) of dHPC; scale bar represents 1 mm (second left). Field of view of one sample experiment showing Thy1-GCaMP6s expression in green and mCherry expression of putative NAc-projecting neurons in red; outlines show detected components used for analysis (right). **b** Raw (F), denoised and deconvolved (C) and event (S) traces from two representative neurons; red traces indicate mCherry co-expression (dHPC$^{\rightarrow NAc}$). **c** Excerpt from one representative recording session showing behavioral activity (top) and denoised neural activity of identified place cells (bottom). White vertical lines mark new laps. Traces are ordered according to place fields (red: dHPC$^{\rightarrow NAc}$; green: dHPC$^{-}$). Black traces are example neurons shown in **d, e. d, e** Spatial activity of one neuron without (**d**) and with (**e**) mCherry co-expression over one session.

**f–k** NAc-projecting neurons (red) contain a higher proportion of place cells (**f**, $\chi^2(1, 5372) = 6.364$, $P = 0.0116$); these place cells contain more spatial information per second (**g**, Mann-Whitney $U = 161846$, $P < 0.001$), show increased sparsity (**h**, Mann-Whitney $U = 160810$, $P < 0.001$), higher reliability (**i**, probability of firing maximally within their place field per lap; Mann-Whitney $U = 149324.5$, $P = 0.0112$), higher place field stability (**j** lap-to-lap correlations; Mann-Whitney $U = 162556$, $P < 0.001$), and higher in-place field activity (**k** Mann-Whitney $U = 150460.0$, $P = 0.0066$). $n = 6$ mice, 19 imaging sessions, 5372 (inc. 444 mCherry-coexpressing) neurons, 1750 (inc. 169 mCherry-coexpressing) place cells; all tests two-tailed. Data in **g–k**: $n = 5/3$ mice with place cells. Solid lines in violin plots represent median values and dashed lines quartiles. Bottom panels in **g–k** show median difference computed from 5000 bootstrapped resamples (black dot, median; black ticks, 95% confidence interval; filled curve, sampling-error distribution). *$P < 0.05$, **$P < 0.01$, ***$P < 0.001$. Source data are provided as a Source Data file.

compared to that of dHPC$^{-}$. To test this, we trained a linear classifier based on a support vector machine (SVM) with calcium activity and reward zone information on odd/even laps and tested decoding accuracy on even/odd laps, using either dHPC$^{-}$ or sample size-matched dHPC$^{\rightarrow NAc}$ populations (see Methods). We found that, within individual sessions, decoding accuracy of the reward anticipation zone was significantly enhanced for dHPC$^{\rightarrow NAc}$ populations ($P = 0.0195$, Wilcoxon's test; Fig. 4g and Supplementary Fig. 5r). These findings demonstrate significant modulation of dHPC$^{\rightarrow NAc}$ neurons by local cue boundaries and enhanced reward zone coding.

### Enhanced coding of low velocities in dHPC$^{\rightarrow NAc}$ neurons

As correct performance in the spatial reward learning task is associated with a reduction in velocity and an increase in licking near the reward zone (see Supplementary Fig. 1k–n), we wondered if such non-spatial task-relevant behavioral features were encoded by dHPC neurons and its projections to the NAc[11,58,59]. We hypothesized that the NAc may have privileged access to information on low velocities as mice generally slow down near the reward zone, presumably to allow for better discrimination and to engage in anticipatory licking.

In line with previous analyses of speed coding in hippocampal and parahippocampal regions[13,60], we averaged each neuron's calcium activity per velocity bin from 2 to 30 cm/s and regressed this activity against velocity. Neurons with a significant regression model (after correcting for false discovery rate) and positive slope were classified as speed-excited (Fig. 5a, b). We found that approximately 15% of neurons were positively speed-modulated, with comparable proportions between dHPC$^{-}$ and dHPC$^{\rightarrow NAc}$ neurons (13 % vs. 16%, $P = 0.109$, $\chi^2$; Fig. 5c, d and Supplementary Fig. 6a). We also identified neurons with a significant velocity regression but a negative slope (Fig. 5e, f). We found approximately 15% of such speed-inhibited neurons, with a significantly larger proportion among the dHPC$^{\rightarrow NAc}$ population (15% vs. 21%, $P = 0.00022$, $\chi^2$; Fig. 5g, h and Supplementary Fig. 6b). These proportions were not different between mice (speed-excited: $\chi^2(4) = 0.34$, $P = 0.99$; speed-inhibited:: $\chi^2(5) = 0.63$, $P = 0.99$). For both speed-modulated populations, proportions of preferred speed bins were not different between projection populations (Supplementary Fig. 6c, d). These results suggest widespread modulation of dHPC neuronal activity by non-spatial features such as velocity, with dHPC$^{\rightarrow NAc}$ neurons specifically overrepresenting low speeds.

### Overrepresentation of appetitive licking-excited dHPC$^{\rightarrow NAc}$ neurons

Besides a decrease in velocity when approaching the reward zone, mice also increasingly engaged in licking behavior (see Fig. 1b–d). Given the NAc's dual role in appetitive and consummatory behaviors[61], we tested whether licking behaviors might be reflected in the neural activity of dHPC$^{\rightarrow NAc}$ neurons. For this, we distinguished between consummatory licking which occurs after a reward is dispensed and allows

the mouse to consume the reward provided, and appetitive licking which is an operant behavior that will lead to reward dispensation when performed at the correct location on the belt. We found a significant decrease of calcium activity during reward consumption in both dHPC$^{-}$ and dHPC$^{\rightarrow NAc}$ populations (Supplementary Fig. 7a–c, g–j).

Appetitive licking, on the other hand, had no apparent effect on neural activity in dHPC$^{-}$ neurons but coincided with a small but significant increase in calcium activity in the dHPC$^{\rightarrow NAc}$ population (Supplementary Figs. 7d–f, 8). We investigated if this population-averaged data is reflected on the single-cell level and if there are individual cells that are reliably modulated by appetitive licking (Fig. 6a, b; Supplementary Figs. 8, 9; Supplementary Movie 3). Comparing the pre- and post-lick neural activity for each neuron for each appetitive lick event, we identified a total of 1268 neurons (24%) that were significantly (negatively or positively) modulated by appetitive licking (Fig. 6c, d). We termed these neurons "lick-inhibited" or "lick-excited", respectively (while noting that "inhibited" neurons are simply showing less average calcium activity immediately after lick onset compared to before, considering that deconvolved calcium activity cannot become negative). Interestingly, we found a significantly larger proportion of lick-excited neurons in dHPC$^{\rightarrow NAc}$ populations (3.8% vs 7.4%, $P < 0.001$, $\chi^2$; Fig. 6e), while the proportion of lick-inhibited neurons was comparable between dHPC$^{-}$ and dHPC$^{\rightarrow NAc}$ populations (19.7% vs 17.1%, $P = 0.20$, $\chi^2$; Fig. 6f). These findings suggest that dHPC does not route reward information per se to NAc, but rather information on appetitive behaviors required to obtain such rewards.

### Enhanced conjunctive coding of space, velocity, and appetitive behaviors in dHPC$^{\rightarrow NAc}$ neurons

We identified cells modulated by space, velocity, and appetitive licking. Previous studies suggested that individual hippocampal neurons do not necessarily exclusively code for one single feature but are instead able to conjunctively encode various environmental properties[11,58,62,63]. Such conjunctive coding may be particularly relevant for downstream linear decoders to select task-appropriate actions[64,65]. We thus investigated speed and lick modulation of projection-specific place cells and interactions between velocity and lick modulation.

We first analyzed speed coding in dHPC$^{\rightarrow NAc}$ place cells and compared it to the dHPC$^{-}$ population (Fig. 7a, b and Supplementary Fig. 10a, b). We found about one third of hippocampal place cells were also speed-inhibited, in contrast to only about 7% of non-place cells. This effect was particularly pronounced in dHPC$^{\rightarrow NAc}$ place cells (43 % vs. 31%, $P = 0.0027$, $\chi^2$, Fig. 7b). Conversely, place cells were significantly less likely to be speed-excited than non-place cells (10% vs. 15%, $P < 0.001$, $\chi^2$, Fig. 7b), an effect that was again more pronounced in dHPC$^{\rightarrow NAc}$ neurons. This shows that dHPC place cells, and in particular those projecting to NAc, are more likely to be speed-inhibited, and less likely to be speed-excited.

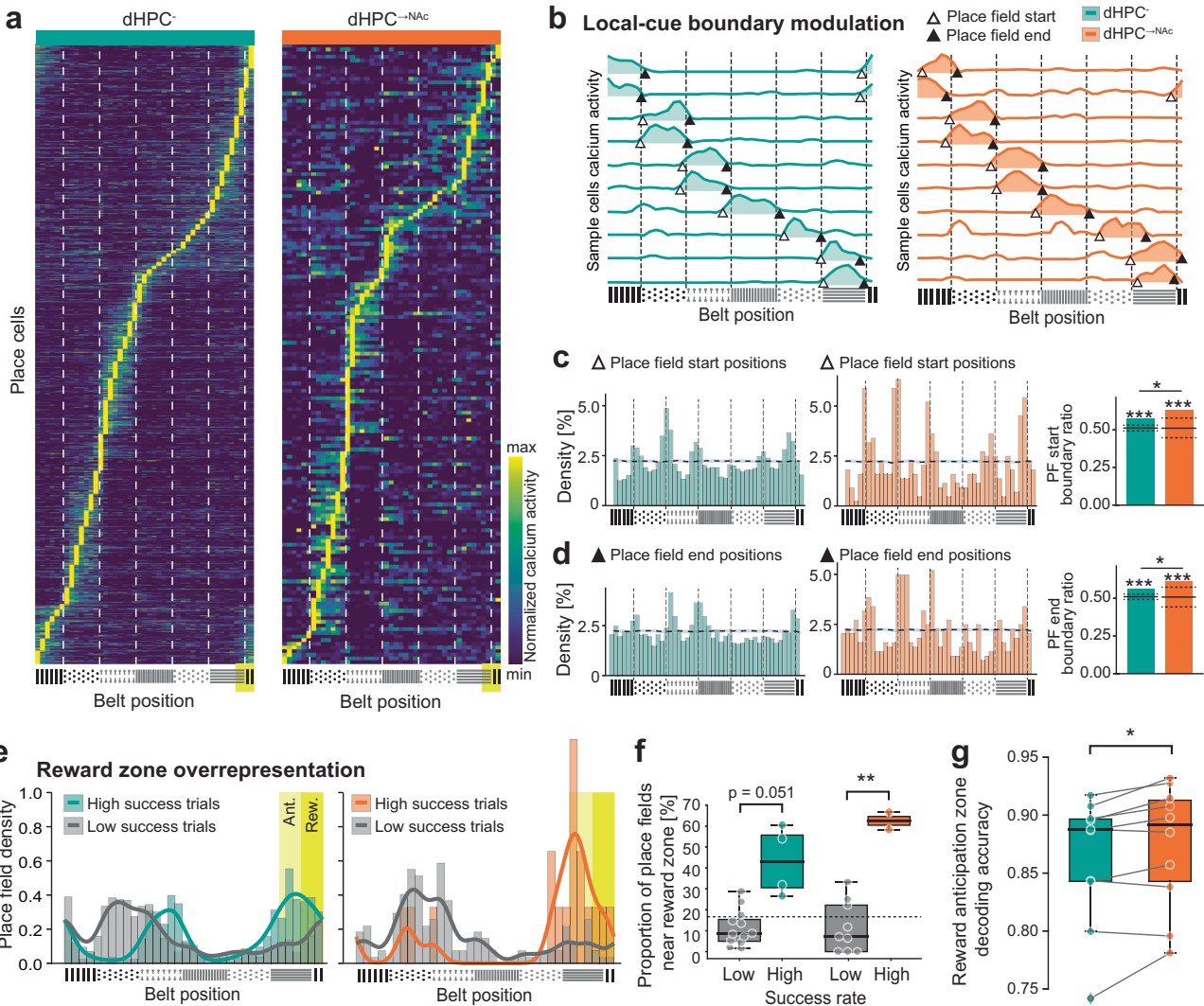

**Fig. 4 | dHPC→NAc place fields are modulated by local cues and reward zone.**
**a** Heat maps of dHPC⁻ (left) and dHPC→NAc (right) place cells' average calcium events, ordered by place field location. White dashed lines: texture boundaries; yellow rectangle: reward zone. **b** Example place fields with edges near belt texture boundaries. Triangles mark start (no fill) and end (fill) points. Dashed black lines: texture boundaries. **c**, **d** Place field edges accumulate near texture boundary areas. Histograms of dHPC⁻ (green) and dHPC→NAc (red) neurons' place field start **c** and end **d**. Dotted line and shade: average and 95th CI of 1000× randomly shuffled place fields. Both dHPC⁻ and dHPC→NAc place field start and end positions are significantly overrepresented at the 99.9th percentile (dotted black lines) compared to a randomly shuffled distribution (mean represented by horizontal black line). Start ($\chi^2$(1, 5332) = 5.136, $P$ = 0.0234) and end positions ($\chi^2$(1, 5332) = 3.869, $P$ = 0.0492) of dHPC→NAc place fields are also significantly overrepresented compared to the dHPC⁻ population; chi-squared test, $n$ = 6 mice, 5372 neurons. **e**, **f** Place cells are over-represented near reward zone in high success trials. **e** Histograms (bars) and kernel density estimations (KDEs; lines) of place field centers for dHPC⁻ (left) and dHPC→NAc (right) neurons, split into high success trials (green/red) and low success trials

(gray). Reward zone (Rew.; yellow, 30 cm) and anticipation zone (Ant.; bright yellow, 30 cm) are indicated as rectangles. **f** Proportion of place fields in reward and anticipation zone (i.e., near reward zone) is significantly higher in high-success trials (colored bars) compared to low-success trials (gray bars) in NAc-projecting neurons (red) but not in dHPC⁻ neurons (green). 2-way mixed ANOVA, $F_{success}$(1,1) = 54.918, $P$ < 0.001, $F_{projection}$(1,1) = 0.958, $P$ = 0.338, $F_{interaction}$(1,1) = 2.969, $P$ = 0.098. Two-tailed post-hoc Welch's $t$-tests with Bonferroni correction: $t_{dHPC⁻}$(3.561) = 3.698, $P$ = 0.0512; $t_{dHPC→NAc}$(3.479) = 8.671, $P$ = 0.0036; $t_{low\_success}$(13.629) = 0.093, $P$ = 1; $t_{high\_success}$(3.952) = 2.075, $P$ = 0.215, $n$ = 16 imaging sessions. Dashed line: even distribution of reward and anticipation zone. **g** A linear classifier shows significantly increased decoding accuracy of reward anticipation zone based on dHPC→NAc neural activity compared to that of sample size-matched dHPC⁻ neurons. Wilcoxon's $t$-test, $W$(9) = 5.0, $n$ = 10 imaging sessions, $P$ (two-tailed) = 0.020. Box-and-whisker plots in **f**, **g** show quartiles represented by the box and outlier-corrected minima and maxima by the whiskers. *$P$ < 0.05, **$P$ < 0.01, ***$P$ < 0.001. Source data are provided as a Source Data file.

We next analyzed lick modulation of place cells and, surprisingly, found that the previously observed lick-related increase in calcium activity (Fig. 5b) was largely carried by place cells and not by non-place cells (Fig. 7c). This effect seems to be mostly carried by lick-excited neurons that are significantly overrepresented in place cells compared to non-place cells (8% vs. 2%, $P$ < 0.001, $\chi^2$), particularly in dHPC→NAc neurons (15% vs. 3%, $P$ < 0.001, $\chi^2$; Fig. 7d and Supplementary Fig. 10c, d). Lick-inhibited neurons, on the other hand, were distributed equally between place and non-place dHPC⁻ and dHPC→NAc neurons.

These findings show that a large majority of lick-excited neurons, especially in the dHPC→NAc population, also code for spatial information.

Finally, we investigated interactions between lick and velocity modulation. Comparing velocity correlations of lick-excited and lick-inhibited neurons, we observed a clear skew of lick-excited cells to have more negative velocity correlations and lick-inhibited cells to have more positive velocity correlations, visible in both dHPC⁻ and dHPC→NAc populations (Fig. 7e). This results in significantly more

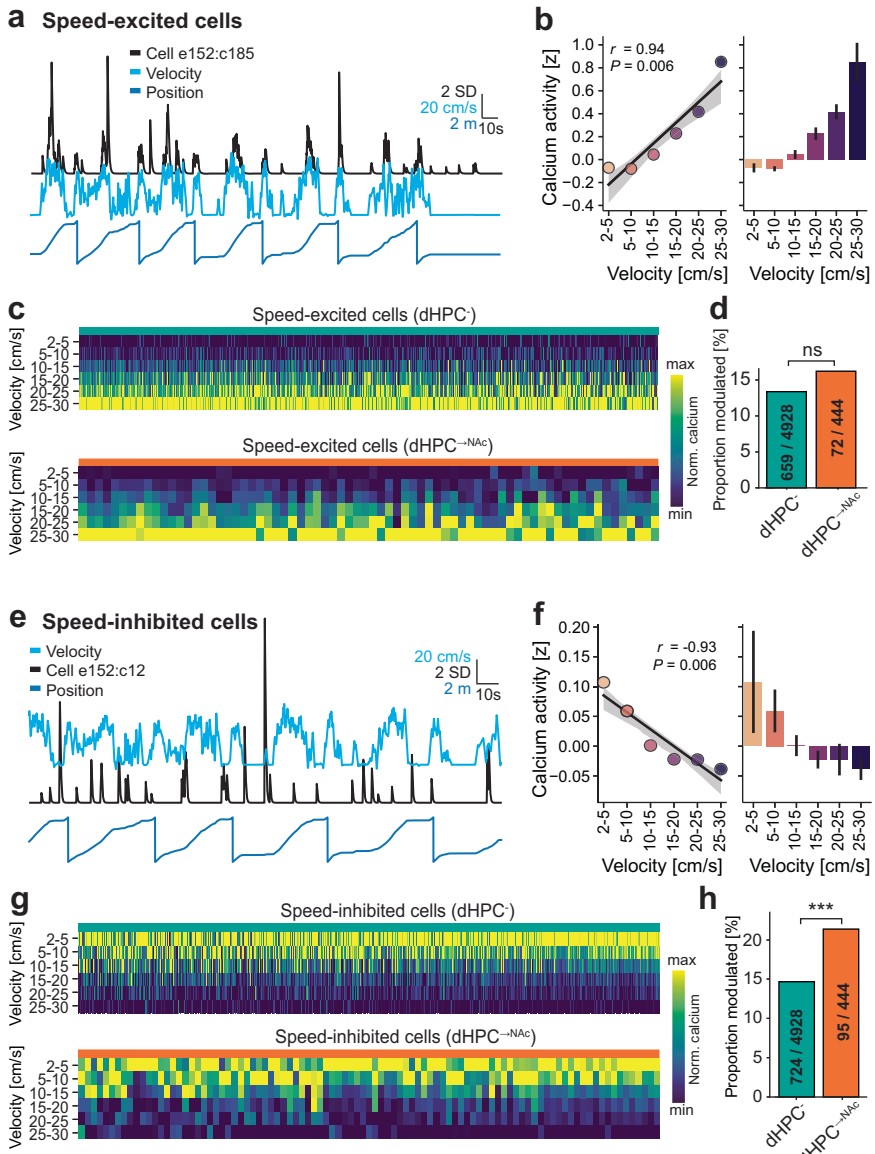

**Fig. 5 | Speed-inhibited cells are overrepresented in dHPC→NAc neurons.**
**a**–**d** Speed-excited dHPC neurons. **a**, **b** Representative example of one speed-excited neuron. **a** Sample traces of velocity, position and one neuron's denoised calcium activity. Note the increased calcium activity in times of high velocity, irrespective of position. **b**, Linear regression on average calcium events per velocity bin shows a significant positive relationship (slope = 7.43 × 10⁻⁴, intercept = −2.097×10⁻³, $r$ = 0.937, $P$ = 0.0058, $n$ = 18,000 time points). **c** Heatmaps of speed-binned normalized calcium activity of all significantly positively speed-modulated dHPC⁻ (top) and dHPC→NAc (bottom) neurons. **d** Proportions of speed-excited neurons are comparable between dHPC⁻ and dHPC→NAc populations ($\chi^2$(1, 5372) = 2.565, $P$ = 0.109, chi-squared test). **e**–**h** Speed-inhibited dHPC neurons. **e**, **f** Representative example of one speed-inhibited neuron. **e** Sample traces of velocity, position and one neuron's denoised calcium activity. Note the increased calcium activity in times of low velocity, irrespective of position. **f** Linear regression on average calcium events per velocity bin shows a significant negative relationship (slope = −1.0437 × 10⁻⁴, intercept = 2.814 × 10⁻³, $r$ = −0.933, $P$ = 0.0065, $n$ = 18,000 time points). **g** Heatmaps of speed-binned normalized calcium activity of all significantly negatively speed-modulated dHPC⁻ (top) and dHPC→NAc (bottom) neurons. **h** Negatively tuned neurons are overrepresented in the NAc-projecting population ($\chi^2$(1, 5372) = 13.66, $P$ = 0.00022, chi-squared test). Speed-modulated neurons were classified as showing a significant linear regression at $P$ < 0.05 after Benjamini/Hochberg FDR correction. All data are presented as mean ± SEM. ns: not significant, ***$P$ < 0.001. Source data are provided as a Source Data file.

speed-inhibited cells to be lick-excited compared to speed-excited cells (11% vs. 2%, $P$ < 0.001, $\chi^2$), an effect that was even more pronounced in dHPC→NAc neurons (20% vs. 1%, $P$ < 0.001, $\chi^2$; Fig. 7f and Supplementary Fig. 10e, f). Conversely, speed-excited neurons were much more likely to be lick-inhibited than speed-inhibited neurons (40% vs. 6%, $P$ < 0.001, $\chi^2$, Fig. 7f). These results demonstrate that there is a strong inverse relationship between lick and velocity modulation.

One caveat of such conjunctive coding analyses is that in our behavioral task, trained mice often show highly stereotypical

behavior, such that mice would mostly lick at one location where they would also slow down (see Supplementary Fig. 1k−n). In light of this, conjunctive coding could be an epiphenomenon of collinear behavioral features. To account for this collinearity, we modeled the influence of three key behavioral features (space, velocity, and appetitive licking) on the activity of each neuron by building a generalized linear model for each neuron (GLM; Fig. 8a). On average, we found that our models could explain close to 40% of the variance observed in our test datasets (Supplementary Fig. 11a, d), with dHPC→NAc neurons showing increased feature importance for position

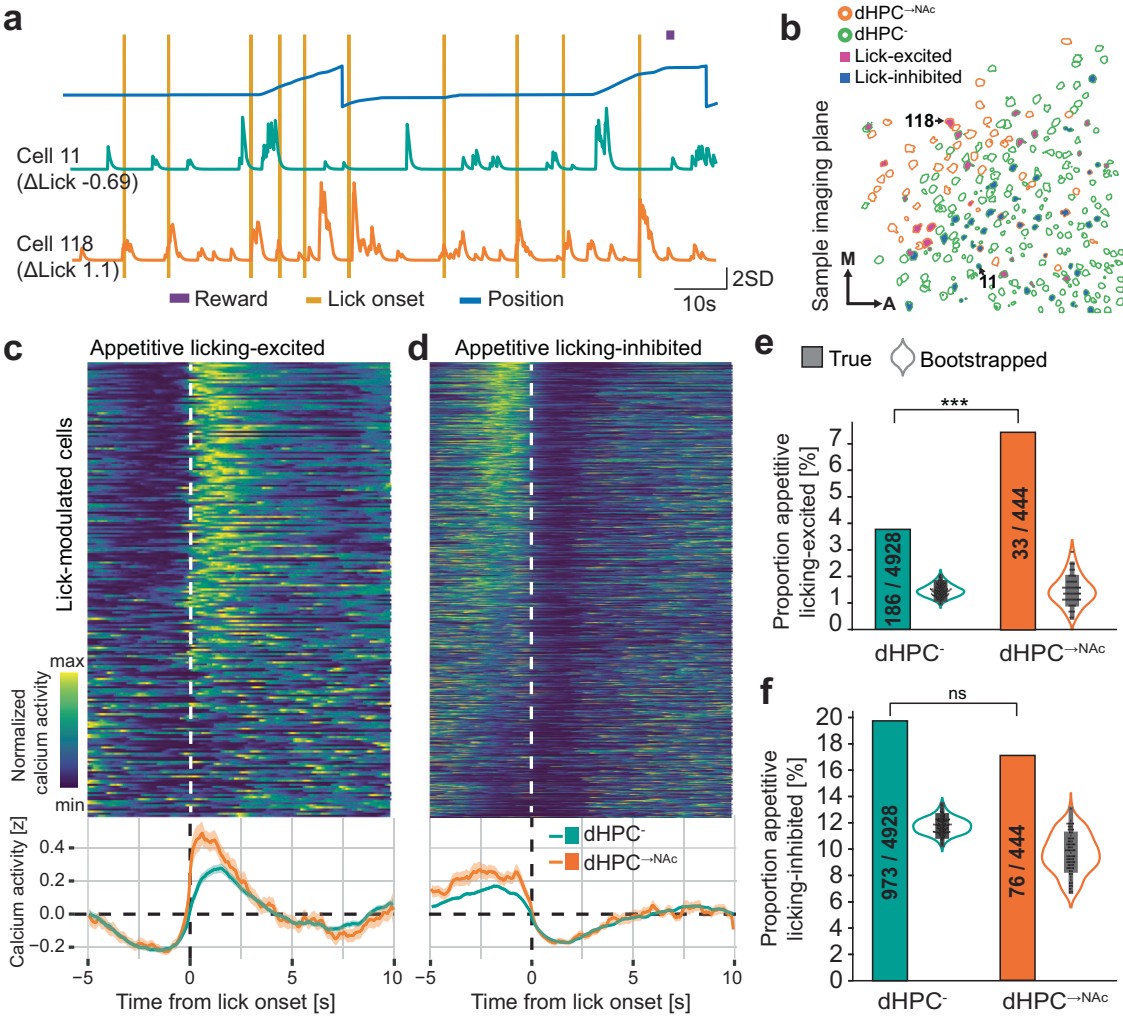

**Fig. 6 | dHPC→NAc neurons are over-represented in lick-excited neurons.**
**a**, **b** Example imaging session and traces showing lick-excited and lick-inhibited neurons. **a** Behavioral traces and calcium activity of sample neurons #11 (lick-inhibited) and #118 (lick-excited). **b** Field of view showing spatial profiles of dHPC⁻ (green outlines) and dHPC→NAc (red outlines), some of which are classified as lick-excited (violet fill) or lick-inhibited (dark blue fill); neurons #11 and #118 are high-lighted. M medial, A anterior. **c**, **d** Heatmaps (top) and event-triggered calcium activity averages (bottom) of neurons classified as appetitive licking-excited **c** and licking-inhibited **d**. **e** Proportion of appetitive licking-excited neurons is

significantly higher than that expected by chance and higher in NAc-projecting neurons; $\chi^2(1, 5372) = 13.018$, $P = 0.00031$, $n = 5372$ neurons, chi-squared test. **f** Proportion of appetitive licking-inhibited neurons is significantly higher than that expected by chance but not different between populations; $\chi^2(1, 5372) = 1.626$, $P = 0.202$, $n = 5372$ neurons, chi-squared test. Violin plots in **e**, **f** represent popula-tion proportions of 100× randomly shifted lick traces. Box-and-whisker plots in **e**, **f** show quartiles represented by the box and outlier-corrected minima and maxima by the whiskers. All data are presented as mean ± SEM. ns: not significant, *$P < 0.05$, ***$p < 0.001$. Source data are provided as a Source Data file.

and licking (Supplementary Fig. 11b). To determine significant con-tributions of the three behavioral features, we next built 3 × 100 models in which one of the behavioral features was randomly shuf-fled against time, and compared the variance explained to the ori-ginal model[66] (Fig. 8a). Thus, if neural activity spuriously coincided with the activity of one behavioral feature that could be similarly explained by a largely collinear behavioral feature, the model's pre-dictive performance should be mostly unaffected by this shuffling. For this, we first investigated the average difference of the shuffled models compared to the full model for each neuron ($\Delta R^2$). The average drop in variance explained to the full model was significantly stronger in dHPC→NAc neurons compared to dHPC⁻ for position and velocity but not licking (Supplementary Fig. 11c). We classified sig-nificant modulation as behavioral features whose shuffling led to a reduction in variance explained in more than 95% of shuffled models. We found that cells thus encoding space, velocity, and licking were overrepresented in dHPC→NAc compared to dHPC⁻ neurons (space: $P < 0.001$; velocity: $P < 0.001$; licking: $P = 0.0015$; $\chi^2$; Fig. 8b and

Supplementary Fig. 11e–g). Importantly, we also found significantly increased proportions of conjunctive coding for all three feature combinations as well as triple-conjunctive neurons in dHPC→NAc neurons compared to dHPC⁻ (all combinations $P < 0.001$, $\chi^2$; Fig. 8c, d and Supplementary Fig. 11h–k). This results in a significantly higher proportion of conjunctive coding neurons in dHPC→NAc neurons compared to dHPC⁻ (44% vs. 19%, $P < 0.001$, $\chi^2$; Fig. 8e).

As conjunctive coding has been suggested to aid downstream linear decoders to select task-appropriate actions[58,64], we wondered if this increased conjunctive code might allow linear decoders to identify the presence of the reward zone more correctly in our task. We thus trained an SVM-based linear classifier on each trial's odd/even laps' reward zones and tested the decoding accuracy on even/odd laps, based on conjunctive or randomly sample size-matched non-con-junctive coding neurons (Fig. 8f). We found enhanced reward zone decoding accuracy for conjunctive compared to non-conjunctive coding neurons ($P < 0.001$, Wilcoxon's test; Fig. 8g). Thus, our data suggest that the enhanced conjunctive coding observed in dHPC→NAc

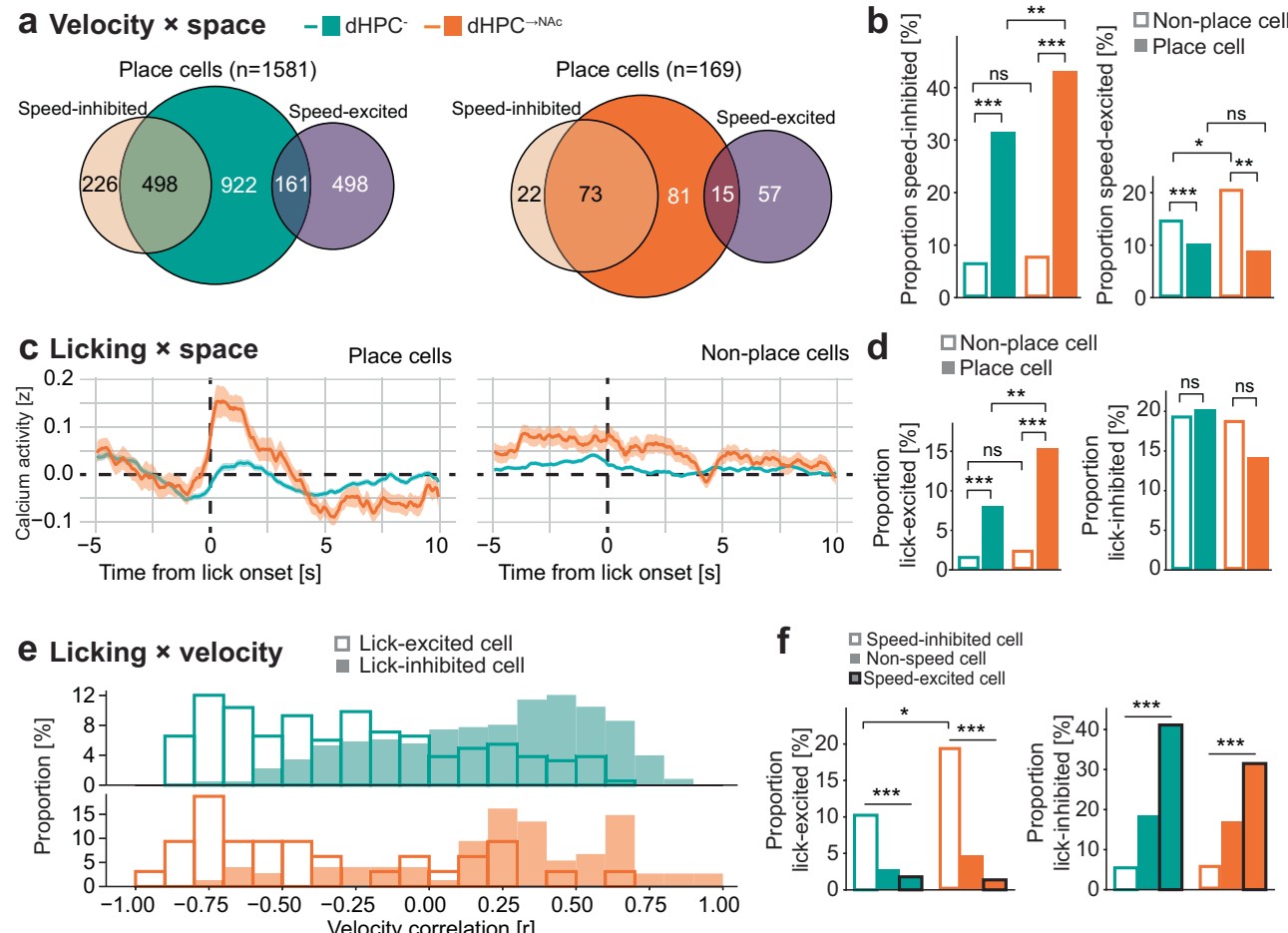

**Fig. 7 | Enhanced conjunctive coding of space, velocity, and licking in NAc-projecting neurons. a**, **b** Conjunctive coding of space and velocity. **a** Venn diagrams of dHPC⁻ (left) and dHPC→NAc (right) place cells and their overlaps with negatively (ocher) and positively (purple) tuned speed cells. Shown are absolute numbers of neurons classified. **b** Proportions of dHPC⁻ (green) and dHPC→NAc (red) place cells (fill) and non-place cells (no fill) that are also significantly speed-modulated. A larger proportion of place cells is speed-inhibited compared to non-place cells (dHPC⁻: $\chi^2(1, 4928) = 522.71$, $P < 0.001$; dHPC→NAc: $\chi^2(1, 444) = 75.02$, $P < 0.001$). dHPC→NAc place cells also have a higher proportion of speed-inhibited cells than dHPC⁻ place cells ($\chi^2(1, 1750) = 10.219$, $P = 0.0014$). Speed-excited neurons are overrepresented in non-place cells compared to place cells (dHPC⁻: $\chi^2(1, 4928) = 22.048$, $P < 0.001$; dHPC→NAc: $\chi^2(1, 444) = 10.497$, $P = 0.0012$). dHPC→NAc non-place cells also have a higher proportion of speed-excited cells than dHPC⁻ non-place cells ($\chi^2(1, 3622) = 6.223$, $P = 0.0126$). **c**, **d** Conjunctive coding of space and licking. **c** Event-triggered average calcium traces for dHPC⁻ (green) and dHPC→NAc (red) place cells (left) and non-place cells (right). **d** Proportions of lick-excited neurons are significantly enriched in dHPC⁻ ($\chi^2(1, 4928) = 114.515$, $P < 0.001$) and

dHPC→NAc ($\chi^2(1, 444) = 23.248$, $P < 0.001$) place cells compared to non-place cells. The proportion of dHPC→NAc lick-excited place cells is also higher than the proportion of dHPC⁻ lick-excited place cells ($\chi^2(1, 1750) = 9.442$, $P = 0.0021$). **e**, **f** Conjunctive coding of velocity and licking. **e** Histogram of lick-excited (no fill) and lick-inhibited (fill) cells' velocity correlations (green: dHPC⁻; red: dHPC→NAc). **f** Proportions of lick-excited (left) and lick-inhibited (right) cells among speed-inhibited (no fill), non-speed-modulated (fill) and speed-excited (fill, black stroke) dHPC⁻ (green) and dHPC→NAc (red) cells. Proportions of lick-excited cells are overrepresented in speed-inhibited cells (dHPC⁻: $\chi^2(2, 4928) = 100.484$, $P < 0.001$; dHPC→NAc: $\chi^2(2, 444) = 27.608$, $P < 0.001$). Lick-excited neurons are further enriched in speed-inhibited dHPC→NAc neurons compared to dHPC⁻ neurons ($\chi^2(1, 830) = 6.564$, $P = 0.0104$). Proportions of lick-inhibited cells are overrepresented in speed-excited cells (dHPC⁻: $\chi^2(2, 4928) = 290.832$, $P < 0.001$; dHPC→NAc: $\chi^2(2, 444) = 18.825$, $P < 0.001$). All statistical tests performed are chi-squared tests. All data are presented as mean ± SEM. ns not significant, *$P < 0.05$, **$P < 0.01$, ***$P < 0.001$. Source data are provided as a Source Data file.

neurons allows NAc neurons to better identify the reward zone and guide task-appropriate appetitive behavior.

## Discussion

The hippocampus is one of the most studied brain regions in neuroscience[67]. Yet, the information content and acute behavioral relevance of its various output streams remain largely unknown. The NAc, with its proposed role as an integrator between limbic and motor systems[23] seems an ideal candidate to transform hippocampal mnemonic and contextual information into task-relevant behaviors[24], but evidence for such a role has remained sparse.

We find that the HPC→NAc projection is not only required for spatial memory tasks such as the water maze[29] or conditioned place

preference[22,31] but also for the acquisition and retrieval of learned reward locations in a head-fixed linear navigation task. Specifically, we found that optogenetic excitation of this pathway during learning enhances the animal's ability to memorize the reward zone location, as indicated by a higher rate of anticipatory licking already on the second day of training. Such facilitation of learning complements previous studies demonstrating the induction of spatial preference by optogenetic stimulation[30,31] and learning-related increases in HPC→NAc coupling[41]. Conversely, inhibition specifically decreased the amount of licking directly preceding the reward zone without affecting the overall success rate. This decrease indicates either a lack of spatial precision, as suggested by the flattened shape of licking across spatial positions, or a decrease in confidence to lick

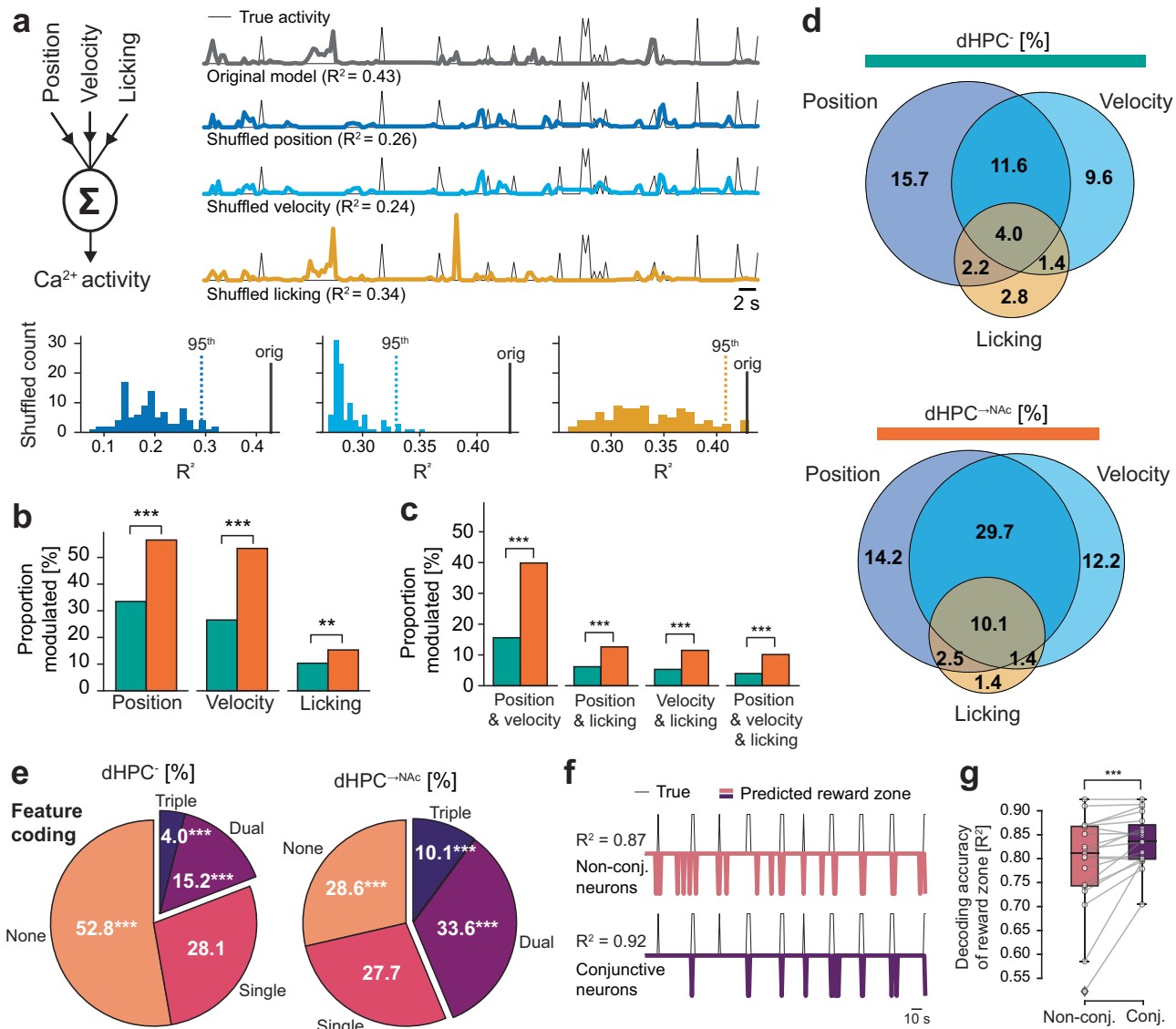

**Fig. 8 | A generalized linear model confirms enhanced conjunctive coding in NAc-projecting neurons. a** Schematic of the generalized linear model (GLM) (left). Example modeling approach for a triple conjunctive neuron (right). Upper traces show true calcium activity (downsampled and normalized) of test dataset (thin black line) as well as predictions of original model (thick gray) and one example for each shuffled feature. Bottom histograms show $R^2$ distributions for 100 shuffled models for each feature. Dotted line represents value of 95th percentile, thick black line represents original model's $R^2$. **b** Increased proportions of dHPC$^{\rightarrow NAc}$ neurons modulated by position ($\chi^2(1, 5372) = 93.634$, $P < 0.001$), velocity ($\chi^2(1, 5372) = 141.86$, $P < 0.001$), and licking ($\chi^2(1, 5372) = 10.050$, $P = 0.0015$). **c** Increased proportions of conjunctive coding in dHPC$^{\rightarrow NAc}$ neurons for position & velocity ($\chi^2(1, 5372) = 163.97$, $P < 0.001$), position & licking ($\chi^2(1, 5372) = 26.029$, $P < 0.001$), velocity & licking ($\chi^2(1, 5372) = 27.145$, $P < 0.001$), and position & velocity & licking ($\chi^2(1, 5372) = 34.993$, $P < 0.001$). **d** Venn diagrams showing overlap of neurons GLM-classified as modulated by position, velocity and licking in dHPC$^-$

(top) and dHPC$^{\rightarrow NAc}$ (bottom) neurons. **e** Proportions of $n$-feature coding neurons in dHPC$^-$ (left) and dHPC$^{\rightarrow NAc}$ (right) populations. Non-coding neurons are overrepresented in dHPC$^-$ neurons ($\chi^2(1, 5372) = 35.382$, $P < 0.001$); single-coding neurons are comparably distributed ($\chi^2(1, 5372) = 0.0057$, $P = 0.940$); dual-coding ($\chi^2(1, 5372) = 61.336$, $P < 0.001$) and triple-coding ($\chi^2(1, 5372) = 30.447$, $P < 0.001$) neurons are overrepresented in dHPC$^{\rightarrow NAc}$ neurons. **f, g** A linear classifier to decode the presence of the reward zone. **f** Example true presence of reward zone (thin line) and decoder predictions based on non-conjunctive neurons (pink; top) and conjunctive-coding neurons (dark purple; bottom). **g** Conjunctive-coding neurons allow a linear decoder to classify the presence of reward zone more accurately than non-conjunctive coding neurons (Wilcoxon's $W(18) = 14.0$, $P$(two-tailed) $<0.001$, $n = 19$ imaging sessions). The box represents quartiles and whiskers represent outlier-corrected minima and maxima of the distribution. Statistical tests used in **b**, **c**, **e** are chi-squared tests. **$P < 0.01$, ***$P < 0.001$. Source data are provided as a Source Data file.

near the learned reward location, as evidenced by the lick decrease observed. Given that overall (appetitive) lick rates were not affected, the animals' willingness to expend effort seems not to have been impacted by this inhibition. These data thus suggest that information from dHPC is required for the NAc to engage in appetitive behaviors guided by learned reward locations and stands in contrast to BLA afferents that are required for appetitive behaviors triggered by learned cues[35,68]. Taken together, these findings highlight a dual role

for the HPC→NAc pathway in both learning and retrieval of spatial memories.

To understand the information content of dHPC$^{\rightarrow NAc}$ neurons while animals performed this task, we used dual-color two-photon imaging of identified projection neurons. We identified a larger proportion of place cells among the dHPC$^{\rightarrow NAc}$ population compared to a simultaneously recorded general dHPC$^-$ population. These dHPC$^{\rightarrow NAc}$ place cells also encoded more spatial information and showed both

enhanced trial-to-trial reliability and in-place-field activity. Our data present direct evidence that the NAc receives spatial information from dHPC[24], and likely explains previously described spatial tuning of NAc neurons[28]. These data also call into question previous models of hippocampal memory processing that assumed largely homogeneous cell populations[69,70]. Indeed, hippocampal principal neurons are increasingly recognized as structurally and functionally diverse in terms of their morphology, electrophysiology, transcriptomic expression, and anatomical differences across all three axes[37,71]. Together with a recent study in dorsal subiculum[36], our study suggests that projection identity represents a further dimension that determines dorsal hippocampal coding properties. Such heterogeneity may provide the hippocampus with the intrinsic flexibility to meet the various demands of diverse environments[37].

Hippocampal projection-specific coding has been previously demonstrated in ventral CA1[40]. Indeed, functional differences along the hippocampus' septotemporal axis have long been established, with the dorsal part credited with mapping cognitive relationships in the world, and the ventral part with valence processing[72]. This view is also supported by a recent study on odor-outcome learning that found robust odor coding in dCA1, while vCA1 odor responses only emerged after acquiring positive valence[73]. In line with this, much of the past literature on spatial reward learning has focused on the prominent NAc afferents from vHPC[30,31,74–76]. Only recently have the anatomically weaker dHPC afferents to the NAc started to receive attention. Trouche et al.[22] clearly demonstrated the existence of such dCA1→NAc projections at an ultrastructural level and found these to be necessary for CPP. Interestingly, a recent study compared dHPC vs vHPC interactions with the NAc in a spatial reward learning task and found that only dHPC−NAC ensembles were tuned to task- and reward-related information[77]. These recent findings, along with our data, raise the possibility that the dHPC may be much more relevant for valence processing than had been previously assumed, at least as it pertains to its NAc projections.

This is further illustrated by our finding that spatial coding was not distributed uniformly across the environment: we found place cells overrepresenting the reward zone during high-success trials, particularly in the dHPC→NAc population. These findings are in line with previous projection-agnostic observations in CA1 and subiculum[33,54,55,78–80]. While previous studies suggested a role for a largely undefined dedicated population of dHPC cells in this overrepresentation[79] and differences have been observed along the radial dCA1 axis[33], evidence for projection-specific reward zone representations has remained sparse. Although synchronous activity has been observed between dHPC and NAc during spatial reward learning tasks[41,77,81], in line with a central role of NAc in reward learning[82], our findings present direct evidence of reward zone overrepresentation by dHPC→NAc neurons. Interestingly, a previous study[40] showed projection-specific ambiguity of reward zone activity depending on vCA1→NAc collaterals. Future studies should investigate potential functional differences in collaterals of dHPC projections[42].

The mechanistic basis of such spatial reward overrepresentations likely depends on the interaction of inputs from entorhinal cortex layer 3, hippocampal NMDA receptors, and dopaminergic inputs from VTA and LC[32,54,57,83,84]. Specifically, dopaminergic release from VTA terminals in dHPC has been suggested to stabilize spatial representations and enhance learning performance: optogenetic activation of dopaminergic VTA terminals in dHPC was shown to induce a shift in place fields and to enhance offline reactivation and memory recall[83,85], possibly through hippocampal D$_{1/5}$ receptors[86]. Two recent articles demonstrated that VTA terminals in dHPC show characteristic "ramp" activity as mice approach a learned reward zone[57,87]. Both loss of reward expectation and optogenetic silencing of these terminals resulted in a widespread degradation of spatial coding throughout dHPC place cells[57]. Given our findings of enhanced spatial and reward-related

coding in dHPC→NAc neurons, it is tempting to speculate these might be the result of enhanced DA receptor expression in these neurons. Interestingly, dopaminergic inputs to dHPC→NAc neurons may form part of a feedback loop by which NAc neurons could modulate VTA activity[88,89] that, in turn, may change DA release in HPC. Such a feedback loop (HPC−NAc−VTA−HPC) could play an important role in memory updating, as functionally similar pathways have been recently found to drive extinction and reconsolidation learning in *Drosophila*[90]. Future studies could investigate specific dopaminergic contributions using novel sensitive neurotransmitter indicators[91].

Successful behavioral performance in our spatial reward learning task depended on a decrease in velocity and increased appetitive lick activity as mice approached the hidden reward zone. In line with previous work[11,13,16,36], we found a considerable proportion of speed-modulated dHPC neurons. Interestingly, NAc-projecting neurons were more likely to be speed-inhibited, suggesting an elevated role in reward approach behaviors[16]. Similarly, we observed significant modulation of hippocampal neurons during appetitive and consummatory licking[16,59]. This modulation was dichotomous: consummatory licking resulted in widespread suppression of dHPC activity, while appetitive licking led to enhanced activity in dHPC→NAc neurons. Activity of dHPC neurons, and especially those projecting to NAc, thus seems to mirror the activity within NAc: Inhibition of NAc D1R medium spiny neurons (MSNs) has been observed during lick behavior[74,92,93] and was shown to be both necessary[93,94] and sufficient[93,95] for consummatory licking. We found that the origin of these signals may be localized upstream, similar to recent findings of an inhibitory permissive drive for licking in ventral HPC (vHPC) projections to the NAc[74,75] and that inhibiting these projections facilitates licking[74]. This suggests a similar involvement of dHPC- and vHPC-accumbens projections for consummatory licking.

Contrasting this consumption-related inhibition, we found appetitive lick-related dHPC→NAc excitation and optogenetic induction of lick-related behaviors. This dichotomy likely reflects previously described differences in the NAc's role for consummatory and appetitive behaviors[61]: vHPC→NAc activity was shown to increase during active food investigation[75] and around the time of lever pressing to obtain a reward[96], while optogenetic stimulation of this projection facilitated nose poke and lever pressing behaviors[31,34]. Given that optogenetic stimulation of D2R MSNs in the NAc shell was also found to produce repetitive jaw movements[97], it is likely that D1R and D2R MSNs, both of which are targeted by HPC axons[30,98,99], might play dichotomous roles in appetitive and consummatory behaviors[30,100]. Future studies should aim to resolve the behavioral roles of specific NAc cell types targeted by HPC neurons.

One general caveat for in vivo two-photon calcium imaging studies pertains to the potential of behaviorally induced motion artefacts along the imaging plane's $xy$ and $z$ axes to distort inferred calcium signals[101,102]. We have controlled for this possibility by comparing signals from our dynamic GCaMP channel to the simultaneously collected static mCherry signal. While we found a small positive inflection upon appetitive licking in our static channel, we note that this signal is relatively small and occurs in both lick-excited and lick-inhibited populations, making it unlikely to significantly affect our results. After controlling for these static signal fluctuations post hoc[103], the observed lick modulation holds true. Future studies on lick-related calcium dynamics, especially when sampling subcellular neural processes, should beware of such potential confounds.

In our analysis of dHPC→NAc coding properties, we uncovered more neurons individually coding for two or three aspects of space, velocity, and lick activity specifically in identified NAc-projecting neurons. Such conjunctive coding, or mixed selectivity, has been previously found in hippocampal and parahippocampal neurons to combine coding of space with direction, velocity, appetitive behaviors, behavioral tasks, future paths ("splitter cells"), context, or objects[11,15,16,58,78,104–106]. These conjunctions are hypothesized to combine "where" and "what"

information to disambiguate different experiences occurring in the same location, thus building a "scaffold" for episodic memories that allows unambiguous later retrieval[17,107,108]. Relatedly, theoretical and experimental studies demonstrated that such high-dimensional coding facilitates action selection by putative downstream linear decoders tasked with action selection and generating behavior[64,65,109], and was found to scale with task demands and performance[58,65,105,109,110]. Indeed, our data show that conjunctive coding improved reward zone detection by a linear decoder, similar to a recent study in retrosplenial cortex that demonstrated enhanced decoding near a reward zone by conjunctive coding neurons[105]. In line with previous findings of increased conjunctive coding in prefrontal cortex→NAc projection neurons[111] and elevated proportions of splitter cells in dSub→NAc neurons[36], we thus propose that dHPC routes strongly conjunctive task-relevant information to facilitate NAc action selection.

The NAc has been described as a key node transforming motivational information from the limbic system into motor behaviors[23], but the information flow of specific projection neurons has remained elusive. Our data demonstrate that dHPC routes enhanced spatial information to the NAc that is conjunctively enriched by further spatial and non-spatial task-relevant features. We show that this conjunctive code improves linear decoding to guide downstream action selection, and that dHPC can drive the execution of appetitive motor behaviors via the basal ganglia, as early studies proposed[112]. Thus, our findings identify a direct role for the hippocampus in the generation of motor behaviors and build an important bridge in our understanding of how sensory and mnemonic processes guide behavioral action.

## Methods

### Animals

Experiments were performed in adult male and female mice. C57Bl/6 ($n = 26$, out of which $n = 9$ (3 males, 6 females) for behavioral training and $n = 17$ (8 males, 9 females) for optogenetic experiments) and Thy1-GCaMP6s (GP4.3; $n = 6$ males for imaging) mice (The Jackson Laboratory, Bar Harbor, USA) were bred under specific pathogen-free conditions. Heterozygous mice were group-housed with 1–5 littermates per cage with 12 h reversed dark light cycle at 21 °C and 40–70% humidity and *ad libitum* food/water access until mice had recovered from surgery. Experiments were performed during the dark phase. All experiments were performed according to the Directive of the European Communities Parliament and Council on the protection of animals used for scientific purposes (2010/63/EU) and were approved by the animal care committee of North Rhine-Westphalia, Germany.

### Viral vectors

To co-express the static red fluorophore mCherry in Thy1-GCaMP6s animals, we injected retrograde (rg) serotype[113] adeno-associated virus (AAV) under control of the polyglycokinase (pgk) promoter (AAVrg-pgk-Cre, Catalog #24593-AAVrg, titer: $1 \times 10^{13}$ viral genomes (vg) ml$^{-1}$, Addgene, Watertown, USA) in the NAc to enter axons and induce Cre expression in NAc-projecting neurons. Then, we used double-floxed inverse open reading frame (DIO) Cre-dependent mCherry under the human synapsin promoter in dHPC (AAV5-hSyn-DIO-mCherry, titer: $1.1 \times 10^{13}$ vg/ml, Catalog # 50459-AAV5, Addgene, Watertown, USA). For optogenetic activation experiments, we used unfloxed ChR2[114] or control EYFP in excitatory neurons under the CaMKII promoter in the dHPC of C57Bl/6 mice (AAV2-CaMKII-hChR2(H134R)-EYFP-WPRE, titer: $4 \times 10^{12}$ transducing units (TU), UNC #AV4381E, Vector Core at the University of North Carolina, Chapel Hill, USA; rAAV2-CaMKII-EYFP, titer: $4.3 \times 10^{12}$ TU, UNC #AV6650, Vector Core at the University of North Carolina, Chapel Hill, USA). For optogenetic inhibition experiments, we used unfloxed ArchT[115] or control EGFP in retrograde (rg) serotype[113] adeno-associated virus (AAV) under control of the CaMKII promoter in the NAc of C57Bl/6 mice (AAVrg-CamKII-ArchT-GFP (PV2527), Catalog #99039-AAVrg, titer: $2.4 \times 10^{13}$ viral genomes (vg)

ml$^{-1}$, Addgene, Watertown, USA; AAVrg-CamKIIa-EGFP, Catalog #50469-AAVrg, titer: $2.5 \times 10^{13}$ viral genomes (vg) ml$^{-1}$, Addgene, Watertown, USA). pAAV-CamKII-ArchT-GFP (PV2527) was a gift from Edward Boyden (Addgene viral prep # 99039-AAVrg; http://n2t.net/addgene:99039; RRID:Addgene_99039). pAAV-CaMKIIa-EGFP was a gift from Bryan Roth (Addgene viral prep # 50469-AAVrg; http://n2t.net/addgene:50469; RRID:Addgene_50469).

### Stereotactic virus injections

For stereotactic injection of AAVs, 9–18 week old mice were anesthetized with an intraperitoneal (i.p.) injection of a mixture of ketamine (0.13 mg/g) and xylazine (0.01 mg/g). Mice were head-fixed using a head holder (MA-6N, Narishige, Tokyo, Japan), placed into a motorized stereotactic frame (LuigsNeumann, Ratingen, Germany) and warmed by a self-regulating heat pad (Fine Science Tools, Heidelberg, Germany). After skin incision (5 mm) and removal of the periosteum, placement of the injection was determined in relation to bregma. A 0.5 mm wide hole was drilled through the skull (Ideal micro drill, World Precision Instruments, Berlin, Germany). Stereotactic coordinates were taken from Franklin and Paxinos, 2008 (The Mouse Brain in Stereotaxic Coordinates, Third Edition, Academic Press). To induce retrograde labeling of NAc-projecting neurons in dHPC, $2 \times 500$ nl of AAVrg-pgk-Cre were injected into the ipsilateral (right) nucleus accumbens (−1.3 mm anterior-posterior, −1.0 mm lateral, 5.0 and 4.3 mm ventral, relative to Bregma) at 100 nl/min, using an Ultra-MicroPump, 34 G cannula and Hamilton syringe (World Precision Instruments, Berlin, Germany). To label projection neurons in red, 200 nl of AAV5-hSyn-DIO-mCherry were injected into the ipsilateral dorsal hippocampus (3.38 mm anterior-posterior, −2.5 mm lateral, 1.8 mm ventral, relative to Bregma, at a 10° angle). For optogenetic inhibition experiments, $2 \times 500$ nl of either AAVrg-CaMKII-ArchT-GFP (PV2527) or AAVrg-CaMKII-EGFP were injected each bilaterally into the medial parts of the NAc (−1.3 mm anterior-posterior, −1.0 mm lateral, 5.0 and 4.3 mm ventral, relative to Bregma). For optogenetic activation experiments, 200 nl of either AAV2-CaMKII-hChR2(H134R)-EYFP-WPRE or rAAV2-CaMKII-EYFP were injected each bilaterally into dorsal hippocampus (3.38 mm anterior-posterior, ± 2.5 mm lateral, 1.8 mm ventral, relative to Bregma, at a 10° angle). After surgery, buprenorphine (0.05 mg/kg) was administered thrice daily for 3 consecutive days. Implant surgery followed two weeks after AAV injection.

### Cranial window surgery

For awake hippocampal Ca$^{2+}$ imaging, a window was surgically implanted over the right dorsal hippocampus, one week after virus injections. The hippocampal window was assembled from a 1.7 mm long stainless-steel cannula (3 mm outer diameter) and a round cover slip (3 mm diameter). The coverslip was glued to the end of the cannula using UV curable adhesive (NOA81, Thorlabs, Dachau/Munich, Germany). Mice were anesthetized and prepared for surgery as described above. The skin over the parietal skull and the periosteum were removed and wound edges sealed with Vetbond tissue adhesive (3 M Animal Care Products, St Paul, USA). Skull surface was roughened by briefly applying gel etchant phosphoric acid (37.5%; Kerr Dental, Scafati, Italy), carefully washing the skull surface and applying two-component dental adhesive (OptiBond FL, Kerr Dental, Scafati, Italy). A circular piece of the skull (3 mm in diameter) centered over the injection was carefully cut out using a sharp drill (Ideal micro drill, World Precision Instruments, Berlin, Germany). The dura was removed with forceps and mild vacuum suction was used to slowly remove the cortex within the craniotomy. Blood and aspirated tissue were washed out using a constant flow of artificial cerebrospinal fluid (ACSF). Intracranial aspiration was continued until the external capsule was exposed. The external capsule remained intact. The hippocampal window was manually inserted. During continuous perfusion with ACSF, the hippocampal window was lowered until sealing onto the

external capsule. The hippocampal window was fixated, and the craniotomy sealed using UV curable dental cement (Gradia Direct Flo, GC Corporation, Tokyo, Japan). An angular metal bar (Luigs & Neumann, Ratingen, Germany) for head fixation was placed paramedian on the skull. After surgery, buprenorphine (0.05 mg/kg) was administered thrice daily for 3 days.

### Behavioral task

3-4 days after recovery from surgery, mice were provided with spinning wheels in their cages and were placed under a reverse light/dark cycle. One week after surgery, food restriction and two-week habituation schedules were initiated. Mice were food-restricted by providing about 80% of their measured daily food pellets every 24 h. In the course of this, mice lost about 10–20% of their original weight before the start of training. Habituation consisted of progressive exposure of mice to manual handling by the experimenter, obtaining milk rewards through a metal cannula, gentle manual head fixation, the treadmill apparatus, and, finally, head-fixed running on an unmarked treadmill belt with random rewards provided through a metal cannula lick spout after licking on it. The self-propelled treadmill (Luigs & Neumann) consisted of three rotating cylinders covered by a 7 cm wide and 360 cm long textile belt (Luigs & Neumann) including six differently textured zones: horizontal and vertical glue stripes, glue dots, Velcro dots, vertical tape stripes and upright nylon spikes. The reward zone for imaging experiments was 30 cm long and was placed between the end of the horizontal glue stripes and the beginning of the vertical tape stripes. The position of the mouse was recorded via an optical sensor (Luigs & Neumann) measuring the rotation of the treadmill cylinder underneath the mouse. Lick signals were measured by an analog piezo sensor; one full belt rotation was measured by an optical infrared sensor. All signals were collected at 10 kHz by an I/O board (USB-6212 BNC, National Instruments, Austin, USA) and recorded using custom-written Python software. After two weeks of habituation, 12–20 week old mice were placed daily for 10/15 minutes on a cued (see above) treadmill belt on which they needed to lick on the metal spout in this hidden reward zone to obtain a liquid reward in the form of condensed milk. Milk was released by a miniature peristaltic pump (RP-Q1-S-P45A-DC3V, Takasago Fluidic Systems, Nagoya, Japan) that was triggered by custom-written Python software via an I/O board (USB-6212 BNC, National Instruments, Austin, USA). After five days of training, calcium activity was recorded while mice performed the learned task.

### Infrared camera behavioral tracking

Headfixed mouse behavior was continuously monitored by simultaneously using two monochrome CCD cameras (Basler acA 780-75 gm) positioned at approximately 15 cm from the mouse. To capture face dynamics, we used a high-resolution zoom lens (50 mm FL, Thorlabs MVL50TM23); for body dynamics, we used a wide-angle lens (12 mm FL, Edmund Optics #33-303). Infrared illumination was provided via two 850 nm LED arrays (Thorlabs LIU850A), and cameras were outfitted with 850/40 nm bandpass filters (Thorlabs FB850-40). Both cameras' positions were aligned for each mouse before the start of recordings. Camera images were acquired at 25 or 75 Hz with 782 × 582 pixels using pylon Camera Software Suite (Basler), each frame triggered by TTL pulses from the recording software. Files were saved in compressed MP4 format before further processing.

### Two-photon calcium imaging

Two-photon imaging was performed in $n = 6$ male mice aged between 15 and 25 weeks using an upright LaVision BioTec (Bielefeld, Germany) TrimScope II resonant scanning microscope, equipped with a Ti:sapphire excitation laser (Chameleon Ultra II, Coherent, Santa Clara, USA)

operated at 920 nm for GCaMP6s fluorescence excitation, a second 1045 nm fixed-wavelength laser (Spectra Physics HighQ-2-IR, Newport Corp., Irvine, USA) for mCherry fluorescence excitation, and a 16x objective (N16XLWD-PF, Nikon, Düsseldorf, Germany). GCaMP6s fluorescence emission was isolated using a band-pass filter (525/40, Semrock, Rochester, USA) and detected using a GaAsP PMT (H7422-40, Hamamatsu, Herrsching am Ammersee, Germany). mCherry fluorescence emission was isolated using a band-pass filter (590/40, Semrock, Rochester, USA) and detected using a GaAsP PMT (H7422-40, Hamamatsu, Herrsching am Ammersee, Germany). Both lasers were aligned such that they excited the same focal plane. Imspector software (LaVision BioTec) was used for microscope control and image acquisition. Image series (2 channels, 1024 × 1024 pixels or 512 × 512 pixels, -350–850 µm square field of view) were acquired at 15.2 Hz or 30.5 Hz, respectively. Frame capture signals were recorded by an I/O board (USB-6212 BNC, National Instruments, Austin, USA) that allowed subsequent data synchronization.

### Optogenetic inhibition

For optogenetic inhibition experiments, directly following virus injections, the skin over the parietal skull and the periosteum were removed and wound edges sealed with Vetbond tissue adhesive (3 M Animal Care Products, St Paul, USA) in $n = 10$ mice (5 female, 5 male), aged between 14 and 15 weeks. Skull surface was roughened by briefly applying gel etchant phosphoric acid (37.5%; Kerr Dental, Scafati, Italy), carefully washing the skull surface and applying two-component dental adhesive (OptiBond FL, Kerr Dental, Scafati, Italy). Two small 0.5 mm wide holes were drilled through the skull (Ideal micro drill, World Precision Instruments, Berlin, Germany) and two 400 µm diameter mono fiberoptic cannulas (MFC_400/470-0.37_3mm_ZF1.25_FL, Doric Lenses, Quebec, Canada) were implanted bilaterally on top of the subicular-CA1 border (coordinates: −3.35 mm anterior-posterior, ±2.8 mm lateral, −1.75 mm ventral from brain surface, relative to bregma, at a 10 degrees angle). The craniotomy was then sealed using UV curable dental cement (Gradia Direct Flo, GC Corporation, Tokyo, Japan). An angular metal bar (Luigs & Neumann, Ratingen, Germany) for head fixation was placed paramedian along the midline and between the optic cannulas on the skull. After surgery, buprenorphine (0.05 mg/kg) was administered thrice daily for 3 days. In addition, mice received the analgesic metamizol (2 mg/ml) through their drinking water, starting 12 h before surgery until 72 h after surgery.

After two weeks of recovery, food restriction and two-week habituation schedules were started. After successful habituation, head-fixed mice aged 18–20 weeks received light stimulation via a fiber-coupled 561 nm diode laser (MGL-FN-561-100mW, PhotonTec, Berlin, Germany) through a splitter branching fiberoptic patchcord (SBP(2)_400/440/LWMJ/900-0.37_2m_FCM-2xZF1.25; Doric Lenses, Quebec, Canada), at 15–20 mW light intensity, measured at the fiber output. Custom-written Python scripts were used to activate the laser once animals reached 90 cm ahead of the start of the respective reward zone; light stimulation was provided up to 10 seconds per lap or until mice left the reward zone, whichever occurred earlier.

### Optogenetic activation

For optogenetic activation experiments, directly following virus injections, the skin over the parietal skull and the periosteum were removed and wound edges sealed with Vetbond tissue adhesive (3 M Animal Care Products, St Paul, USA) in $n = 7$ mice (4 female, 3 male), aged between 10 and 11 weeks. Skull surface was roughened by briefly applying gel etchant phosphoric acid (37.5 %; Kerr Dental, Scafati, Italy), carefully washing the skull surface and applying two-component dental adhesive (OptiBond FL, Kerr Dental, Scafati, Italy). Two small 0.5 mm wide holes were drilled through the skull (Ideal micro drill, World Precision Instruments, Berlin, Germany) and a two-ferrules fiber-optic cannula (TFC_200/230-0.37_5.5mm_TS2.0_FLT, Doric

Lenses, Quebec, Canada) was implanted bilaterally on top of NAc (coordinates: +1.3 mm anterior-posterior, ±1 mm lateral, −4.9 mm ventral from brain surface, relative to bregma). The craniotomy was then sealed using UV curable dental cement (Gradia Direct Flo, GC Corporation, Tokyo, Japan). An angular metal bar (Luigs & Neumann, Ratingen, Germany) for head fixation was placed paramedian on the skull. After surgery, buprenorphine (0.05 mg/kg) was administered thrice daily for 3 days.

After one week of recovery, food restriction and two-week habituation schedules were started. After successful habituation, head-fixed mice aged 13 to 15 weeks received light stimulation via a fiber-coupled 473 nm diode laser (LuxX 473-80, Omicron-Laserage) through a branching fiberoptic patchcord (BFP(2)_200/220/900-0.37_2m_FCM*−2xZF1.25; Doric Lenses, Quebec, Canada), at 5 mW light intensity, measured at the fiber output. Custom-written Python scripts were used to send a trigger signal to an analog isolated pulse generator (Model 2100, A-M Systems, Sequim, USA) once animals entered a hidden optogenetic stimulation zone. The pulse generator produced 5 ms long pulses at 20 Hz for a maximum duration of 10 seconds or until the mouse passed beyond the stimulation zone.

## Behavioral analysis

Analog signals for position, licking, reward pump, full rotation of belt, and digital signals from infrared camera triggers, two-photon scanning triggers, and optogenetic stimulation triggers were collected at 10 kHz and saved as TDMS files. Data processing and analysis were performed in Python. Position signals in cm were reconstructed using the optical sensor detecting a full rotation of the treadmill belt. Velocity was calculated using a Kalman filter applied to the position signal. Behavioral data were then downsampled to either match the sampling rate of camera tracking (25/75 Hz) for training data or to match the sampling rate of two-photon imaging (15/30 Hz), by using each time window's arithmetic mean (velocity), median (position, lap number, camera/scanner trigger), maximum (reward pump, optogenetic trigger), or standard deviation (licking). Discrete lick events were detected using Scipy's *find_peaks* function with a minimum temporal distance of 0.33 s and a dynamic minimum height threshold that was individually determined by inspecting the synchronized infrared camera video. Lick bouts were classified as lick events that were <2 s apart from one another. Appetitive lick onsets were defined as lick bout onsets that were preceded by at least 3 seconds absence of lick events. Consummatory licking onset was defined as the first lick event between 0.5 s and 5 s after reward pump trigger. Rewarded laps were defined as laps in which the reward pump was triggered by the animal's licking. Relative licking refers to the cumulative analog lick signal per position bin or reward zone. Reward anticipation zone is defined as 30 cm before the reward zone up to its start. Successful laps were defined as laps in which mice showed at least 50% of relative licking in reward and anticipation zones and received a reward. High-success trials were defined as trials consisting of at least 50% of successful laps. "Near reward zone start" refers to belt locations from 20 cm before reward zone up to its start.

## Calcium signal processing

Two-photon imaging data was processed using custom-written software in Python, largely based on CalmAn (v1.6.2[46]). Green and red channel 16-bit TIFF stack files (512 × 512 or 1024 × 1024 pixels times 9000 or 18,000 frames) were first resampled to 1 px/μm before motion-correcting the red static channel using NormCorre piecewise rigid (parameters: *max_shifts* = 40, *num_frames_split* = 2000, *overlaps* = 46, *splits_els* = 4, *strides* = 255). The motion-corrected red channel image was then averaged over *t* and used for later identification of mCherry-positive components. Motion correction vectors were then applied on the dynamic green channel before using constrained non-negative matrix factorization (CNMF) for cell segmentation. Resulting traces were detrended and

deconvolved before filtering spatiotemporal components using quality criteria followed by identity- and behavior-blind manual curation based on visual inspection of spatial and temporal footprints and the quality of deconvolution. To identify mCherry-positive components, resulting spatial footprints were overlaid over the red channel average and a dynamic threshold applied to visually match the optimal signal discovery. To account for potential motion artefacts captured by our static mCherry channel, we used Two-channel Motion Artifact Correction (TMAC)[103] using standard parameters, based on detrended raw fluorescence data. Residual motion was estimated using optical flow and template matching algorithms: Dense optical flow for *xyt* was estimated using the Farneback algorithm, implemented in Python OpenCV (*cv2.FarnebackOpticalFlow*). Optical flow magnitude was z-scored and averaged across *t* for each recording. Template matching was performed using Python OpenCV (*cv2.matchTemplate*) with each recording's average image from frames 500:1500 taken as reference. Red static signals were extracted using CNMF's spatial components from the dynamic channel and applying these as a mask on motion-corrected red channel TIFF stacks, using CalmAn's *extract_traces_from_masks()* method. For comparisons with GCaMP deconvolved signals, red channel traces were first detrended using *caiman.source_extraction.cnmf.utilities. fast_prct_filt* and then deconvolved using *caiman.source_extraction.cnmf. deconvolution.constrained_foopsi*.

## Place field analysis

Continuous belt positions were binned into 45 bins of 8 cm length. For calcium signals, deconvolved events were used to avoid differential effects of GCaMP6s calcium signal tails at different animal velocities across space. Only time points with a velocity >2 cm/s were considered for spatial tuning analysis. Spatial information (*SI*) was calculated as follows[49]:

$$SI = \sum_i o_i a_i \log_2(a_i/\bar{a}) \tag{1}$$

where *i* denotes the *i*-th spatial bin, $o_i$ is the animal's occupancy at spatial bin *i*, $a_i$ is the mean of deconvolved events at spatial bin *i*, and $\bar{a}$ is the overall mean calcium activity. Place cells were defined as cells whose *SI* was higher than the 95th percentile of 1000x randomly position-shuffled *SI* values[79]. Each shuffle consisted of randomly circularly shifting the activity time course by at least 500 frames, segmenting the activity in *n* blocks that were randomly permuted. *n* was automatically chosen for each session to match approximately twice the number of laps run to account for running differences between animals. The resulting shuffled trace was then used to calculate *SI* as above. Spatial information rate in bits per second was calculated by multiplying *SI* with the average firing rate during times of movement (>2 cm/s). Firing rate was defined as the number of deconvolved calcium events >0 per second. Sparsity was calculated[50], using the same denotation as above:

$$\text{Sparsity} = \frac{\left(\sum o_i a_i\right)^2}{\left(\sum o_i a_i^2\right)} \tag{2}$$

Place field reliability was calculated as the fraction of laps in which the maximum spatially binned deconvolved calcium activity occurred within the cell's place field. Δin-out place field activity was defined as the average deconvolved calcium event activity within a cell's place field subtracted by that activity outside the place field. Place fields were obtained by first averaging deconvolved calcium event activity at velocities >2 cm/s per 45 spatial bins, replicating the resulting trace by a factor of two to account for circularity of the belt, and applying a Savitzky-Golay filter (using Scipy's *savgol* function with a window of 5 frames at the second order polynomial) to account for skewed place field activity. The resulting filtered spatial calcium activity was then

searched for peaks using Scipy's *find_peaks* function (width: 1 frame; prominence: 1.5 standard deviations; relative height: 0.8), with the most prominent peak used as primary place field and left/right interpolated intersection points as place field beginning/end. Place fields that would stretch into the replicated positions were translated back to the original 45 spatial bins. Place field boundary ratio was calculated as segmenting position bins into texture transition or middle zones, depending on their proximity to a texture transition, and calculating the ratio of place field beginnings/ends in a transition zone over those in a middle zone. To compare this ratio with a shuffled distribution, all place fields underwent 1000 iterations of randomly circularly shifting both beginnings and ends simultaneously and saving the resulting ratio for both cell populations after each iteration.

Place field centers were defined as the center of mass (COM) of significantly spatially modulated neurons (see above). For this, calcium activity at velocities >2 cm/s per 45 spatial bins was transformed to polar coordinates with $q$ as position bin and $r$ as average deconvolved calcium event activity at that position[79]. The two-dimensional centroid was calculated, and the resulting angle transformed back to belt position to yield the place field center. "Place fields near reward zone" refers to place cells with COMs in either the anticipation zone (starting 30 cm before reward zone) or reward zone.

Overdispersion was calculated as previously reported[116]. In brief, for each identified place cell's place field, the expected calcium activity, *exp*, was calculated by averaging all non-zero deconvolved calcium events within its place field. The observed calcium activity, *obs*, was the average non-zero deconvolved activity within the place field per pass. Only passes with detected deconvolved calcium activity were used to calculate overdispersion. For each place cell and place field pass, we calculated the normalized standard deviation of *obs* as:

$$z = \frac{(\text{obs} - \text{exp})}{\sqrt{\text{exp}}} \tag{3}$$

$z$ thus measures the deviation of observed calcium activity from expected in standard deviation units. Overdispersion is the variance of the $z$ distribution for a set of passes.

### Decoding of reward and anticipation zones
For decoding of reward and anticipation zones, all non-zero deconvolved calcium event activity was normalized into quantiles. Position and calcium activity were only considered from time points with velocity >2 cm/s. Data were downsampled to 1 Hz by accumulating calcium activity and averaging position data. Positions in the 30 cm reward zone or the preceding 30 cm anticipation zone were binarized (1 inside zone, 0 outside zone). Models were trained using calcium data of either (non-)projecting or (non-)conjunctive neurons and 100 randomly sample size-matched neurons of the respectively larger population, ultimately using the average $R^2$ of the random sampling procedure. Neurons with fewer than 10 time bins of non-zero calcium activity in train/test datasets were excluded from training/testing of the model. A linear support vector machine (SVM) classifier (Sklearn's *linear_model.SGDClassifier* function; loss: hinge, penalty: L2) was cross-validated by training on even laps and testing on odd laps, and vice versa, using the average of the resulting $R^2$. For models comparing (non-)projection neurons only imaging sessions with >10 projection neurons were included.

### Velocity and lick coding
Velocity modulation was assessed by linear regression of each neuron's average deconvolved calcium event activity against 1 cm/s velocity bins from 2 up to 30 cm/s using Scipy's *stats.linregress* function[60]. Cells were considered to be significantly velocity modulated at $P_{\text{adjusted}} < 0.05$ after correcting for false discovery rate by using Benjamini/Hochberg correction. Significantly velocity modulated cells

with positive slopes were termed "speed-excited" and those with negative slopes "speed-inhibited".

To assess population-level appetitive/consummatory lick modulation, average deconvolved calcium activity before and after each lick event (see above for definition) were compared within the respectively described time windows. For single-cell level appetitive lick modulation analysis, each neuron's average pre/post deconvolved calcium activity was compared across events using Wilcoxon's non-parametric paired test. Cells were classified as "licking-excited" if they were significantly modulated and showed a positive difference, or "licking-inhibited" if they showed a significant negative difference.

### Generalized linear model
To determine cellular coding of space, velocity, and licking, we used a generalized linear model to predict each neuron's calcium activity. We normalized calcium activity by dividing each neuron's deconvolved calcium events by their signal-to-noise ratio and the resulting standard deviation. Normalized calcium activity was then smoothed using a Gaussian window of 0.5 seconds (SD = 2) and temporally downsampled by averaging to 3 Hz. Finally, downsampled non-zero calcium data were normalized into quintiles.

The feature matrix consisted of normalized position, velocity, and lick data. Position data were binned into 45 equally sized spatial bins (8 cm each) and median-averaged to 3 Hz, used as factors. Velocity data were normalized by dividing data by its mean and averaging to 3 Hz. Lick bouts were calculated as described above. Appetitive lick bouts were defined as all lick bouts occurring up to 5 seconds before and at least 5 seconds after reward dispensation. Appetitive lick bouts were downsampled by median-averaging to 3 Hz. Both downsampled velocity and downsampled licking (appetitive lick bouts) were replicated with time shifts spanning 3 seconds before to 3 s after original timing at 3 Hz to account for anticipatory or delayed calcium activity.

Generalized linear Poisson models were trained, validated, and tested using Python's *H2O* library (H2OGeneralizedLinearEstimator; lambda: 0) with a train/validation/test ratio of 0.8/0.1/0.1. The resulting $R^2$ score based on the test dataset was saved and compared to that of randomly shuffled feature models. For each predictor group (position, velocity, licking), 100 shuffled models were generated by circularly shifting the respective group's data randomly between 200 and 700 time points, saving each model's $R^2$ score. Significant predictors for each neuron's calcium activity were defined as those whose random shuffling procedure resulted in reductions in $R^2$ for more than 95 % of cases. This means that at least 96 of the 100 models with one randomly shuffled predictor group had to have decreased $R^2$ values compared to the full model, in order for this predictor to be considered significant. Conjunctive-coding neurons were defined as those with at least two significant predictor groups.

### Infrared camera recordings analysis
Markerless pose estimation (Deep Graph Pose[117] and DeepLabCut[118]) was used to detect facial and body movements for both types of videos. For this, a deep neural network was trained to automatically discriminate 15 markers for videos of the body (paw, tail, and head segments) and 13 markers for videos of the face (6 for pupil, eye, nose, mouth, etc.). The network was trained on a large variety of lighting conditions and angles until it reached satisfactory performance. Mouth facial regions of interest (ROIs) were automatically segmented using video-averaged marker points of nose tip, the eye's tear duct and the mouth as stable landmarks. Mouth motion energy was calculated as the ROI's average pixel-by-pixel intensity differences from $t_{-1}$ to $t_0$, and z-scoring the resulting activity.

### Statistical analysis
Statistical analysis was performed using Python libraries *pingouin*[119], *scipy.stats* and *DABEST*[120], as well as R library *rstatix*. Statistical tests are

indicated in the figure legends and text. To evaluate statistical significance, data from Fig. 1d, S1g–j were subjected to a one-way repeated measures ANOVA. Data in Figs. 1i, j, 2f, h, 4f, 6c and S2a, f, S7c, f, S8l, n were subjected to two-way mixed ANOVAs followed by Bonferroni post-hoc tests. Data in Figs. 3f, 5d, h, 6e–f, 7b, d, f, 8b, c, e and S6c, d were subjected to chi-square tests. Data in Fig. 3g–k were subjected to Mann-Whitney U tests. Data in Figs. S2g, S3g, h, S8d, e, S11a–c were subjected to Welch's $t$-tests. Data in Fig. 2c–f were subjected to independent samples $t$-tests. Data in Fig. S9e were subjected to paired $t$-tests. Data in Fig. 4c, d, S5j were subjected to bootstrapping analysis (see above) and chi-square tests. Data in Figs. 4g and 8g were subjected to Wilcoxon's test for paired nonparametric data. Data in Figs. 3f–k, S5a–f, S5k–m, S5p, S5r, S6a, b, S8a–c, S8g, S10a–f, S11d–k were subjected to estimation statistics: data were plotted against a mean (median or paired mean, as indicated) difference between two conditions (as indicated) and this difference was compared against zero using 5000 bootstrapped resamples. In these estimation graphics, each black dot indicates a mean (median or mean paired, as indicated) difference and the associated black ticks depict error bars representing 95% confidence intervals; the shaded area represents the bootstrapped sampling-error distribution. Data in Fig. S5g were compared using the Levene test for equal variances. Data in Figs. S2b and S5i were compared using the Kolmogorov–Smirnov test. Data in Figs. S5q and S8h–j show the results of Pearson's correlation. Data in Supplementary Tables 1-3 show the results of 3-way mixed ANOVAs. Supplementary Tables 5-7 show the results of 2-way ANOVAs. Supplementary Table 4 shows the results of a linear mixed model. For all analyses data are presented as mean ± SEM, unless stated otherwise, and the threshold for significance was at $P < 0.05$.

### Reporting summary
Further information on research design is available in the Nature Portfolio Reporting Summary linked to this article.

## Data availability
The raw and processed data generated in this study have been deposited on Zenodo under https://doi.org/10.5281/zenodo.10698565. Source data are provided in this paper.

## Code availability
The code used to create all main manuscript figures based on the data provided above can be found on https://github.com/obarnstedt/dHPC-NAc.

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

## Acknowledgements

We would like to thank Janelle Pakan, Simon Musall, Hugo Spiers, Magdalena Sauvage, Marlene Bartos, and Thomas Mrsic-Flogel as well as the rest of the Cellular Neuroscience Department for their insightful comments and discussions while preparing this manuscript. In addition, the authors would like to thank Falko Fuhrmann, Pavol Bauer, and Rüdiger Geis for their help with the treadmill setup, cranial surgery, and two-photon imaging. The resources for all experiments were provided by the German Center for Neurodegenerative Diseases (DZNE) in Bonn and the Leibniz Institute for Neurobiology (LIN) in Magdeburg. The authors received funding from the German Research Foundation (DFG) SFB grants 1089 and 1436 and IRTG 2413 (to S.R.) and ERC Consolidator grant Sub-D-Code (to S.R.).

## Author contributions

O.B. and S.R. conceived and designed experiments. Imaging experiments were performed and analyzed by O.B. Optogenetic experiments were performed by O.B. and P.M., and analyzed by O.B. Manuscript was written by O.B. with help from S.R. and P.M.

## Funding

## Competing interests

The authors declare no competing financial interests.
