## [Peer Review File · Nature Communications]

REVIEWER COMMENTS

Reviewer #1 (Remarks to the Author):

This manuscript uses 2P imaging in Thy1-GCaMP6s mice combined with retrograde labeling to probe general neuronal activity and activity specifically in a subset of dorsal hippocampus of neurons projecting specifically to the nucleus accumbens while head-fixed mice run on a rotating track and encounter milk reward. The majority of the manuscripts reports results from thorough and rigorous analyses of this data from neurons recorded in 18 mice (?), revealing enrichment of place cells amongst the dHPC-NAc neurons (relative to the general dHPC population) as well as increased decoding accuracy of a reward zone.

Among other findings, they identify an interesting dissociation in the neural correlates of consummatory licking and appetitive licking and an enrichment of appetitive-lick excited neurons amongst dHPC-NAc neurons that hints at a possible specialized roles for this projection pathway in behaviorally relevant conveying information in the context of reward rather than reward per se. This is potentially very interesting and could be further developed, particularly with causal manipulations with nuanced behavioral readouts.

A second experiment examined the effect of optogenetic stimulation of dHPC-NAc terminals within the same behavioural design. ChR2- (n=4), but not EYFP-expressing (n=3), mice showed reduced velocity and increased licking at the onset of optogenetic stimulation. This is interpreted as further evidence that dHPC-NAc conveys task-relevant information about appetitive behaviors required to obtain rewards.

This is an interesting and thorough investigation of how neural activity in dHPC, and dHPC-NAc in particular, is modulated as animals run in a rewarding environment. The experimental approach is simple yet powerful and the analyses of neural data provide new insight into task-relevant information that is recoverable from this neural signal. However, data directly demonstrating that this signal is 'guiding goal-directed appetitive behavior' are somewhat limited.

Major comments

- While the analysis of neural data is thorough and compelling, the findings from the optogenetic experiments remain preliminary. More work is needed to establish the claim that dHPC-NAc is a driving force in generating task-relevant behaviour. For example, it would have been interesting to follow up more directly on the neural dissociation between consummatory and appetitive licking. Is the 'mouth motion' induced in other contexts or is it specific to a context where reward has been available? An

inhibition experiment could also help to bolster the present results and clarify what the neural signal is conveying. Further, given the central claim of the paper of a specialized function for the dHPC-NAC pathway, as compared to dHPC in general, it would be interesting to see some comparison between a pathway-specific vs non-specific manipulation.

- Methods indicate that experiments were conducted in male and female mice but no further information is provided. The numbers of each sex must be indicated for each experiment. Were any analyses performed to determine if there were sex differences? At a minimum, indicating the sex of all individual data points (e.g. by using 2 different colours) would increase transparency and facilitate comparison of any potential sex differences by interested readers.

Minor comments

- dHPC-NAC neurons are 'more likely to be speed-inhibited and less likely to be speed-excited' (line 246-247. Is it surprising then that optogenetic stimulation of dHPC-NAC suppresses velocity?

- The optogenetic experiment includes two groups (ChR2 and EYFP) with within group comparisons (pre, post stimulation). The data are analyzed as two separate paired t-tests. The appropriate analysis is a two-way repeated measures ANOVA.

- Descriptions of number of mice included in each experiment are confusing. The methods refer to 16 C57/Bl6 mice and n= 6 Thy1-GCaMP6s mice (or are these the original mice used to generate the experimental mice?). Figure 1 and FigS1 refer to n= 18/ 2 cohorts of n=9 mice. The presented optogenetic experiments present data from 7 mice although reference is made to n=6/group in the methods for this experiment. The discrepancy between the starting and final n for the optogenetic experiment should be clearly explained. The sample size for this experiment is very low (n=3/4). How was the appropriate sample size determined?

- What age were mice at the time of surgery/experimentation?

- What is the rationale for choosing 20Hz/5 ms stimulation? How does this related to known physiological firing of dHPC-NAC neurons or neural activity recorded in the present manuscript?

- What kind of 'milk reward' was used?

Reviewer #2 (Remarks to the Author):

This is an exciting paper examining the role for dCA1 neurons that project to the nucleus accumbens in head-fixed, behaving mice. The authors use an established reward-memory task to compare the encoding properties and function of dCA1-Nac-projecting versus non-Nac-projecting neurons. They find Nac-projecting cells are more likely to have place fields and they tend to have enhanced spatial encoding properties. They also tend to encode more than just space, and conjunctively encode deceleration and appetitive licking which helps decode locations where rewards are expected. Further, the authors show that opto-activation of CA1 terminals in Nac drives deceleration and licking behavior. Altogether these results are an important addition to the literature. I have some suggestions to improve the paper:

Are dCA1-Nac neurons distributed evenly throughout dCA1? Some sort of quantification of their location should be included.

The FOVs of CA1 in Fig. S3 look a little odd compared to typical dCA1 FOVs under 2P imaging, especially mouse 210. Do the authors have a reason why the pyramidal neurons in these FOVs are not packed tightly together as is typical? Why does the expression of GCaMPs look so sparse? Each FOV from these mice look like a recording from a different location within HPC. Should data from these mice be pooled given the difference in these FOVs? Mouse 210 looks especially different from what is expected. Relatedly, the spatial activity of the cells from mouse 210, and somewhat from mouse 207, looks sparse, with many laps showing no activity in the place fields. It is well known that place field firing varies extensively from lap-to-lap (over dispersion), but these cells seem to go beyond what is expected based on other papers. The authors should quantify over-dispersion or some related measure such as lap-to-lap reliability to see if their place cells are behaving in an expected way compared to what has been reported. If they are not, then the authors should explain why. I know they compare similar measures across Nac-projecting and non-projecting neurons, but it would be useful to put their results in the context of the many papers that report these features of place cells.

Fig. 3 compares the distribution of PFs around boundaries and reward zones in Nac-projecting and non-projecting place cells. Significance is reported in most comparisons using the Wilcoxon t-test, but this test is highly sensitive to the number of samples used, and p-values are driven very low when n is high, over-estimating differences. Given the high n in these comparisons, and what looks like small differences between groups, I recommend the authors conduct estimation approaches on these datasets¹. This emphasizes reporting effect sizes with expressions of uncertainty (interval estimates) to make better inferences: it pushes back against over-confident claims from inadequate samples, improves comparisons of results across contexts, and normalizes the publication of negligible effects¹. I think this method of comparison should be used in all cases where n is high and the authors are comparing 2 groups (so not just in Fig. 3). This is especially important in Fig. 3c,d, and g. Relatedly, in Fig. 3f, the authors state that the proportion of place fields in reward and vicinity zone is significantly higher in high-success trials (colored bars) compared to low-success trials (gray bars) in NAc-projecting neurons (red) but not in dHPC- neurons (green). This conclusion is based on the green bar having a Pval of 0.051 compared to the gray bar. If this pval was only marginally lower, say 0.049, the authors would have a

different conclusion. Again, this is why reporting effect sizes is more informative than drawing conclusions based on arbitrary Pval thresholds.

1 Calin-Jageman, R. J. & Cumming, G. Estimation for Better Inference in Neuroscience. *eNeuro* 6, doi:10.1523/ENEURO.0205-19.2019 (2019).

Lines 157-158 – “...previous work demonstrated the dependence of such an overrepresentation on individual task success (Dupret et al., 2010; Danielson et al., 2016).” Relatedly, a recent paper showed that reward over-representation in a similar experimental setup was dependent on the animal’s reward expectation (Krishnan et al, 2022) and another showed that the number of place fields in an environment is dependent on behavioral engagement (Pettit et al., 2022). The authors may want to add a comment about these papers here or elsewhere in the manuscript as they both seem highly relevant to the results in this paper:

2 Krishnan, S., Heer, C., Cherian, C. & Sheffield, M. E. J. Reward expectation extinction restructures and degrades CA1 spatial maps through loss of a dopaminergic reward proximity signal. *Nat Commun* 13, 6662, doi:10.1038/s41467-022-34465-5 (2022).

3 Pettit, N. L., Yuan, X. C. & Harvey, C. D. Hippocampal place codes are gated by behavioral engagement. *Nat Neurosci* 25, 561-566, doi:10.1038/s41593-022-01050-4 (2022).

The optogenetic activation of dCA1 terminals in Nac is very interesting. Did the authors think about or attempt to perform an inhibitory experiment whereby these terminals are inhibited in the learned reward zone? This would strengthen the argument that the CA1-Nac pathway is involved in appetitive behaviors associated with reward.

Lines 314-315 in reference to dCA1/vCA1 comparisons, the authors could add this very recent and relevant paper from the Kheirbek lab:

4 Biane, J. S. et al. Neural dynamics underlying associative learning in the dorsal and ventral hippocampus. *Nat Neurosci*, doi:10.1038/s41593-023-01296-6 (2023).

Line 323, the authors mention dopaminergic VTA inputs to hippocampus as they may relate to their Nac projecting neurons. The Krishnan et. al. 2022 paper mentioned above recorded from these DA inputs in a similar behavioral setup. The authors could briefly discuss their results in the context of the reported

VTA-CA1 signals. It could help give mechanistic insight into how DA might influence encoding by Nac CA1-projecting neurons associated with reward.

Reviewer #3 (Remarks to the Author):

Barnstedt et al. provide a very interesting and much needed characterization of hippocampal output to the nucleus accumbens (NAc), a key structure in the transformation of spatial state information into value-guided action selection. Using two-photon imaging of dorsal hippocampal (dHPC) subiculum and the CA1/subiculum border, the authors monitor the activity of the dHPC population in addition to the activity of the NAc-projecting subpopulation (dHPC->NAc), labeled by injection of a retrograde Cre virus in NAc and Cre-dependent mCherry in dHPC. They find that spatial coding is enhanced in the dHPC->NAc population compared to non-mCherry-labeled cells (dHPC-), as supported by a higher fraction of place cells, higher spatial information, higher stability, etc. in the dHPC->NAc subpopulation. They further find that the hidden reward zone in their task is overrepresented more prominently in the dHPC->NAc cells, particularly on successful behavioral trials. The dHPC->NAc subpopulation prefers to fire at slower movement speeds, and the authors are able to more accurately decode the animal's occupancy of the anticipatory zone before the reward from the dHPC->NAc population compared to the dHPC- population, suggesting the dHPC->NAc subpopulation is biased to encode upcoming reward locations. Together, these results are strong and make an important contribution to our understanding of how the hippocampus might bias its outputs to route specific information to downstream circuits. However, I have significant concerns about the latter part of the manuscript that claims lick coding in the same hippocampal neurons. I've outlined these concerns in detail below, in addition to some more minor comments.

Major:

1. Given the very low amplitude of the lick-onset-triggered signal that you observe (e.g. in Fig. 5f-g, z-score of 0.2-0.4), I have a strong concern that this lick "coding" is driven by z-motion artefact from the mouse slowing down and licking. In our own data using 2P imaging (unpublished, unfortunately), we have observed very similar signal in dorsal CA1 neurons expressing static GFP. I believe that the mouse's licking movement generates a slight z-offset in the brain relative to the imaging plane. This z-motion creates apparent increases in fluorescence in a subset of ROIs, and a decrease in fluorescence in another subset of ROIs that is time-locked to the onset of detected licking – exactly what you observe. This artefact can lead to a ~5% or greater dF/F change in GFP signal, or a z-score of ~0.2-0.4. For the positive-going ROIs, we often see a small decrease in signal before the lick, an increase at licking onset, followed by another small decrease. For the negative-going ROIs, we see an increase preceding the lick, followed by a decrease after lick onset. Again, this looks exactly like what you see in Fig. 5. The fact that you see

this to a greater degree in dHPC->NAc neurons is curious, but I hypothesize that it could be explained by 2 variables: (1) the dHPC->NAc neurons are anatomically located in higher density in what I assume is the antero-medial part of your field of view (FOV) shown in Fig. 5 (see point #7 requesting anatomical axes labels here). I assume this is also where the FOV transitions to the subiculum, where the brain may slope downward relative to the imaging glass – is this correct? This depth discrepancy could cause the relative fluorescence change to be higher as a result of licking-induced brain motion. (2) The dHPC->NAc neurons are more active at slower speeds according to your results, which is also when the animal is licking. This means the artefactual dF/F from licking would be more likely to get picked up if the cell is already firing in a place field occurring at low speeds. Moreover, it's important to note that I am unaware of any other paper demonstrating reliable licking signals in subiculum/CA1, a point the authors do not address. I suggest the following controls to account for this possible artefact in your data:

1a. First, perform the same lick-bout-triggered analysis using your static red mCherry signal. If you see a comparable increase/decrease in signal in some of the ROIs at the onset of licking, you definitely have a problem. However, I am not sure if this artefact is the same with red fluorophores as with green, as the red fluorophore signal transmits further through tissue, so it might be less affected by z-offsets.

1b. Therefore, you should perform control experiments with animals expressing static GFP in pyramidal cells (n=2 mice would be sufficient). If you see lick-induced dF/F in these animals, imaged at the same anatomical location and putative cell population, then your GCaMP lick signal is likely an artefact.

1c. Another control analysis is to estimate the z-motion in your tissue and ask how well that correlates with your signal. There are several ways to do this and you will have to see what works best with your data. For instance, you can look at the change in ROI shape induced at the onset of licking: if the ROI diameters change in a way that correlates with the direction of z-scored activity change, as cells move slightly in and out of the imaging plane, this suggests an artefact. Another way to estimate the motion artefact is outlined briefly in the methods of Gauthier & Tank 2018 on page e3: correlate the mean FOV image at slow or still speeds (using your own speed thresholds for these categories) with the FOV image at each frame. You could also estimate z-motion using the methods available from suite2p. Then for each ROI, plot the motion artefact estimate (one sample per imaging frame) against the ROI's dF/F at each frame. A linear relationship between the dF/F values and the motion artefact will indicate that z-motion is driving small changes in dF/F. In theory you could then fit a linear regression to these data to regress out the effect of z-motion. However, doing the control experiments with GFP is more powerful than these analyses.

To be clear, I believe the manuscript could stand on its own without the lick coding conclusions, so if you find an artefact, I encourage you to just omit the lick coding from your revision. The place and speed-related findings from imaging dHPC->NAc cells are novel and interesting enough on their own to publish. I do hope I am wrong about the artefact, and I look forward to seeing your controls.

2. As a less major point, do the neural findings especially in Fig. 2 hold up within animal, at least trend-wise? It would be important to know that the result of higher spatial coding in the dHPC->NAc cells is not driven by one or two animals having really good labeling and/or better place cells, or greater labeling in dCA1 compared to subiculum. Could the authors show panels 2d-h (and maybe 3f and 4h) per animal, in a supplemental figure? I think it is ok if there aren't enough cells to reach significance in each animal, as long as the trend is similar across animals.

3. The optogenetic results seem more in line with the findings of Britt et al. 2012 to me than direct evidence for a dHPC->NAc projection role in spatially-guided appetitive licking per se. That is, Britt et al. found that axonal stimulation of pretty much any glutamatergic input to the NAc can elicit "appetitive" behavior, i.e. CPP in their task. The mouth movement elicited here is consistent with that, more so than "a driving force in the generation of task-relevant behavior" (line 230) as the authors claim. Perhaps if the dHPC->NAc stimulation had elicited licking on a novel treadmill after being trained to lick in a hidden reward zone on a previous treadmill, this would suggest that a place-reward association was being activated. I don't think the authors need to do this experiment, but I do suggest they adjust their discussion of the optogenetic results.

Minor:

4. Could you clarify in the methods and main text whether there were 2 groups of mice with different reward zone locations? Upon first read it sounded like all the mice had the same location at the same cue transition, and I didn't realize that this might not have been the case until Fig. S1. It actually strengthens the results if there are 2 cohorts with different locations, as this suggests the properties of dHPC->NAc neurons are specific to a hidden reward zone and not to a particular cue combination; but in the main figures, the reward zone is always shown at the cohort 2 location (before the vertical stripes). Are the data in the main text only from cohort 2? Were the 18 mice included in Fig. S1 and Fig. 1 only used for behavior, and not imaging? Please clarify which results come from cohort 1, 2, or both.

5. Can you show the distribution of speed preferences for the speed-excited and speed-inhibited neurons in each subpopulation? Specifically, I'm wondering whether all dHPC->NAc neurons are active when the mouse is basically stopped, or are some active at other intermediate speeds?

6. Fig. 5 needs vertical scale bars for dF/F (or z-scored deconvolved activity, whichever is shown) in all panels with example traces. It would be nice to see dF/F or z-score instead of a.u. in Fig. 4 as well.

7. Fig. 5d: Please add anatomical axes for AP/ML.

8. I found the description of GLM results on lines 265-286 a bit confusing. On line 277-279 you say “the average drop in variance explained to the full model was significantly stronger in dHPC->NAc neurons compared to dHPC- for position and velocity but not licking.” But subsequently, you seem to claim that conjunctive coding of all 3 variables is significant, whereas it sounds like the modeling result only indicates significance for velocity and position? If the lick coding analysis stands after addressing comment #1, I would appreciate clarification on this point.

9. In the introduction on line 56-57, you say that the NAc receives “strong” projections from both the dorsal and ventral subiculum, but as far I know this is not the consensus in the field. Ventral subiculum and CA1 send strong projections, but the dorsal subiculum projection is weaker anatomically and the direction innervation from dCA1 wasn’t confirmed until a few years ago with the Trouche et al. 2019 paper, to my knowledge. It would be worth emphasizing this contribution of the Trouche paper since you are putatively recording from the same direct-projecting CA1 cells, and therefore your study adds a nice confirmation that this direct projection is contributing to the computation of reward locations.

We thank the reviewers for their general enthusiasm and critique. We have followed their suggestions very closely and have consequently made extensive additions/revisions. For ease of reading, we have italicized the reviewers' comments and explain our responses to each point in blue font.

Reviewer #1 (Remarks to the Author):

This manuscript uses 2P imaging in Thy1-GCaMP6s mice combined with retrograde labeling to probe general neuronal activity and activity specifically in a subset of dorsal hippocampus of neurons projecting specifically to the nucleus accumbens while head-fixed mice run on a rotating track and encounter milk reward. The majority of the manuscripts reports results from thorough and rigorous analyses of this data from neurons recorded in 18 mice (?), revealing enrichment of place cells amongst the dHPC-NAc neurons (relative to the general dHPC population) as well as increased decoding accuracy of a reward zone. Among other findings, they identify an interesting dissociation in the neural correlates of consummatory licking and appetitive licking and an enrichment of appetitive-lick excited neurons amongst dHPC-NAc neurons that hints at a possible specialized roles for this projection pathway in behaviorally relevant conveying information in the context of reward rather than reward per se. This is potentially very interesting and could be further developed, particularly with causal manipulations with nuanced behavioral readouts.

A second experiment examined the effect of optogenetic stimulation of dHPC-NAc terminals within the same behavioural design. ChR2- (n=4), but not EYFP-expressing (n=3), mice showed reduced velocity and increased licking at the onset of optogenetic stimulation. This is interpreted as further evidence that dHPC-NAc conveys task-relevant information about appetitive behaviors required to obtain rewards.

This is an interesting and thorough investigation of how neural activity in dHPC, and dHPC-NAc in particular, is modulated as animals run in a rewarding environment. The experimental approach is simple yet powerful and the analyses of neural data provide new insight into task-relevant information that is recoverable from this neural signal. However, data directly demonstrating that this signal is 'guiding goal-directed appetitive behavior' are somewhat limited.

Major comments

- While the analysis of neural data is thorough and compelling, the findings from the optogenetic experiments remain preliminary. More work is needed to establish the claim that dHPC-NAc is a driving force in generating task-relevant behaviour. For example, it would have been interesting to follow up more directly on the neural dissociation between consummatory and appetitive licking. Is the 'mouth motion' induced in other contexts or is it specific to a context where reward has been available? An inhibition experiment could also help to bolster the present results and clarify what the neural signal is conveying. Further, given the central claim of the paper of a specialized function for the dHPC-NAc pathway, as compared to dHPC in general, it would be interesting to see some comparison between a pathway-specific vs non-specific manipulation.

Based on the reviewer's suggestion to further clarify the causal role of dHPC→NAc projections in driving spatial reward memory task-relevant behaviours, we have now added three new sets of data:

1. We have injected a new cohort of n=10 mice (5f/5m) with the retro-AAV-driven optogenetic silencer ArchT (n=5) or EYFP control (n=5) under the CaMKIIa promoter bilaterally into the NAc, and implanted bilateral optic light fibres above dHPC (Figure 1e, f). After recovery, habituation, and food restriction, we trained mice for four days on the previously employed head-fixed spatial reward learning task until they reliably engaged in appetitive licking as they approached the reward zone. On the fifth day, we provided 561 nm light stimulation to silence dHPC→NAc neurons on the last third of the track, leading up to and including the reward zone. We found that **ArchT-expressing animals showed a significant reduction in appetitive licking immediately preceding the reward zone** compared to EYFP control animals (Figure 1g-i). Interestingly, we found no effect of our optogenetic inhibition during reward consumption, and mice still obtained a comparative number of rewards per trial (Supplementary Figure 2a), suggesting a specific deficit for mice to determine the precise location at which to engage in licking. These data suggest that dHPC→NAc neurons are acutely required for appetitive, but not consummatory, licking as mice approach a learned reward zone.
2. To understand if this pathway is also relevant for spatial reward learning, we used the same ArchT cohort and, on days 6-8, replaced the previous reward zone with a new reward zone on the opposite side of the track. Animals received light stimulation on one third of the track leading up to and including the new reward zone from the first day of training to the last (days 6-8). We found that **ArchT animals showed a significant deficit in learning the precise spatial location of this new reward zone**, as evidenced by significantly lower appetitive licking preceding the reward zone compared to control animals (Figure 1j). Again, the number of rewarded laps was comparable across groups (Supplementary Figure 2a). These data show that dHPC→NAc neurons are not only required for the expression but also the learning of spatial reward locations.
3. Finally, we analysed a new dataset in which our previous ChR2 cohort (CaMKIIa-driven ChR2 in dHPC, light fibers implanted in NAc) learned to lick in a reward & stimulation zone in which they needed to lick to obtain reward and were also stimulated with 473 nm light (Supplementary Figure 2g). We found that **ChR2 mice showed a significant shift in their licking towards the area preceding the stimulation & reward zone very early in the training period (after only two days)**, while control mice showed relatively less anticipatory licking at these early training time points. These data suggest that dHPC→NAc projections not only elicit appetitive behaviours, but that they can also enhance learning evidenced by increased appetitive licking even before stimulation.

Regarding the reviewer's question on reward context specificity, we would like to point out that in our previous ChR2 experiments, mice demonstrated mouth motion even in the absence of any task-specific reward. Mice were, however, previously habituated to the treadmill and trained to lick in a location-independent and random manner to obtain some reward. This protocol was necessary to maintain engagement throughout the experiment. As such, this provides evidence that driving the dHPC→NAc pathway, even in the absence of a specific reward context, can drive appetitive behaviours.

While we agree with the reviewer that comparing the causal effects of dHPC→NAC projections with other similar projections or a projection-unspecific activation would be interesting for future investigations, we think that such experiments are beyond the scope of our current work.

- Methods indicate that experiments were conducted in male and female mice but no further information is provided. The numbers of each sex must be indicated for each experiment. Were any analyses performed to determine if there were sex differences? At a minimum, indicating the sex of all individual data points (e.g. by using 2 different colours) would increase transparency and facilitate comparison of any potential sex differences by interested readers.

We agree with the reviewer that the effects of sex in animal models need to be fully considered and have taken this opportunity to clarify any potential sex effects in our data. For this, we have done the following:

- 1) We are now **reporting the numbers for each sex** for each experiment in the Methods (lines 476-478). They are 3m/6f for behaviour, 3m/4f for ChR2 optogenetics, 5m/5f for ArchT optogenetics. Additionally, 6m were used for calcium imaging; no differences in cellular coding properties based on calcium dynamics have been reported between sexes, and many imaging studies frequently use male-only cohorts to reduce behavioural variability and animal numbers (see Bowler & Losonczy, 2023, Neuron, Priestley et al., 2022, Neuron; Grosmark et al., 2021, Nature Neuroscience; Doron et al., 2022, Nature; Cholvin & Bartos, 2022, Nature Communications; Woods et al., 2020, Neuron).
- 2) We have **indicated the sex of all individual data points** by circles (female) or squares (male) in Figures 1i, j, o, and Supplementary Figure 2f, g. We saw no clear difference in effects between sexes.
- 3) Furthermore, we have now performed 3-way mixed ANOVAs and 2-way ANOVAs to include **sex as a factor** (between-subject factors: group, sex, within-subject factor: optogenetic stimulation), and found no significant main effects or interaction effects with sex as a factor. In the following, we report these statistical results, and have included them now in the manuscript as **Supplementary Tables 1-6**.

Related to Figure 1i:

Effect	DFn	DFd	F	p	p<.05	ges
Group	1	6	2.84	0.143		0.133
Sex	1	6	0.0130	0.912		7.10E-04
Day	1	6	3.19	0.124		0.264
Group:Sex	1	6	3.78	0.1		0.170
Group:Day	1	6	0.596	0.469		0.0630
Sex:Day	1	6	5.25E-06	0.998		5.91E-07
Group:Sex:Day	1	6	0.0120	0.917		0.0010

DFn: degrees of freedom numerator (based on factor levels); DFd: degrees of freedom denominator (based on animals); ges: effect size generalized eta squared

Related to **Figure 1j**:

Effect	DFn	DFd	F	p	p<.05	ges
Group	1	5	5.90	0.059		0.393
Sex	1	5	1.55	0.268		0.146
Day	1.1	5.52	0.183	0.006	*	0.622
Group:Sex	1	5	0.134	0.729		0.015
Group:Day	1.1	5.52	7.62	0.034	*	0.407
Sex:Day	1.1	5.52	0.289	0.634		0.025
Group:Sex:Day	1.1	5.52	0.99	0.372		0.082

DFn: degrees of freedom numerator (based on factor levels); DFd: degrees of freedom denominator (based on animals); ges: effect size generalized eta squared

Related to **Figure 1o**:

Effect	DFn	DFd	F	p	p<.05	ges
Group	1	3	0.171	0.707		0.0540
Sex	1	3	2.38	0.221		0.441
Opto	1	3	0.228	0.017	*	0.0190
Group:Sex	1	3	0.0570	0.827		0.0180
Group:Opto	1	3	8.29	0.064		0.0070
Sex:Opto	1	3	0.266	0.641		2.25E-04
Group:Sex:Opto	1	3	0.0890	0.784		7.56E-05

DFn: degrees of freedom numerator (based on factor levels); DFd: degrees of freedom denominator (based on animals); ges: effect size generalized eta squared

Related to **Supplementary Figure 2g, Day 0**

Source	SS	DF	MS	F	p-unc	np2
Group	0.0271	1.0	0.027136	0.243799	0.655364	0.075159
Sex	0.0522	1.0	0.052221	0.469162	0.542549	0.135238
Group * Sex	0.0161	1.0	0.016061	0.144299	0.729341	0.045892
Residual	0.3339	3.0	0.111306	NaN	NaN	NaN

SS: sums of squares; MS: mean squares; p-unc: uncorrected p-values; np2: partial eta-square effect sizes

Related to **Supplementary Figure 2g, Day 1**

Source	SS	DF	MS	F	p-unc	np2
Group	0.0809	1.0	0.080869	3.978938	0.140040	0.570135
Sex	0.0246	1.0	0.024573	1.209033	0.351847	0.287247
Group * Sex	0.0052	1.0	0.005224	0.257057	0.647068	0.078923
Residual	0.0610	3.0	0.020324	NaN	NaN	NaN

SS: sums of squares; MS: mean squares; p-unc: uncorrected p-values; np2: partial eta-square effect sizes

Related to **Supplementary Figure 2g, Day 2**

Source	SS	DF	MS	F	p-unc	np2
Group	0.2125	1.0	0.212550	9.036332	0.057392	0.750755
Sex	0.0001	1.0	0.000074	0.003135	0.958870	0.001044
Group * Sex	0.0057	1.0	0.005687	0.241776	0.656656	0.074581
Residual	0.0706	3.0	0.023522	NaN	NaN	NaN

SS: sums of squares; MS: mean squares; p-unc: uncorrected p-values; np2: partial eta-square effect sizes

Minor comments

- *dHPC-NAc neurons are 'more likely to be speed-inhibited and less likely to be speed-excited'(line 246-247. Is it surprising then that optogenetic stimulation of dHPC-NAc suppresses velocity?*

Indeed, using calcium imaging, we found that dHPC^{→NAc} neurons as a whole were more speed-inhibited than dHPC⁻ neurons (Figure 4), and that dHPC place cells were more likely to be speed-inhibited and less likely to be speed-excited (Figure 6a, b). In addition, we found appetitive lick modulation of dHPC^{→NAc} neurons (Figure 5). Our optogenetic findings that activating dHPC^{→NAc} terminals drives deceleration (Supplementary Figure 2e, f) and lick-related mouth motion (Figure 1k-o) are thus in line with these imaging results. Given that mice slow down when approaching a learned reward zone, we interpret this induced deceleration as a form of appetitive behaviour. Together, both methods converge to demonstrate an important role of this pathway for guiding spatial memory-driven appetitive behaviours.

- *The optogenetic experiment includes two groups (ChR2 and EYFP) with within group comparisons (pre, post stimulation). The data are analyzed as two separate paired t-tests. The appropriate analysis is a two-way repeated measures ANOVA.*

We have now performed two-way mixed ANOVAs with within-subject comparisons (pre/post stimulation) and between-subject comparisons (ChR2/EYFP). We have extended the figure legend and statistical methods sections accordingly. We note that all ANOVAs show significant main effects for the stimulation, and that we have performed Bonferroni post-hoc tests to compare group-specific differences in stimulation outcome.

Legend Figure 6:

i, Mouth motion is significantly increased with optogenetic stimulation in ChR2 animals ($t(3) = 7.485$; $P = 0.00494$) but not EFYP animals ($t(2) = 1.353$; $P = 0.309$). Mixed ANOVA, main effect group $F(1, 5) = 0.036$, $P = 0.856$, main effect stimulation $F(1, 5) = 42.47$, $P = 0.0013$, interaction $F(1, 5) = 14.72$, $P = 0.0122$, followed by post-hoc Bonferroni-corrected t-tests. [...]

k, Velocity is significantly decreased with optogenetic stimulation in ChR2 animals ($t(3) = -3.551$; $P = 0.0381$) but not EFYP animals ($t(2) = -1.263$; $P = 0.334$). Mixed ANOVA, main effect group $F(1, 5) = 0.112$, $P = 0.752$, main effect stimulation $F(1, 5) = 12.31$, $P = 0.017$, interaction $F(1, 5) = 2.80$, $P = 0.155$, followed by post-hoc Bonferroni-corrected t-tests.

Also, for the additional data we present now on optogenetic inhibition experiments (Figure 1i, j), we include ANOVA test results as reported in the respective figure legends.

- *Descriptions of number of mice included in each experiment are confusing. The methods refer to 16 C57/Bl6 mice and $n = 6$ Thy1-GCaMP6s mice (or are these the original mice used*

to generate the experimental mice?). Figure 1 and FigS1 refer to $n = 18/2$ cohorts of $n = 9$ mice. The presented optogenetic experiments present data from 7 mice although reference is made to $n = 6/\text{group}$ in the methods for this experiment. The discrepancy between the starting and final n for the optogenetic experiment should be clearly explained. The sample size for this experiment is very low ($n = 3/4$). How was the appropriate sample size determined?

We agree that the previous presentation of mouse numbers could be confusing and have updated the text in the Methods and Results. We now provide the animal numbers used in a clearer way. In brief, they included 18 mice for behavioural experiments, of which 6 were used for calcium imaging. In addition to these, we used 7 mice for optogenetic activation and 10 mice for optogenetic inhibition experiments.

- We have added further clarifications of animal numbers in our **Results** section:
 - line 84 (“we trained food-restricted mice ($n = 18$)”)
 - line 93 (“we used dual-color two-photon imaging in a subset of mice ($n = 6$)”).
 - Optogenetic sample sizes have been previously accurately reported, e.g., line 220-221 (“we injected animals with either CaMKII α -driven ChR2 or EYFP into dHPC and implanted light fibers in the NAc ($n = 4/3$ mice)”), and have been updated to include the optogenetic inhibition experiments, lines 98-99 (“[...] we injected C57Bl/6 mice ($n = 5/5$ mice) with retrograde-targeted CaMKII α -driven ArchT or EGFP into the NAc [...]”).
- We also added a clarification in the **Methods** part:
 - lines 393-394: “C57Bl/6 ($n = 26$, out of which $n = 9$ (3 males, 6 females) for behavioral training and $n = 17$ (8 males, 9 females) for optogenetic experiments) and Thy1-GCaMP6s ($n = 6$ males for imaging) mice [...]”.
 - line 487: “Two-photon imaging was performed in $n = 6$ mice using...”
 - line 569: “in $n = 10$ mice (5 female, 5 male), aged between 14 to 15 weeks”
 - lines 502-504: “For optogenetic activation experiments[...], the skin over the parietal skull [...] in $n = 7$ mice”
- We have determined the sample size based on previous literature (cp. Turi et al., 2019, Neuron, Fig. 6; Kaufman et al., 2020, Neuron, Fig. 3; Dudok et al., 2021, Neuron, Fig. 3; Etter et al., 2019, Nature Communications, Fig. 3; Li, Lu, Ren, et al., 2022, Nature Communications; Britt et al., 2012, Neuron, Fig. 6; Jiang, Xu, Dudman, 2022, Nature Neuroscience, Fig. 6; Park et al., 2016, eLife) and our own power analysis, keeping in mind the 3Rs in an attempt to reduce the number of animals needed and taking into account the technical challenge of the experiments. While we would agree that the number of animals used for the optogenetic activation experiment is on the lower end, we would like to point out that these ChR2 experiments are now complemented with additional ArchT silencing experiments, showing opposite effects on appetitive licking. Altogether, we show optogenetic data from $n = 17$ mice demonstrating a light- and virus-specific bidirectional effect.

- What age were mice at the time of surgery/experimentation?

We have now included the age spans of all mice for all experiments reported in the Methods section of our manuscript. Overall, mice were between 9-25 weeks old.

- Line 506 (surgery): “For stereotactic injection of AAVs, 9-18 week old mice...”
- Lines 564-565 (behavioral training): “12-20 week old mice were placed daily...”

- Line 582 (imaging): “Two-photon imaging was performed in n = 6 mice aged between 15 to 25 weeks...”
- Line 599 (optogenetic inhibition): “For optogenetic inhibition experiments, [...] aged between 14 to 15 weeks.” and line 613: “head-fixed mice aged 18-20 weeks received light stimulation”
- Line 623 (optogenetic activation): “For optogenetic activation experiments..., aged between 10 to 11 weeks” and line 635 “head-fixed mice aged 13 to 15 weeks received light stimulation”

- What is the rationale for choosing 20Hz/5 ms stimulation? How does this related to known physiological firing of dHPC-NAc neurons or neural activity recorded in the present manuscript?

We chose these stimulation parameters in accordance with the most relevant recent literature. Britt et al. (2012, Neuron) used 5ms stimulation and 20Hz for all nose-poke and self-stimulation experiments to investigate the effects of stimulating ventral hippocampal terminals in the NAc. Lindenbach et al. (2022, BehBrainRes) employed a 20Hz/5ms pattern to explore vSub to NAc shell projections and its effect on DA release. Similarly, Prado et al. (2016, JNeurosci) stimulated glutamatergic inputs reaching the NAc in a nose-poke task at 20Hz, as well as Trouche et al. (2019, Cell) who used 5ms ChR2 stimulation of glutamatergic dCA1 terminals in NAc. Finally, Cole et al. (2018, PLoS One) stimulated NAc MSNs with 25Hz in self-stimulation protocols, and 20 Hz was also the chosen frequency of optogenetic stimulation for dissecting the role of amygdala projections to the NAc (see Howland et al., 2002, JNeurosci; Stuber et al., 2011, Nature). Importantly, all these studies used similar frequencies for brief bouts of stimulation (lasting up to a few seconds), such as the one we employed in our study.

As it is impossible to infer exact spike rates from calcium imaging (see Ali & Kwan, 2019, Neurophotonics; Sabatini, 2019, bioRxiv; Wei et al., 2020, PLoS CompBio), we do not claim to know the spike rates of the neurons we recorded from. However, a wide range of studies have recorded single-unit and field potential activity from HPC CA1 neurons, and it is well accepted that pyramidal CA1 neurons can fire at 20 Hz frequencies in the so-called low-gamma rhythm (Carr et al., 2012, Neuron; Vaidya and Johnston, 2013, Nature Neuroscience). Gamma oscillations are considered a mechanism for routing of different information sources in the brain (Colgin et al., 2009, Nature; Gattas et al., 2022, eLife). Given our multi-sensory task including spatial, textile, and reward-related information, low-gamma patterned stimulation was favoured in our settings instead of other frequencies.

To illustrate this reasoning for the readers of the manuscript, we have now added three key citations to the respective text in our Results, lines 116-119 (added text in bold): “After habituating mice to run on the treadmill and receive rewards upon licking on the lick spout, mice were given 5 mW of 473 nm 20 Hz (5 ms duration) pulsed laser light for up to 10 seconds (Stuber et al., 2011; Britt et al., 2012; Lindenbach et al., 2022) upon entry into a hidden light stimulation zone.”

- What kind of ‘milk reward’ was used?

This has been previously described in the Methods section, lines 565-567: “they needed to lick the metal spout in this hidden reward zone to obtain a liquid reward in the form of condensed milk.”

We used 12% fat condensed milk as an alternative to sucrose or soy milk reward as we previously observed condensed milk to be preferred over other alternatives by our mice (unpublished observations). Condensed milk reward has also been employed by other groups (e.g., Mead & Stephens, 2003, JNeuro; Rodriguez et al., 2019, eNeuro; Reisel et al., 2002, NatNeuro; Dent et al., 2014, EJN).

Reviewer #2 (Remarks to the Author):

This is an exciting paper examining the role for dCA1 neurons that project to the nucleus accumbens in head-fixed, behaving mice. The authors use an established reward-memory task to compare the encoding properties and function of dCA1-Nac-projecting versus non-Nac-projecting neurons. They find Nac-projecting cells are more likely to have place fields and they tend to have enhanced spatial encoding properties. They also tend to encode more than just space, and conjunctively encode deceleration and appetitive licking which helps decode locations where rewards are expected. Further, the authors show that opto-activation of CA1 terminals in Nac drives deceleration and licking behavior. Altogether these results are an important addition to the literature. I have some suggestions to improve the paper:

The FOVs of CA1 in Fig. S3 look a little odd compared to typical dCA1 FOVs under 2P imaging, especially mouse 210. Do the authors have a reason why the pyramidal neurons in these FOVs are not packed tightly together as is typical? Why does the expression of GCaMPs look so sparse? Each FOV from these mice look like a recording from a different location within HPC. Should data from these mice be pooled given the difference in these FOVs? Mouse 210 looks especially different from what is expected.

We thank the reviewer for their input and opportunity to clarify some of our findings. The main reason why several FOVs do not look like typical dCA1 FOVs is because they have been recorded from the dCA1-subiculum border region, encompassing both parts of dorsal proximal subiculum and dorsal distal CA1 (cp. Gauthier & Tank, 2018, Neuron). We have previously mentioned this in our Results, e.g., lines 134-135: “we injected AAVrg-Cre in NAc and the static red fluorophore DIO-mCherry in dHPC (CA1/subiculum border region) of Thy1-GCaMP6s mice”, and lines 141-143: “Optical access to dorsal CA1 and prosubiculum [...] was established by implanting a chronic hippocampal window”. We also point out putative delineations in Supplementary Figure 3a, c. We have previously pooled data from putative dCA1 and dSub regions because post hoc delineations between the two regions remain difficult to classify with absolute certainty and because the focus of our work lies on the projection identity of these neurons. Regardless, when we analyse the properties of these putative regions separately, we found largely consistent effects (see Figure R1 below). For instance, the spatial tuning properties and GLM results split into putative dCA1 and dSub groups show the same trends throughout, although some measures are not significant in putative subicular neurons, which may be due to the lower N (subiculum: 1099 (of which NAc-projecting: 145), CA1: 4181 (of which NAc-projecting: 297)).

Figure R1: Spatial tuning characteristics and GLM results based on putative delineations between dorsal subiculum (Sub) and dorsal CA1. **a-f**, Data based on Figure 2f-k. **g, h**, Data based on Figure 7b-c. ns: not significant, * $P < 0.05$, ** $P < 0.01$, *** $P < 0.001$.

To further clarify this for the reader, we have added the following in our Results text, lines 144-145 (addition highlighted in bold): “Imaging data from CA1 and prosubiculum were acquired after 5 days of behavioral training, [...]”

Relatedly, the spatial activity of the cells from mouse 210, and somewhat from mouse 207, looks sparse, with many laps showing no activity in the place fields. It is well known that place field firing varies extensively from lap-to-lap (over dispersion), but these cells seem to go beyond what is expected based on other papers. The authors should quantify overdispersion or some related measure such as lap-to-lap reliability to see if their place cells are behaving in an expected way compared to what has been reported. If they are not, then the authors should explain why. I know they compare similar measures across Nac-projecting and non-projecting neurons, but it would be useful to put their results in the context of the many papers that report these features of place cells.

- Indeed, Supplementary Figure 4 (previously S3) contains both place and non-place cells, as indicated by the asterisks behind each neuron’s spatial information (SI) score. For example, mouse 210 also contains 4 non-place cells and mouse 207 contains 3 non-place cells. These cells were included to provide the reader with an intuition for what, in our dataset, we consider a place cell and what not. We thank the reviewer for bringing to our attention that we have previously missed to include in the figure legend an explanation for what these asterisks mean. We have now updated the figure legend accordingly, by adding the following: “Asterisks refer to significant spatial modulation (classified place cells) according to a 1,000x randomly shuffled distribution (* > 95th percentile; ** > 99th percentile; *** > 99.9th percentile).”
- Based on the reviewer’s suggestion, we have now **calculated overdispersion** according to Fenton et al. (2010, JNeurosci). In brief, we calculated an expected calcium activity

within a place field by averaging all z-scored deconvolved calcium events within the calculated place field, and subtracted this expected activity from the observed activity of all calcium event-containing place field passes per lap, divided by the square root of the expected activity (new **Supplementary Figure 5g**). We find higher overdispersion values for dHPC⁻ ($\sigma^2 = 2.42$, N=24,627 passes) compared to dHPC[→]NAc place cells ($\sigma^2 = 2.04$, N=2,460 passes), significantly different at $P < 0.001$ with Levene test for equal variances. To the best of our knowledge, these measures have not previously been reported for calcium imaging data, so we would caution any direct comparisons of electrophysiologically recorded spike rates with our calcium event data, but we find these values to be largely in line with previous reports:

- $1.74 < \sigma^2 < 4.87$ (Olypher et al., 2002, Neuroscience)
- $4.6 < \sigma^2 < 6$ (Jackson & Redish, 2007, Hippocampus)
- $3 < \sigma^2 < 5$ (Wikenheiser & Redish, 2010, JNeurophysiol)
- $2.1 < \sigma^2 < 5.5$ (Fenton et al., 2010, JNeurosci)
- $4 < \sigma^2 < 6$ (Hok et al., 2012, JNeurosci)
- $2 < \sigma^2 < 3.5$ (Hok et al., 2013, JNeurosci)
- $2.8 < \sigma^2 < 3.8$ (Thibault et al., 2018, JNeurosci)
- $1.6 < \sigma^2 < 2.33$ (Sanders et al., 2019, Hippocampus)
- $\sigma^2 \sim 3.2$ (Duveller et al., 2019, JNeurosci)
- As another measure of overdispersion, we have previously shown “**place field reliability**”, which has been defined in the Methods section (lines 694-695) as “the fraction of laps in which the maximum spatially binned deconvolved calcium activity occurred within the cell’s place field”. Supplementary Figure 3 shows reliability (“Rel”) values for all neurons depicted below the FOV. Reliability is significantly higher in dHPC[→]NAc (0.56) neurons compared to dHPC⁻ (0.50) (Fig. 2i).
- In response to the reviewer’s suggestion to measure **lap-to-lap reliability**, we are now including the measure of “**place field stability**” (**Figure 2j**) which is measured as the average Pearson’s r of position-binned deconvolved calcium activity patterns across all individual laps. We find median values of 0.23 for dHPC⁻ and 0.33 for dHPC[→]NAc neurons. This is comparable to previous studies. For example, Cholvin et al. (2021, Neuron, Figure 2C) and Hainmueller & Bartos (2018, Nature, Figure 3f) both report a stability of ~0.25 for CA1 pyramidal cells using calcium activity in mice, while Markus et al. (1994, Hippocampus) reported reliability around 0.2 for rat CA1 cells.

Fig. 3 compares the distribution of PFs around boundaries and reward zones in Nac-projecting and non-projecting place cells. Significance is reported in most comparisons using the Wilcoxon t-test, but this test is highly sensitive to the number of samples used, and p-values are driven very low when n is high, over-estimating differences. Given the high n in these comparisons, and what looks like small differences between groups, I recommend the authors conduct estimation approaches on these datasets¹. This emphasizes reporting effect sizes with expressions of uncertainty (interval estimates) to make better inferences: it pushes back against over-confident claims from inadequate samples, improves comparisons of results across contexts, and normalizes the publication of negligible effects¹. I think this method of comparison should be used in all cases where n is high and the authors are comparing 2 groups (so not just in Fig. 3). This is especially important in Fig. 3c,d, and g. Relatedly, in Fig. 3f, the authors state that the proportion of place fields in reward and vicinity zone is significantly higher in high-success trials (colored bars) compared to low-success trials (gray bars) in NAc-projecting neurons (red) but not in dHPC- neurons (green). This

conclusion is based on the green bar having a Pval of 0.051 compared to the gray bar. If this pval was only marginally lower, say 0.049, the authors would have a different conclusion. Again, this is why reporting effect sizes is more informative than drawing conclusions based on arbitrary Pval thresholds.

1 Calin-Jageman, R. J. & Cumming, G. Estimation for Better Inference in Neuroscience. *eNeuro* 6, doi:10.1523/ENEURO.0205-19.2019 (2019).

We concur with the reviewer that estimation statistics offer many advantages for statistical interpretation of data compared to the more established and widely used statistical methods we have previously displayed. We would like to stress though that the latter are also valid and may be more easily interpreted. To account for this, while also addressing the reviewer's concerns, we have included estimation statistics in supplementary figures. For this, we used the Python package DABEST from the Claridge-Chang lab (Ho et al., 2019, *NatMethods*) based on 5,000 bootstrapped samples each, and reported effect sizes and confidence intervals in the figure legends. Specifically, we did the following:

- Supplementary Figure 5a-f shows estimation statistics for data shown in Figure 2f-k.
- Supplementary Figure 5j-l shows estimation statistics for Figures 3c,d and S5i.
- Supplementary Figure 5p-q shows estimation statistics for Figure 3f, g.
- Supplementary Figure 6a, b shows estimation statistics for Figure 4d, h.
- Supplementary Figure 8a-c shows estimation statistics for Figure 5h-i.
- Supplementary Figure 9 shows estimation statistics for Figure 6b, d, f.
- Supplementary Figure 10d-k shows estimation statistics for Figure 7b, c.

We find that all previously observed significant results also remain significant at a 95% confidence interval.

Regarding Figure 3f, estimation statistics (Supplementary Figure 5p) suggests that dHPC⁻ neurons also have a significantly higher proportion of place fields near the reward zone in high-success trials compared to low-success trials, and that contrasting high-success trials in dHPC⁻ with dHPC^{→NAc} neurons does not show a significant difference despite a visible trend at P = 0.136. We have thus toned down our conclusions in the text as follows:

- Lines 204-208: "At the same time, dHPC⁻ neurons showed a trend towards significance using t-statistics (P = 0.051, Welch's t-test; Fig. 3e, f) that reached significance using estimation statistics (Supplementary Fig. 5p). Based on this, we conclude that both populations overrepresent the reward zone during high-success trials, but this effect is particularly pronounced in dHPC^{→NAc} neurons."

Lines 157-158 – "...previous work demonstrated the dependence of such an overrepresentation on individual task success (Dupret et al., 2010; Danielson et al., 2016)." Relatedly, a recent paper showed that reward over-representation in a similar experimental setup was dependent on the animal's reward expectation (Krishnan et al, 2022) and another showed that the number of place fields in an environment is dependent on behavioral engagement (Pettit et al., 2022). The authors may want to add a comment about these papers here or elsewhere in the manuscript as they both seem highly relevant to the results in this paper:

2 Krishnan, S., Heer, C., Cherian, C. & Sheffield, M. E. J. Reward expectation extinction restructures and degrades CA1 spatial maps through loss of a dopaminergic reward proximity signal. *Nat Commun* 13, 6662, doi:10.1038/s41467-022-34465-5 (2022).

3 Pettit, N. L., Yuan, X. C. & Harvey, C. D. Hippocampal place codes are gated by behavioral engagement. *Nat Neurosci* 25, 561-566, doi:10.1038/s41593-022-01050-4 (2022).

We agree with the reviewers that these articles are highly relevant for our work, and incorporated these into our text as follows (additions highlighted in bold):

- Lines 197-201: “However, mouse behavior may show high variability, and previous work has demonstrated **the importance of behavioral engagement for spatial representations** (Pettit et al., 2022) and the dependence of spatial reward overrepresentations on individual task success **and reward expectation** (Dupret et al., 2010; Danielson et al., 2016; Krishnan et al., 2022).”
- Lines 398-400: The mechanistic basis of such spatial reward overrepresentations likely depends on the interaction of inputs from entorhinal cortex layer 3, hippocampal NMDA receptors, and dopaminergic inputs from VTA and LC (Dupret et al., 2010; McNamara et al., 2014; Kaufman et al., 2020; Grienberger and Magee, 2022; **Krishnan et al., 2022**).

The optogenetic activation of dCA1 terminals in Nac is very interesting. Did the authors think about or attempt to perform an inhibitory experiment whereby these terminals are inhibited in the learned reward zone? This would strengthen the argument that the CA1-Nac pathway is involved in appetitive behaviors associated with reward.

We have now performed a **new set of experiments** (n = 10 mice) in which we **optogenetically inhibited NAc-projecting dHPC neurons** during the retrieval of learned spatial reward memories and during the learning of a novel reward zone (new Figure 1e-j). Our new data show that silencing dHPC^{→NAc} neurons during the approach of the reward zone significantly reduces the amount of appetitive licking in direct anticipation of the reward zone. We also show that silencing these neurons during the learning of a novel reward zone significantly reduces learning outcomes. These new data thus clearly demonstrate the necessity of dHPC^{→NAc} neurons to guide spatial reward memory-guided appetitive behaviours.

Lines 314-315 in reference to dCA1/vCA1 comparisons, the authors could add this very recent and relevant paper from the Kheirbek lab:

4 Biane, J. S. et al. Neural dynamics underlying associative learning in the dorsal and ventral hippocampus. *Nat Neurosci*, doi:10.1038/s41593-023-01296-6 (2023).

We thank the reviewer for their suggestion. We have now incorporated this citation in a new discussion paragraph on ventral/dorsal HPC differences, lines 368-373 (addition highlighted in bold):

“Hippocampal projection-specific coding has been previously demonstrated in ventral CA1 (Cocchi 2015). Indeed, functional differences along the hippocampus’ septotemporal axis have long been established, with the dorsal part credited with mapping cognitive relationships in the world, and the ventral part with valence processing (Fanselow 2010). **This view is also supported by a recent study on odor-outcome learning that found robust odor coding in dCA1, while vCA1 odor responses only emerged after acquiring positive valence (Biane 2023).**”

Line 323, the authors mention dopaminergic VTA inputs to hippocampus as they may relate to their Nac projecting neurons. The Krishnan et. al. 2022 paper mentioned above recorded from these DA inputs in a similar behavioral setup. The authors could briefly discuss their results in the context of the reported VTA-CA1 signals. It could help give mechanistic insight into how DA might influence encoding by Nac CA1-projecting neurons associated with reward.

We thank the reviewer for their suggestion and have now extended our Discussion by an additional paragraph on the putative role of DA for our findings, with a focus on Krishnan et al.'s findings, see lines 398-416:

"The mechanistic basis of such spatial reward overrepresentations likely depends on the interaction of inputs from entorhinal cortex layer 3, hippocampal NMDA receptors, and dopaminergic inputs from VTA and LC (Dupret et al., 2010; McNamara et al., 2014; Kaufman et al., 2020; Grienberger and Magee, 2022; Krishnan et al., 2022). Specifically dopaminergic release from VTA terminals in dHPC has been suggested to stabilise spatial representations and enhance learning performance: optogenetic activation of dopaminergic VTA terminals in dHPC was shown to induce a shift in place fields and to enhance offline reactivation and memory recall (McNamara et al., 2014; Mamad et al., 2017), possibly through hippocampal D1/5 receptors (Retailleau and Morris, 2018). Two recent articles demonstrated that VTA terminals in dHPC show characteristic "ramp" activity as mice approach a learned reward zone (Krishnan et al., 2022; Heer and Sheffield, 2023). Both loss of reward expectation and optogenetic silencing of these terminals resulted in a widespread degradation of spatial coding throughout dHPC place cells (Krishnan et al., 2022). Given our findings of enhanced spatial and reward-related coding in dHPC^{→NAc} neurons, it is tempting to speculate these might be the result of enhanced DA receptor expression in these neurons. Interestingly, dopaminergic inputs to dHPC^{→NAc} neurons may form part of a feedback loop by which NAc neurons could modulate VTA activity (Floresco et al., 2001; Lisman and Grace, 2005) that, in turn, may change DA release in HPC. Such a feedback loop (HPC-NAc-VTA-HPC) could play an important role for memory updating, as functionally similar pathways have been recently found to drive extinction and reconsolidation learning in *Drosophila* (Felsenberg et al., 2017). Future studies could investigate specific dopaminergic contributions using novel sensitive neurotransmitter indicators (Zhuo et al., 2023)."

Reviewer #3 (Remarks to the Author):

Barnstedt et al. provide a very interesting and much needed characterization of hippocampal output to the nucleus accumbens (NAc), a key structure in the transformation of spatial state information into value-guided action selection. Using two-photon imaging of dorsal hippocampal (dHPC) subiculum and the CA1/subiculum border, the authors monitor the activity of the dHPC population in addition to the activity of the NAc-projecting subpopulation (dHPC->NAc), labeled by injection of a retrograde Cre virus in NAc and Cre-dependent mCherry in dHPC. They find that spatial coding is enhanced in the dHPC->NAc population compared to non-mCherry-labeled cells (dHPC-), as supported by a higher fraction of place cells, higher spatial information, higher stability, etc. in the dHPC->NAc subpopulation. They further find that the hidden reward zone in their task is overrepresented more prominently in the dHPC->NAc cells, particularly on successful behavioral trials. The dHPC->NAc subpopulation prefers to fire at slower movement speeds, and the authors are able to more

accurately decode the animal's occupancy of the anticipatory zone before the reward from the dHPC->NAc population compared to the dHPC- population, suggesting the dHPC->NAc subpopulation is biased to encode upcoming reward locations. Together, these results are strong and make an important contribution to our understanding of how the hippocampus might bias its outputs to route specific information to downstream circuits. However, I have significant concerns about the latter part of the manuscript that claims lick coding in the same hippocampal neurons. I've outlined these concerns in detail below, in addition to some more minor comments.

Major:

1. Given the very low amplitude of the lick-onset-triggered signal that you observe (e.g. in Fig. 5f-g, z-score of 0.2-0.4), I have a strong concern that this lick "coding" is driven by z-motion artefact from the mouse slowing down and licking. In our own data using 2P imaging (unpublished, unfortunately), we have observed very similar signal in dorsal CA1 neurons expressing static GFP. I believe that the mouse's licking movement generates a slight z-offset in the brain relative to the imaging plane. This z-motion creates apparent increases in fluorescence in a subset of ROIs, and a decrease in fluorescence in another subset of ROIs that is time-locked to the onset of detected licking – exactly what you observe. This artefact can lead to a ~5% or greater dF/F change in GFP signal, or a z-score of ~0.2-0.4. For the positive-going ROIs, we often see a small decrease in signal before the lick, an increase at licking onset, followed by another small decrease. For the negative-going ROIs, we see an increase preceding the lick, followed by a decrease after lick onset. Again, this looks exactly like what you see in Fig. 5. The fact that you see this to a greater degree in dHPC->NAc neurons is curious, but I hypothesize that it could be explained by 2 variables: (1) the dHPC->NAc neurons are anatomically located in higher density in what I assume is the antero-medial part of your field of view (FOV) shown in Fig. 5 (see point #7 requesting anatomical axes labels here). I assume this is also where the FOV transitions to the subiculum, where the brain may slope downward relative to the imaging glass – is this correct? This depth discrepancy could cause the relative fluorescence change to be higher as a result of licking-induced brain motion. (2) The dHPC->NAc neurons are more active at slower speeds according to your results, which is also when the animal is licking. This means the artefactual dF/F from licking would be more likely to get picked up if the cell is already firing in a place field occurring at low speeds. Moreover, it's important to note that I am unaware of any other paper demonstrating reliable licking signals in subiculum/CA1, a point the authors do not address. I suggest the following controls to account for this possible artefact in your data:

1a. First, perform the same lick-bout-triggered analysis using your static red mCherry signal. If you see a comparable increase/decrease in signal in some of the ROIs at the onset of licking, you definitely have a problem. However, I am not sure if this artefact is the same with red fluorophores as with green, as the red fluorophore signal transmits further through tissue, so it might be less affected by z-offsets.

1b. Therefore, you should perform control experiments with animals expressing static GFP in pyramidal cells (n=2 mice would be sufficient). If you see lick-induced dF/F in these animals, imaged at the same anatomical location and putative cell population, then your GCaMP lick signal is likely an artefact.

1c. Another control analysis is to estimate the z-motion in your tissue and ask how well that correlates with your signal. There are several ways to do this and you will have to see what works best with your data. For instance, you can look at the change in ROI shape induced at the onset of licking: if the ROI diameters change in a way that correlates with the direction of z-scored activity change, as cells move slightly in and out of the imaging plane, this suggests an artefact. Another way to estimate the motion artefact is outlined briefly in the methods of Gauthier & Tank 2018 on page e3: correlate the mean FOV image at slow or still speeds (using your own speed thresholds for these categories) with the FOV image at each frame. You could also estimate z-motion using the methods available from suite2p. Then for each ROI, plot the motion artefact estimate (one sample per imaging frame) against the ROI's dF/F at each frame. A linear relationship between the dF/F values and the motion artefact will indicate that z-motion is driving small changes in dF/F . In theory you could then fit a linear regression to these data to regress out the effect of z-motion. However, doing the control experiments with GFP is more powerful than these analyses.

To be clear, I believe the manuscript could stand on its own without the lick coding conclusions, so if you find an artefact, I encourage you to just omit the lick coding from your revision. The place and speed-related findings from imaging dHPC→NAc cells are novel and interesting enough on their own to publish. I do hope I am wrong about the artefact, and I look forward to seeing your controls.

We thank the reviewer for raising this important issue. Based on the reviewer's suggestions, we have thoroughly re-examined our data and conclude that the lick modulation we described is not the result of motion artefacts. While it is impossible to rule out with absolute certainty, this conclusion is based on the following observations and additional analyses:

1. First, we would like to point out that there is indeed literature describing lick-modulated hippocampal activity, which we have previously acknowledged in lines 422-423: "Similarly, we observed significant modulation of hippocampal neurons during appetitive and consummatory licking (see also (Wiener et al., 1989; Ho et al., 2011)." Ho et al. (2011), for example, identified 112 out of 208 CA1 pyramidal neurons as lick-modulated (see their Table 1) using in vivo electrophysiological methods, which should not be subject to the same motion artefacts as calcium imaging. More recently, Hassan et al. (2023, Neuron) demonstrated social odour coding of CA2 pyramidal neurons that, notably, were also significantly modulated by licking (see Fig. S6B-C) even though the authors' emphasis was more focused on stimulus/memory coding aspects. We also note that our findings pertain specifically to appetitive licking, while consummatory licking (arguably capable of inducing similar motion) does not yield significant calcium responses (see Supplementary Figure 7).
2. Based on the reviewer's suggestion (1a), we have now **extracted each neuron's red static channel's signal** (average fluorescence per spatial component) **and performed the same lick-bout triggered analysis** as for the dynamic green channel (new Supplementary Figure 8g-n). For illustration purposes, we now complement the dynamic signals reported in Figure 5 with the static red signals we extracted, for both dHPC^{→NAc} and dHPC⁻ neurons (Supplementary Figure 8g-i). Unsurprisingly, we see no fluorescence modulation of dHPC⁻ neurons given that they have almost no red static signal to speak of. We have, nonetheless, also included this data. We are now showing single-cell traces and the population average (calcium and red static) from the same traces as previously shown

in Figure 5a, e (Supplementary Figure 8g, h). Both population average and single-cell traces of $\text{dHPC}^{\rightarrow\text{NAc}}$ neurons suggest that the weak fluorescent fluctuations we see in the static channel are unlikely to fully account for the robust calcium responses we observe in our dynamic channel. This is again confirmed when aligning the average calcium and static signal across all experiments to the onset of appetitive licking (Supplementary Figure 8i). While we observe a significant increase in the static channel's signal for both neuronal populations (Supplementary Figure 8j), the observed difference in dynamics suggests the calcium dynamics could not simply be explained by the differences we see in our static channel.

3. To further rule out that static shifts in our signal are responsible for the observed lick modulation and as per reviewer's suggestion (1c), **we accounted for any residual changes in static signal by two methods:** 1) subtraction and 2) Two-channel Motion Artifact Correction (TMAC) (Creamer et al., 2022, PLoS CompBio):
 - We subtracted each neuron's denoised z-scored calcium activity with its z-scored red static signal. As can be seen in Supplementary Figure 8i, this subtraction did not lead to a decrease in lick-triggered calcium response, and resulted in a corrected cleaned-up calcium trace response that was still significantly modulated by appetitive licking (Supplementary Figure 8i, k).
 - Because subtraction methods assume simple linear relationships, we also used a novel tool from Jonathan Pillow's group (TMAC) that aims at correcting static signals from a dynamic calcium signal using generative modelling and Bayesian inference. We used this tool to correct our calcium signals and, again, found that the calcium responses thus corrected were still significantly modulated by appetitive licking (Supplementary Figure 8i, l).
 - We also reasoned that if the components of the lick modulation observed in the static channel originated from the same underlying motion artifacts and were therefore responsible for the observed lick-triggered calcium responses, we would expect to see a higher correlation of calcium activity with their respective red static channel. We thus calculated Pearson's correlation r values for each neuron but found no significant difference between $\text{dHPC}^{\rightarrow\text{NAc}}$ and dHPC^- populations (Supplementary Figure 8m).
 - Furthermore, we reasoned that if correlations between calcium and static signals were the cause for our observed appetitive lick modulation, we would expect a significant relationship between calcium~static correlations and the magnitude of appetitive lick calcium responses, particularly for the $\text{dHPC}^{\rightarrow\text{NAc}}$ population. Supplementary Figure 8n shows that this is not the case; indeed, there is only a weak negative relationship for dHPC^- neurons ($r = -0.05$) and no significant relationship for $\text{dHPC}^{\rightarrow\text{NAc}}$ neurons ($r = 0.03$).

This suggests to us that, indeed, the weak static modulation we see around lick onset, is unlikely to account for the lick-modulated calcium responses we observe.

4. As per reviewer's suggestion (1c), we also **estimated residual motion frame-by-frame using template matching computer vision algorithms**. Specifically, we calculated a template by averaging across frames 500-1500 for each recording, then calculated (1) normalised correlation values, (2) normalised correlation coefficients, and (3) normalised squared differences between each frame and this template. To illustrate the efficacy of this approach to detect residual motion, we identified a brief episode in one of our recordings where residual motion is visible around frame 1599 (Supplementary Figure 8o). This is reliably being picked up by in-/decreases in all three of our template matching measures

(Supplementary Figure 8p). To see if there is consistent residual motion around the time of appetitive lick onsets, we aligned all three motion measures with the times of appetitive licking for each experiment. When comparing the same pre-/post lick values as for our calcium analysis, we found no significant differences in any of these three measures (Supplementary Figure 8q, r). This suggests that the effects of residual motion around the time of lick onsets are negligible and cannot explain the robust modulation we observe in a subset of neurons.

5. In response to the reviewer's question about the **relative anatomical location of dHPC^{→NAc} neurons**, we have now included additional data showing the distribution of all imaged neurons relative to their respective FOVs (Supplementary Figure 3g, h). We find that dHPC^{→NAc} neurons are found more frequently towards the medial side of the FOV. To see if this region is also more prone to motion artefacts, as suggested by the reviewer, we used dense optical flow algorithms (see Giovannucci et al., 2019, eLife) to estimate any residual xy motion across the FOV per time point. When looking at the resulting average optical flow magnitude along the anatomical axes, we do indeed find significant differences across the FOV, but these are mostly confined towards the lateral and anterior-posterior FOV edges. Notably, the central and medial parts of the FOV (overrepresented by dHPC^{→NAc} neurons) show comparatively little residual motion (Supplementary Figure 8f). These data suggest that, based on their xy positions in the FOV, dHPC^{→NAc} are in fact less likely to encounter residual motion artefacts.
6. While we concur with the reviewer that average calcium z-values around lick onset are relatively small (as seen in Figure 5a-c), we believe this is mostly due to the fact that the effects are largely carried by a subpopulation with consistent lick-associated calcium responses. This is supported by statistically classifying neurons as lick-excited or lick-inhibited (Figure 5d-i), revealing only a minority of about 7% in the dHPC^{→NAc} population (33/444, Figure 5h) that are reliably positively lick-modulated (compared to 4% in the dHPC⁻ population). Based on this subset, averaging calcium responses to the onset of appetitive licking, shows a subpopulation average response of >0.6 SD, which is higher or similar in amplitude compared to related studies (cp. Jarzebowski et al., 2022, CurrentBio, Fig. 3). On a single-cell, single-action level, calcium responses were seen as high as 6 SD (Figure 5e). We would also like to note that we do **not** think that most of these neurons code exclusively for appetitive licking, but our conjunctive-coding analysis (Figure 7) shows most lick-modulated neurons, especially in the dHPC^{→NAc} population, are also modulated by speed and position, again decreasing the relative z-score of calcium responses associated with licking.
7. We have previously employed **three largely independent lines of analysis** that converge on our conclusion of lick-modulated calcium activity:
 1. We took the population average of calcium activity time-locked to the onset of appetitive licking and found a small but significant increase in calcium levels specific to the dHPC^{→NAc} population (Figure 5a-c).
 2. On a single-cell level, we compared calcium levels before and after each lick onset using a paired t-test. This identified approx. 200 neurons (out of >5000 total) that were reliably lick-excited, with a significantly higher proportion in the dHPC^{→NAc} population (Figure 5d-i).
 3. We generated a generalized linear model (GLM) for each neuron and shuffled our true licking traces randomly in time to understand for which neurons calcium activity was reliably less well-predicted compared to the true lick trace (Figure 7). This

analysis, again, suggested that approx. 10% of our neurons are lick-modulated, with a higher proportion in the dHPC^{→NAc} population.

While we note that all three lines of analysis ultimately rely on the calcium signal we recorded, we believe that the convergence of these three different methods suggests a small but effective relationship of a subset of dHPC neurons with appetitive licking.

8. To reduce the potential for motion artefacts affecting our calcium signals, we have previously processed our calcium imaging data in line with what we consider best practice, informed by frequent contact with several developers of the calcium imaging analysis toolbox CalmAn (Giovannucci et al., 2019, eLife). Specifically, we have employed **piecewise non-rigid motion correction** based on the Python implementation of the NoRMCorre algorithm that uses patch-based subpixel registration (Pnevmatikakis & Giovannucci, 2017, Journal of Neuroscience Methods) on our red static channel, minimising the potential for FOV subregions to harbour more residual motion than others. As can be seen in our new Supplementary Figure 8 and our new sample video (see point 9), this method allows us to almost completely remove motion artefacts. In a second step, we then used the **CNMF** algorithm (Pnevmatikakis et al., 2016, Neuron) to simultaneously denoise, deconvolve and demix our cellular calcium signals. In contrast to more traditional ROI averaging-based methods, this modelling-based approach additionally allowed us to filter out small changes in fluorescence that were not associated with true calcium activity.
9. Finally, we have created and uploaded a **new sample video** based on motion-corrected (rolling average of 10 frames) but otherwise raw imaging data of the same recording as shown in Figure 5, highlighting lick-excited dHPC neurons. This video demonstrates two points: 1) there is little to no residual motion, in particular around lick onset, and 2) there are neurons whose calcium activity is reliably increased by appetitive licking but not consummatory licking (as occurs towards the end of the video).

Overall, we believe the evidence strongly favours our original interpretation of the observed calcium modulation resulting from a dHPC^{→NAc}-overrepresented population of neurons that are positively lick-modulated. For this, we have gone through great lengths to ensure we obtained the cleanest possible calcium signals and deployed various analysis techniques that converge on these findings. The additional analyses we performed in Supplementary Figure 8 confirm that residual motion is an unlikely alternative explanation and, in our view, render additional GFP-based experiments unnecessary. We presume such lick-modulated neurons may not have been picked up in previous studies as most lick-related hippocampal recordings likely did not sufficiently dissociate between appetitive and consummatory licking, the latter of which shows a reduced calcium signal in our data (Supplementary Figure 7).

2. As a less major point, do the neural findings especially in Fig. 2 hold up within animal, at least trend-wise? It would be important to know that the result of higher spatial coding in the dHPC->NAc cells is not driven by one or two animals having really good labeling and/or better place cells, or greater labeling in dCA1 compared to subiculum. Could the authors show panels 2d-h (and maybe 3f and 4h) per animal, in a supplemental figure? I think it is ok if there aren't enough cells to reach significance in each animal, as long as the trend is similar across animals.

We have now included all individual data points and the average values per animal for the requested panels:

- Figure 2f-k: Supplementary Figure 5a-f

- Figure 3f: Supplementary Figure 5p
- Figure 4d, h: Supplementary Figure 6a, b

We have also performed additional statistics to determine and account for between-animal variance: For the proportion of place cells (Figure 2f), we have performed a chi-square test on the proportion of place cells across projection subpopulations per mouse and found no significant difference. We have added this in the text as follows (lines 162-163): “These proportions were not different between mice ($\chi^2(4) = 0.24, P = 0.99$).”

We also compared the relative proportions of speed-excited and speed-inhibited neurons across subpopulations per mouse and found no significant difference, see lines 235-236: “These proportions were not different between mice (speed-excited: $\chi^2(4) = 0.34, P = 0.99$; speed-inhibited: $\chi^2(5) = 0.63, P = 0.99$).”

Regarding spatial metrics reported in Figure 2g-k, we have additionally performed **linear mixed modelling** with projection identity as fixed effect and subject identity as random effect. We find that results hold for spatial information rate ($\beta=0.364, z=4.20, P<0.001, CI=0.19-0.53$) and sparsity ($\beta=0.056, z=5.07, P<0.001, CI=0.034-0.078$), while reliability ($\beta=-0.009, z=-0.391, P=0.696, CI=-0.053-0.035$), stability ($\beta=0.011, z=0.739, P=0.460, CI=-0.018-0.039$), and delta place field activity ($\beta=1.178, z=1.446, P<0.148, CI=-0.418-2.77$) failed to reach significance.

3. The optogenetic results seem more in line with the findings of Britt et al. 2012 to me than direct evidence for a dHPC->NAc projection role in spatially-guided appetitive licking per se. That is, Britt et al. found that axonal stimulation of pretty much any glutamatergic input to the NAc can elicit “appetitive” behavior, i.e. CPP in their task. The mouth movement elicited here is consistent with that, more so than “a driving force in the generation of task-relevant behavior” (line 230) as the authors claim. Perhaps if the dHPC->NAc stimulation had elicited licking on a novel treadmill after being trained to lick in a hidden reward zone on a previous treadmill, this would suggest that a place-reward association was being activated. I don’t think the authors need to do this experiment, but I do suggest they adjust their discussion of the optogenetic results.

We agree with the reviewer that our data do not demonstrate the “activation of a place-reward association”, which we also did not claim to show. Instead, our imaging results suggest that dHPC conveys a rich set of task-relevant information to NAc including allocentric spatial and egocentric action information, which directs NAc activity to engage in appetitive licking in the correct spatial location. The mouth movement we find upon stimulation suggests that this pathway is indeed capable of driving such behaviours.

In further support of this model, we have now included **three pieces of additional optogenetic data**: 1) We trained head-fixed food-restricted mice to lick for reward in a designated reward zone in which dHPC→NAc terminals were also optogenetically activated. We found increased appetitive licking immediately preceding the reward zone on the second day (new Supplementary Figure 2g). 2) We trained mice for four days to lick in a designated reward zone before optogenetically silencing dHPC→NAc neurons using ArchT around the reward zone on the fifth day. We found a significant reduction in appetitive licking near the reward zone onset that was specific to ArchT-expressing animals but not control animals (new Figure 2e-i). 3). We retrained ArchT-expressing and control mice to lick for reward in a new reward zone around which mice received light stimulation from the first day of learning on. We saw that ArchT-expressing mice showed significantly impaired learning of this new reward zone indicated by reduced appetitive licking by day 3 (new Figure 2j).

Altogether, our optogenetic data thus (1) demonstrate that dorsal (not only ventral) hippocampal projections into NAc can elicit appetitive behaviours, (2) show that such appetitive behaviours extend beyond mere spatial approach behaviours (as during CPP) into the realm of acute orofacial movements, (3) that such activation in combination with reward can improve spatial reward memories to increase anticipatory licking, and (4) that this pathway is necessary for the formation and expression of spatial reward memories. Overall, our findings thus point towards the dHPC→NAc projection as causally relevant in driving spatial memory-guided appetitive behaviours. As such, we do not see our optogenetic data in contrast to those of Britt et al. (2012), but rather as an extension that incorporates a bidirectional influence on head-fixed spatial reward learning and direct effects on orofacial and locomotor movements. Besides including these new experimental results in our revised manuscript, we have now also adjusted our Discussion as follows:

(lines 339-352): “We find that HPC→NAc projections are not only required for spatial memory tasks such as the water maze (Ito et al., 2008) or conditioned place preference (Britt et al., 2012; Trouche et al., 2019) but also for the acquisition and retrieval of learned reward locations in a head-fixed linear navigation task. Specifically, we found that optogenetic excitation of this pathway during learning enhances the animal’s ability to memorize the reward zone location, as indicated by a higher rate of anticipatory licking already on the second day of training. Such facilitation of learning complements previous studies demonstrating the induction of spatial preference by optogenetic stimulation (Britt et al., 2012; LeGates et al., 2018) and learning-related increases in HPC→NAc coupling (Sjulson et al., 2018). Conversely, inhibition specifically decreased the amount of licking directly preceding the reward zone without affecting the overall success rate. This suggests that information from dHPC is required for the NAc to engage in appetitive behaviors guided by learned reward locations and stands in contrast to BLA afferents that are required for appetitive behaviors triggered by learned cues (Ambroggi et al., 2008; Stuber et al., 2011). Taken together, these findings highlight a dual role for the dHPC→NAc pathway in both learning and retrieval of spatial memories.”

Minor:

4. Could you clarify in the methods and main text whether there were 2 groups of mice with different reward zone locations? Upon first read it sounded like all the mice had the same location at the same cue transition, and I didn’t realize that this might not have been the case until Fig. S1. It actually strengthens the results if there are 2 cohorts with different locations, as this suggests the properties of dHPC→NAc neurons are specific to a hidden reward zone and not to a particular cue combination; but in the main figures, the reward zone is always shown at the cohort 2 location (before the vertical stripes). Are the data in the main text only from cohort 2? Were the 18 mice included in Fig. S1 and Fig. 1 only used for behavior, and not imaging? Please clarify which results come from cohort 1, 2, or both.

We thank the reviewer for highlighting this potential point of confusion. Indeed, we trained a total of 18 mice on the behavioural task, 9 of which were trained with the reward zone in the centre, 9 with the reward zone at the end of the belt. 6 of the latter were later used for calcium imaging. To avoid confusion, we have updated our text to include further clarifications in lines 87-91: “To investigate coding of spatial information and goal-directed behavior in dHPC→NAc neurons, we trained three independent cohorts of food-restricted mice (n = 18) [...] 30 cm long fixed reward zone (located in the “center” of the track for one cohort, at the “end” for the others)”

5. Can you show the distribution of speed preferences for the speed-excited and speed-inhibited neurons in each subpopulation? Specifically, I'm wondering whether all dHPC->NAc neurons are active when the mouse is basically stopped, or are some active at other intermediate speeds?

We have now included an additional supplementary figure (new Supplementary Figure 6c, d) that shows the distribution of speed preference bins (each neuron's velocity bin with the highest average denoised calcium events) for speed-excited (6c) and speed-inhibited (6d) neurons for both dHPC- and dHPC->NAc neurons. For full transparency, we have also included a velocity bin of speeds below 2cm/s (fully stopped or walking backwards). As the reviewer will appreciate, most speed-inhibited neurons are most strongly active at speeds of 2-5cm/s, with similar densities between dHPC- and dHPC->NAc neurons. We also confirmed this with a multi-level chi-square test that indicated no significant proportional differences between dHPC- and dHPC->NAc neurons for speed-inhibited (chi-square(4)=4.80, p=0.31) and speed-excited (chi-square(4)=4.74, p=0.32) neurons.

6. Fig. 5 needs vertical scale bars for dF/F (or z-scored deconvolved activity, whichever is shown) in all panels with example traces. It would be nice to see dF/F or z-score instead of a.u. in Fig. 4 as well.

We have added the z-scored deconvolved activity scale bars in Figures 4 and 5.

7. Fig. 5d: Please add anatomical axes for AP/ML.

We have added these.

8. I found the description of GLM results on lines 265-286 a bit confusing. On line 277-279 you say "the average drop in variance explained to the full model was significantly stronger in dHPC->NAc neurons compared to dHPC- for position and velocity but not licking." But subsequently, you seem to claim that conjunctive coding of all 3 variables is significant, whereas it sounds like the modeling result only indicates significance for velocity and position? If the lick coding analysis stands after addressing comment #1, I would appreciate clarification on this point.

In our description of the GLM's results, we have reported two different measures: one is the average ΔR^2 value obtained from comparing the full model's R^2 with the average of the 3x100 shuffled-feature models. Here, on average, ΔR^2 values are no different between dHPC- and dHPC->NAc populations for licking. This measure takes into account the magnitude of the difference each feature makes for the model. The other measure aims to detect significant contributions of one of the three parameters with a threshold of 95%, i.e., if >95% of shuffled models perform worse than the full model, that feature is taken to be a significant contributor to that neuron's activity. In Figure 7b-e, we quantified the proportions of neurons for which each of the three features (or combinations) were significant in this way. The proportion of neurons in which "licking" makes a significant contribution to explaining those neurons' calcium activity is indeed greater for dHPC->NAc compared to dHPC- neurons. To make this point clearer, we have added an additional sentence to help readers follow our

analysis (lines 309-310): For this, we first investigated the average difference of the shuffled models compared to the full model for each neuron (\$\Delta R^2\$ ).

9. In the introduction on line 56-57, you say that the NAc receives “strong” projections from both the dorsal and ventral subiculum, but as far I know this is not the consensus in the field. Ventral subiculum and CA1 send strong projections, but the dorsal subiculum projection is weaker anatomically and the direction innervation from dCA1 wasn’t confirmed until a few years ago with the Trouche et al. 2019 paper, to my knowledge. It would be worth emphasizing this contribution of the Trouche paper since you are putatively recording from the same direct-projecting CA1 cells, and therefore your study adds a nice confirmation that this direct projection is contributing to the computation of reward locations.

We have toned down our introductory sentence and added reference to the Trouche paper, see lines 56-57: “The NAc receives projections from both dorsal and ventral CA1 and subiculum (Kelley and Domesick, 1982; Groenewegen et al., 1987; Trouche et al., 2019) [...]”. In addition, we have now added a new paragraph in our discussion that, we believe, gives due credit to the Trouche et al. (2019) paper for describing the dCA1→NAc connection, see lines 373-378: “[...] much of the past literature on spatial reward learning has focused on the prominent NAc afferents from vHPC (Britt et al., 2012; LeGates et al., 2018; Reed et al., 2018; Zhou et al., 2019; Wee et al., 2023). Only recently have the anatomically weaker dHPC afferents to the NAc started to receive attention. Trouche et al. (2019) clearly demonstrated the existence of such dCA1→NAc projections at ultrastructural level and found these to be necessary for CPP. “

REVIEWER COMMENTS

Reviewer #1 (Remarks to the Author):

The authors have fully addressed all my comments in a comprehensive revision incorporating substantial new experiments, re-analysis, and additional analyses. This is an exciting manuscript that stands to make an important contribution to the literature.

Reviewer #2 (Remarks to the Author):

I would like to thank the authors for taking all reviewer comments seriously and thoroughly addressing all concerns. This is a great paper that I'm sure will get a lot of attention. Congratulations!

Reviewer #3 (Remarks to the Author):

The authors have made substantial improvements to an already interesting manuscript, and I commend them for addressing many of the critiques from myself and the other reviewers. The added optogenetic experiments are quite compelling and strengthen the findings, particularly the inhibition experiment now shown in Figure 1, demonstrating an impaired ability to learn a new reward location with the dHPC-to-NAc projection inactivated. Despite my enthusiasm for the work and admiration for the authors' efforts, however, my main concern about the claimed lick coding was not sufficiently addressed. I elaborate further below and include a few more minor points of confusion raised by some of the added work.

Major:

The authors have taken several reasonable analytical approaches to address the possibility that an artefact in the calcium signal at the time of licking explains the lick coding they detect in hippocampus. Unfortunately, however, I do not think these approaches sufficiently control for this possibility. While I agree it cannot be ruled out with absolute certainty, data from control mice expressing static GFP would rule it out with greater certainty than the current methods.

To provide a point of comparison for the authors' investigation, I have provided the editor with a figure of my own unpublished data to be sent separately to the authors. This figure illustrates what the lick artefact looks like in the green channel using static GFP in dorsal hippocampus in a similar type of experiment. Note that there may be slight behavioral differences from the behavior used by the authors (and differences in exact anatomical imaging location and method of lick detection). Despite these differences, please note the similarities in shape and z-score magnitude (related to point #6) of this lick-triggered GFP artefact compared to the lick-triggered calcium signal in the authors' data.

Below, I am referencing the authors' numbered responses to my original point to further clarify my specific arguments. Let me also preface this critique by saying, again, that the rest of the paper could stand on its own without such a focus on the lick coding; in my view, there is no need to show that dHPC-to-NAc cells code licking per se – the optogenetic activation could (speculatively) elicit spatial memory recall of the rewarded location, which could then initiate appetitive licking downstream, without the dHPC neurons needing to code the licking action directly (I am not suggesting the authors make this claim, but it is an intriguing possibility that their work supports).

1. "literature describing lick-modulated hippocampal activity" – Thank you for pointing me to these papers. The authors are correct that there is some precedent for lick correlates in the hippocampus, though I believe further work is required to assess whether these correlates reflect licking per se.

2. "While we observe a significant increase in the static channel's signal for both neuronal populations (Supplementary Figure 8j), the observed difference in dynamics suggests the calcium dynamics could not simply be explained by the differences we see in our static channel" – The authors have demonstrated a significant lick-related artefact in the static red channel, which was exactly my concern, as it indicates such an artefact is detectable in their dataset. Unfortunately, one cannot conclude that it does not account for the GCaMP signal without knowing whether the artefact would look the same in the green channel (by using GFP).

2a. As a minor clarification, is the red static signal shown in Fig. S8 deconvolved to match processing of the GCaMP signal? Or is $\Delta F/F$ used for both red and green? It was not clear without a methods section on this control. I think either is fine as long as processing is consistent, though $\Delta F/F$ would probably make more sense here as there are no "events" in the static signal.

3. "...we accounted for any residual changes in static signal by two methods" – These are both reasonable methods to correct for potential artefact contamination. However, in order to verify that either method is appropriate for the authors' dataset, they must perform a GFP control. As stated in the Creamer et al. paper on the TMAC method: "Crucially, whatever the source of these artifacts, as long as they affect the red and green channel equally, two-channel motion correction should be able to account for and remove them." The authors have not assessed whether the artefact at the time of appetitive licking affects the red and green channel equally, because they have not measured it using GFP as done

in Creamer et al. and in other works that use a similar motion correction algorithm (e.g. Hallinen et al. 2021, doi: 10.7554/eLife.66135). Depending on the amount of motion relative to the imaging plane, motion-induced fluorescence changes in short and long wavelengths can have different amplitudes (Tai et al. 2004, <https://journals.physiology.org/doi/full/10.1152/ajpheart.00574.2003>, Fig. 3).

3a. In Fig. S8n, even though there is no strong correlation between the calcium vs. static correlation and Δ pre-post at the population level, is there a positive correlation in the 7% of neurons that are significantly lick-excited? See point #6.

4. "...we also estimated residual motion frame-by-frame using template matching computer vision algorithms" – Unless I misunderstood, all of the approaches taken here and in response points #8 and 9 account for xy motion, not z-motion. I have no doubt that the non-rigid motion correction is sufficient to deal with xy motion, and the videos provided do look very stable in xy. Z-motion, however, is more difficult to detect (Harris et al. 2016, doi: 10.1038/nn.4365; Ryan et al. 2020, doi: 10.1113/JP278957), and is likely to induce fluorescence changes relative to the imaging plane (Flores-Valle & Seelig 2022, doi: 10.1364/BOE.445775). My understanding is that this is what the TMAC algorithm controls for, as long as z-motion affects the green and red fluorescence equally. Z-motion is best accounted for by comparing each frame to the planes of a z-stack taken at the FOV, by using multi-plane imaging (Flores-Valle & Seelig 2022), or by using an ROI-shape based approach (e.g. the method suggested in the Gauthier & Tank 2018 methods, or Ryan et al. 2020). Without a z-stack or these other methods, z-motion is difficult to estimate, thus necessitating the GFP control.

5. The anatomical quantification is helpful, but again it seems xy optical flow has been estimated here as opposed to z-motion.

6. "While we concur with the reviewer that average calcium z-values around lick onset are relatively small (as seen in Figure 5a-c), we believe this is mostly due to the fact that the effects are largely carried by a subpopulation with consistent lick-associated calcium responses." – Thank you for clarifying this; I did not appreciate before that the significantly lick-excited neurons were such a small percentage (7% in dHPC \rightarrow NAC and 4% in dHPC-). This indicates that it is not appropriate to average over the whole population as shown in Fig. 5a-c. Furthermore, 5f-g is still confusing – if the average z-scored traces in 5f-g are the significantly excited or inhibited neurons as stated in the figure legend, why are the z-score values still so small, if values reached up to 6 sd on the single neuron level? Please see the provided figure pdf for an example of z-score magnitude in a GFP artefact (it is on the order of \sim 0.2-0.4 sd, though sometimes greater).

6a. Importantly, do the significantly excited and inhibited neurons comprise larger fractions than one would expect by chance (i.e. as the possible result of an artefact?)

6b. I think an appropriate way to identify neurons that truly encode licking would be to first confirm what the chance detection level is using \sim 2 GFP control mice (i.e. what fraction of GFP ROIs are "significantly" lick-excited/inhibited) and determine what the magnitude of the lick-induced artefact is in

the green channel in the authors' dataset. Next, classify neurons in the GCaMP dataset that have significant lick-bout-triggered responses (using a non-parametric test of post vs. pre) AND have responses that exceed the threshold defined by the lick-bout-triggered GFP magnitude. This may limit the significantly excited cells to those with large and reliable responses like the 6 sd example in Fig. 5e.

6c. I recommend then presenting these highly excited (or reliably suppressed) neurons using their maximum-normalized activity rather than a z-score – this will avoid the negative-going signal for the “lick-inhibited” neurons and make it clear that these neurons are just not firing at lick times, as shown in Fig. 5e cell 11, as opposed to decreasing their activity “below baseline” (line 253). Deconvolved activity cannot fall below baseline since baseline of deconvolved events should be zero.

7. The GLM does indeed complement your other analyses, but I would still be curious to know how much licking modulation it detects from a GFP control.

8. See response #4 about xy vs. z motion.

9. I appreciate the added video. A few of these cells, such as cell 10, do seem to have very strong lick-correlated transients; I would be curious whether these cells produce some of the largest amplitude signal, exceeding any possible artefact contribution.

10. Finally, re: appetitive vs. consummatory licking – I agree with the authors that it is important to separate these as they have done. It is also important to consider that they are associated with different movement dynamics; appetitive licking typically occurs while the animal is slowing down but still walking (see provided figure), while consummatory licking tends to occur once the animal is closer to fully stopped. This observed difference in movement dynamics seems to concur with the videos provided by the authors. When I have isolated consummatory licking in my own data, I have likewise observed a decrease in static GFP fluorescence in the ROI population average as the authors observe with GCaMP, and this change is distinct from what is seen during appetitive licking (see provided figure).

10a. As a separate example of how GFP artefacts can be elicited by certain locomotor dynamics, please see Fig. S2D of Yogesh & Keller 2023, eLife (<https://doi.org/10.7554/eLife.89986.1>)

10b. In conclusion, I strongly recommend that the authors collect a small amount of data from ~2 mice expressing static GFP to verify their findings. These mice would not have to perform the entire task – examining appetitive vs. consummatory licking in the first several days of training would probably be sufficient, as long as the behavior is similar to that of the GCaMP animals. If there is a substantial lick-induced signal in the GFP controls, lick coding in the GCaMP dataset should only be considered significant for neurons that exceed a threshold defined by the GFP signal. If this reduces the fraction of lick-excited neurons such that there is no longer a statistically significant difference between the dHPC-to-NAc and dHPC- populations, the authors should report this. If either population includes neurons with

robust appetitive lick coding beyond the artefact, this would itself be an interesting and novel finding, going beyond what has been reported in the literature previously.

Minor:

1. Re: original point #2 regarding individual animals: I thank the authors for illustrating data points for each animal. In the newly added Supplemental Fig. S5, however, I only see 5 points plotted in (a), although I thought 6 mice were imaged. Mouse 209 seems to be missing? In addition, there are 5 points for dHPC- and 3 for dHPC-to-NAc in (b-f). Were some mice excluded for having zero place cells in the dHPC-to-NAc population as suggested in (a)? Were these same mice excluded from all subsequent analyses? If so, the n of mice that were actually included in each analysis should be reported in each figure legend.

1a. Relatedly, the linear mixed effects model reported in the rebuttal is helpful – those results should ideally also be reported in the supplemental. It suggests that spatial information and sparsity are the most robustly enhanced properties of the dHPC-to-NAc cells.

2. Clarifications about new inhibition results:

2a. In the response to reviewer 1, the authors note: “Interestingly, we found no effect of our optogenetic inhibition during reward consumption”. Where is this shown? Supplemental Fig. 2a only shows number of rewarded laps, not the effect of inhibition on lick rate in the reward zone, as the statement implies. This seems an important result to include for the readers.

2b. What distance is defined as “shortly before the reward zone” for the quantification in Fig. 1h? Similarly in Supplemental Fig. S2G, how is “near reward” defined? Is this the same distance as used for “place fields near reward zone”?

2c. The authors interpret the inhibition experiment as “suggesting a lack of spatial precision to guide appetitive licking” (line 106). However, the reported effect is a reduction in peak licking rate, correct? This could alternatively be interpreted as lower confidence in the spatial location or lower willingness to expend effort, whereas an extension of the “ramp” of anticipatory licking to earlier positions on the belt would more clearly imply lower spatial precision. Perhaps the authors could clarify what they mean by “lack of spatial precision”, and/or try to quantify the shape of the ramp – one possibility would be to use a KS test on the distribution of pre-reward licking positions in controls vs. ArchT animals.

3. Please consider increasing the font sizes in Supplemental Figs. S5 and S8.

Reviewer #1 (Remarks to the Author):

The authors have fully addressed all my comments in a comprehensive revision incorporating substantial new experiments, re-analysis, and additional analyses. This is an exciting manuscript that stands to make an important contribution to the literature.

We are grateful for the reviewer's help in improving the manuscript and are glad to see the reviewer's excitement about our manuscript.

Reviewer #2 (Remarks to the Author):

I would like to thank the authors for taking all reviewer comments seriously and thoroughly addressing all concerns. This is a great paper that I'm sure will get a lot of attention. Congratulations!

We want to reciprocate extending our thanks to the reviewer for their help in improving the manuscript. We are happy to hear about the reviewer's enthusiasm for the current version.

Reviewer #3 (Remarks to the Author):

The authors have made substantial improvements to an already interesting manuscript, and I commend them for addressing many of the critiques from myself and the other reviewers. The added optogenetic experiments are quite compelling and strengthen the findings, particularly the inhibition experiment now shown in Figure 1, demonstrating an impaired ability to learn a new reward location with the dHPC-to-NAc projection inactivated. Despite my enthusiasm for the work and admiration for the authors' efforts, however, my main concern about the claimed lick coding was not sufficiently addressed. I elaborate further below and include a few more minor points of confusion raised by some of the added work.

We are glad to find the reviewer appreciates our extensive revision efforts and continued enthusiasm about our work. We do not fully share the reviewer's concerns about the lick coding though, as elaborated below.

Major:

The authors have taken several reasonable analytical approaches to address the possibility that an artefact in the calcium signal at the time of licking explains the lick coding they detect in hippocampus. Unfortunately, however, I do not think these approaches sufficiently control for this possibility. While I agree it cannot be ruled out with absolute certainty, data from control mice expressing static GFP would rule it out with greater certainty than the current methods.

To provide a point of comparison for the authors' investigation, I have provided the editor with a figure of my own unpublished data to be sent separately to the authors. This figure illustrates what the lick artefact looks like in the green channel using static GFP in dorsal hippocampus in a similar type of experiment. Note that there may be slight behavioral differences from the behavior used by the authors (and differences in exact anatomical imaging location and method of lick detection). Despite these differences, please note the

similarities in shape and z-score magnitude (related to point #6) of this lick-triggered GFP artefact compared to the lick-triggered calcium signal in the authors' data.

We commend the reviewer for putting in the effort of preparing this figure. We agree that some of the traces provided resemble those we present in our data. To clarify similarities and differences, we have prepared a new **Figure R1** that shows z-scored deconvolved and raw $\Delta F/F_0$ signals of GCaMP and mCherry channels for different cell populations. We have added the raw $\Delta F/F_0$ to demonstrate the absolute magnitude of the signals we collected. A few notes to take from this data and the comparison with the data provided by the reviewer:

1. While there is a small positive deflection visible in the static mCherry channel for lick-excited neurons (**Figure R1a**), it is much smaller compared to the respective calcium signal ($z \sim 0.1$ vs $z \sim 0.6$), it is much smaller compared to the reviewer's panel (b) ($z \sim 0.1$ vs $z \sim 0.4$), and, importantly, this small artefact also appears for lick-inhibited neurons (**Figure R1c**) and for consummatory licking (**Figure R1e**), the latter two of which correspond to calcium *decreases*. This suggests that motion-related artefacts in our data are exceedingly small (as can also be appreciated from the video we provided earlier) and result, on average, in minor positive deflections in the static mCherry signal, but are not of the appropriate magnitude or pattern across conditions to explain the calcium changes we observe in our data.
2. In **Figure R1b, d, f, h**, we are now also showing the raw $\Delta F/F_0$ traces we extracted from both GCaMP and mCherry channels to provide the reviewer with an intuition of the scale of signals recorded. While we have mostly relied on z-scores to allow a comparison of calcium signals from cells with slightly different GCaMP expression levels, at different depths, imaged with slightly different parameters, we note that z-scoring on low SNR traces can inflate any spurious signals that may be inherent to the data. We believe this is the case when z-scoring from a static channel (be it GFP or mCherry) because the traces do not contain any notable transients. As such, standard deviations (SD) will accordingly be very small so that dividing any transients by such small SD values will result in amplified signals. This is of course no problem if there is truly no signal contained in the traces, but will boost any potential behaviourally related signal such as event-specific movement artefacts. To illustrate this further, we have taken a 1500-frame sample snippet with calcium transients from our imaging data and compared these two different normalisation methods to a 5x300 frame repeat of this snippet containing no transients (**Figure R2**). The data clearly illustrate how noisy data containing no calcium transients becomes amplified to match the amplitude of real calcium transients. We thus caution against comparing z-score values from data with calcium transients to those without.
3. We would also like to highlight that, as the reviewer notes, it is very difficult to compare their data to ours as behavior, imaging location, and lick detection will all be different. Likewise, we caution against trying to compare our current data with newly collected GFP-only data, as suggested by the proposed new experiments from the reviewer.

Figure R1: Calcium (red/green) and static (grey) signals from dHPC→NAc (red) and dHPC– (green) neurons around times of appetitive lick onset (a-d) and consummatory licking (e-h). Traces a, c, e, g represent z-scored deconvolved signals (using OASIS), while b, d, f, h represent raw $\Delta F/F_0$ signals (F_0 defined as 8th percentile of trace). Excited/inhibited neurons were defined as described in Methods: Wilcoxon signed-rank tests for each neuron’s average signal -2:0 s prior to event were compared to the average signal 0:2 s after event. Significant and positively modulated neurons are termed “excited”; significant and negatively modulated neurons are termed “inhibited”.

Figure R2: Sample normalisation procedure of a trace containing calcium transients (purple) and containing only noisy background (orange). Orange trace is a 5x repeat of the first 300 frames of the purple trace, containing no detectable calcium transients. Z-scoring was performed by subtracting each trace's mean before dividing by its standard deviation (SD purple = 0.087; SD orange = 0.013). F0 was calculated as each trace's 8th percentile. Note that z-scoring seems to amplify noisy signals.

Below, I am referencing the authors' numbered responses to my original point to further clarify my specific arguments. Let me also preface this critique by saying, again, that the rest of the paper could stand on its own without such a focus on the lick coding; in my view, there is no need to show that dHPC-to-NAc cells code licking per se – the optogenetic activation could (speculatively) elicit spatial memory recall of the rewarded location, which could then initiate appetitive licking downstream, without the dHPC neurons needing to code the licking action directly (I am not suggesting the authors make this claim, but it is an intriguing possibility that their work supports).

We agree with the reviewer that dHPC^{→NAc} cells would not necessarily have to be lick coding, yet it is a result we obtained from our data and that we believe is sufficiently backed up by the quality of data collected and our subsequent analysis, the latter of which has been greatly improved by the reviewer's suggestions so far. As we understand the reviewer's concerns though, we have decided to move previous Figure 5a-c into Supplementary Figure 7d-f (see #6) and added the following caveat in our Discussion:

“One general caveat for in vivo two-photon calcium imaging studies pertains to the potential of behaviorally induced motion artefacts along the imaging plane's xy and z axes to distort inferred calcium signals (Harris et al., 2016; Ryan et al., 2020). We have controlled for this possibility by comparing signals from our dynamic GCaMP channel to the simultaneously collected static mCherry signal. While we found a small positive inflection upon appetitive licking in our static channel (Fig. S8k, m), we note that this signal is relatively small and occurs in both lick-excited and lick-inhibited populations, making it unlikely to significantly affect our results. After controlling for these static signal fluctuations post hoc (Creamer et al., 2022), the observed lick modulation holds true. Future studies on lick-related calcium dynamics, especially when sampling subcellular neural processes, should beware of such potential confounds.”

1. *“literature describing lick-modulated hippocampal activity” – Thank you for pointing me to these papers. The authors are correct that there is some precedent for lick correlates in the hippocampus, though I believe further work is required to assess whether these correlates reflect licking per se.*

We are glad the reviewer found the literature cited useful and agree that these publications do not conclusively demonstrate lick modulation. We believe our data add to a growing appreciation that task-related behaviours may also be encoded by hippocampal activity (e.g., McNaughton et al., 1983, Experimental brain research; Wiener et al., 1989, JNeurosci; Markus et al., 1995, JNeurosci; Wood et al., 2000, Neuron; Ledergerber et al., 2021, Cell Reports; Sun et al., 2021, NatComms; Karalis & Sirota, 2022, NatComms).

2. “While we observe a significant increase in the static channel’s signal for both neuronal populations (Supplementary Figure 8j), the observed difference in dynamics suggests the calcium dynamics could not simply be explained by the differences we see in our static channel” – The authors have demonstrated a significant lick-related artefact in the static red channel, which was exactly my concern, as it indicates such an artefact is detectable in their dataset. Unfortunately, one cannot conclude that it does not account for the GCaMP signal without knowing whether the artefact would look the same in the green channel (by using GFP). 2a. As a minor clarification, is the red static signal shown in Fig. S8 deconvolved to match processing of the GCaMP signal? Or is $\Delta F/F$ used for both red and green? It was not clear without a methods section on this control. I think either is fine as long as processing is consistent, though $\Delta F/F$ would probably make more sense here as there are no “events” in the static signal.

We thank the reviewer for pointing out these potential incongruencies. Indeed, we have largely relied on deconvolved calcium data throughout our manuscript to obtain cleaner signals (and after visually inspecting each deconvolved trace and raw trace). Since the extraction of the red static traces does not normally involve a deconvolution step, we have not previously performed this. Based on this suggestion and the reviewer’s suggestion in #6 to focus on lick-excited/-inhibited neurons, we have now modified Figure S8 to include both sample $\Delta F/F_0$ traces of dynamic/static channels and with TMAC correction (**Figure S8f**), as well as dynamic/static deconvolved signals (in keeping with the rest of the manuscript) around appetitive lick onsets for lick-excited (**Figure S8j,k**) and lick-inhibited (**Figure S8l,m**) neurons. We have performed the deconvolution using CalmAn’s OASIS algorithm with standard parameters, the same as used on our calcium data. We note that both lick-excited as well as lick-inhibited neurons show slight increases in the red mCherry channel, despite inverse calcium transients. These calcium increases remain significant even after TMAC correction. Given that our red/green channel’s PSF (see **Figure R3** below) is highly comparable across the Z-axis, we believe the TMAC algorithm is well-suited for this kind of correction. We have also updated our methods section to include the methods employed here.

3. “...we accounted for any residual changes in static signal by two methods” – These are both reasonable methods to correct for potential artefact contamination. However, in order to verify that either method is appropriate for the authors’ dataset, they must perform a GFP control. As stated in the Creamer et al. paper on the TMAC method: “Crucially, whatever the source of these artifacts, as long as they affect the red and green channel equally, two-channel motion correction should be able to account for and remove them.” The authors have not assessed whether the artefact at the time of appetitive licking affects the red and green channel equally, because they have not measured it using GFP as done in Creamer et al. and in other works that use a similar motion correction algorithm (e.g. Hallinen et al. 2021, doi: 10.7554/eLife.66135). Depending on the amount of motion relative to the imaging plane, motion-induced fluorescence changes in short and long wavelengths can have different amplitudes (Tai et al. 2004, <https://journals.physiology.org/doi/full/10.1152/ajpheart.00574.2003>, Fig. 3).

We are not aware of any literature that would suggest differences in motion-induced fluorescence changes between green/red channels using *in vivo* two-photon imaging. Tai et al. 2004 use a vastly different setup and biological tissue (single-photon, *in vitro* heart

preparation) that we believe are not a valid comparison. Indeed, Creamer et al. write “as long as they affect the red and green channel equally” without any citations to cases in which that would not be the case. Hallinen et al. 2021 write “We assume that these artifacts are common to both GCaMP and RFP fluorescence, up to a scale factor, because both experience the same motion.” Indeed, most *in vivo* two-photon calcium imaging studies including various behaviours do not include a second channel to control for motion as we did, let alone compare the sensitivity of those channels to Z-motion. We have also measured the PSF along the Z-axis using fluorescent beads on a comparable setup (since the original setup has been moved), and found the PSFs across the red and green channels to be very comparable (see **Figure R3**).

Figure R3: Point-spread-function (PSF) along Z-axis of a setup comparable to the one used in our manuscript, captured from fluorescent beads with simultaneous red/green channel collection (1040 / 920 nm 2P excitation).

3a. *In Fig. S8n, even though there is no strong correlation between the calcium vs. static correlation and Δ pre-post at the population level, is there a positive correlation in the 7% of neurons that are significantly lick-excited? See point #6.*

We have analysed our data as suggested and find no significant correlations of significantly lick-excited neurons, neither for dHPC⁻ ($r(124) = -0.145$, $P = 0.109$) nor dHPC^{→NAc} neurons ($r(28) = 0.239$, $P = 0.220$). We have added this as **Figure S8i** in the supplementary, along with the correlations for lick-inhibited neurons.

4. “...we also estimated residual motion frame-by-frame using template matching computer vision algorithms” – Unless I misunderstood, all of the approaches taken here and in response points #8 and 9 account for *xy* motion, not *z*-motion. I have no doubt that the non-rigid motion correction is sufficient to deal with *xy* motion, and the videos provided do look very stable in *xy*. *Z*-motion, however, is more difficult to detect (Harris et al. 2016, doi: 10.1038/nn.4365; Ryan et al. 2020, doi: 10.1113/JP278957), and is likely to induce fluorescence changes relative to the imaging plane (Flores-Valle & Seelig 2022, doi: 10.1364/BOE.445775). My understanding is that this is what the TMAC algorithm controls for, as long as *z*-motion affects the green and red fluorescence equally. *Z*-motion is best accounted for by comparing each frame to the planes of a *z*-stack taken at the FOV, by using multi-plane imaging (Flores-Valle & Seelig 2022), or by using an ROI-shape based approach (e.g. the method suggested in the Gauthier & Tank 2018 methods, or Ryan et al.

2020). *Without a z-stack or these other methods, z-motion is difficult to estimate, thus necessitating the GFP control.*

We would like to point out that, naturally, template matching algorithms will also pick up z-motion because they compare the similarity between two images, regardless if the shifts occur in xy or represent global differences induced by z-motion. This is exactly what we illustrate in **Figure S9b,c** where a rapid Z-shift in frame 1599 causes a mismatch to preceding and subsequent frames that is being picked up by all three template matching values.

Harris et al. write “*Generally, point spread functions (PSFs) are extended in Z, making motion artifacts resulting from small Z movements less problematic.*”

Ryan et al. write “*In practise, the majority of two-photon imaging studies in awake animals do not correct for z-motion because calcium signals are usually acquired from the soma of neurons, which are ~10 μm in diameter and therefore change intensity relatively little when displaced a few microns.*” To correct for Z-motion to image at the level of synaptic boutons (<1μm), the authors propose to use a static second channel: “*Estimates of z-motion are made from simultaneously obtained images of an inactive anatomical marker.*”, which is comparable to what we have used in our study.

Flores-Valle & Seelig have used multi-plane imaging for online movement correction in *Drosophila*. This is a valid approach that is necessitated by the small structures that are being imaged in the fly brain and the relatively strong motion that may come from brains in tethered fly heads, but such an approach is generally not used for mouse *in vivo* calcium imaging for practical reasons and because the size of cell bodies and the large PSF along Z does not warrant such an effort.

Dombeck et al. (2007, <https://doi.org/10.1016/j.neuron.2007.08.003>) quantified this Z-motion in a similar hippocampal preparation: “*In general, the average out-of-plane (Z) motion was on the order of the axial point spread function radius (2 μm)*”, or Kong et al. (2016, <https://doi.org/10.1364/BOE.7.003686>, Fig. 5) who find movement-related Z-motion displacement not exceeding **1-2 μm** during hippocampal imaging. Dombeck et al. (2010, <https://doi.org/10.1038/nn.2648>) have also quantified this for hippocampal 2P imaging and write “*during periods of running the Z-motion was 0.7 ± 0.2 μm*”. These values hardly seem to qualify when Harris et al. write “*In cases with large Z movement, multiplane imaging and online motion correction can help to minimize artifacts.*”

Gauthier & Tank (2018) describe that for recordings collected with a resonant scanner (as we did), they used a type of piecewise rigid motion correction and patch-wise CNMF (see page e3 of STAR Methods). This has been later implemented in the new CalmAn version (Giovannuci et al. 2018), which is precisely what we have used for our analysis.

5. The anatomical quantification is helpful, but again it seems xy optical flow has been estimated here as opposed to z-motion.

Indeed, optical flow only estimates xy motion, which is also a potential source of artefacts that we have thus controlled for.

6. “While we concur with the reviewer that average calcium z-values around lick onset are relatively small (as seen in Figure 5a-c), we believe this is mostly due to the fact that the effects are largely carried by a subpopulation with consistent lick-associated calcium responses.” – Thank you for clarifying this; I did not appreciate before that the significantly

lick-excited neurons were such a small percentage (7% in dHPCNAc and 4% in dHPC-). This indicates that it is not appropriate to average over the whole population as shown in Fig. 5a-c. Furthermore, 5f-g is still confusing – if the average z-scored traces in 5f-g are the significantly excited or inhibited neurons as stated in the figure legend, why are the z-score values still so small, if values reached up to 6 sd on the single neuron level? Please see the provided figure pdf for an example of z-score magnitude in a GFP artefact (it is on the order of ~0.2-0.4 sd, though sometimes greater).

We have averaged over identified projecting- and non-projecting neurons to point out that there are population average differences between the two groups, which, we believe is an important piece of information for readers. In a second step, we identified individual neurons that are significantly lick-modulated, and found an elevated proportion of these in dHPC^{→NAc} neurons. For this lick-excited subpopulation, we observe average calcium increases of about 0.6 SD. Even if these values on some occasions reach 6 SD for an individual lick event, we note that this only holds for very strongly-responding neurons and is later averaged for each neuron across dozens of appetitive lick events, not each one of which elicits such a strong response. Importantly, using the Wilcoxon non-parametric test (as also suggested by the reviewer) for each lick event per neuron yields lick-modulated neurons that are reliably modulated by appetitive licking, regardless of the absolute values used. For further clarification, we would also like to point towards **Figure R4** where averaged lick-evoked signals of individual neurons are seen to reach around 1.5 SD. Given the reviewer's valid criticism of the relatively small magnitude of population averaged responses, we have now decided to move Figure 5a-c into the supplementary (Figure S7d-f). We would also like to point out that the fact that only a small subpopulation is significantly modulated by appetitive licking is further evidence that lick-related global z-motion is unlikely to account for the observed calcium responses.

6a. Importantly, do the significantly excited and inhibited neurons comprise larger fractions than one would expect by chance (i.e. as the possible result of an artefact?)

We thank the reviewer for pointing this out. We have previously not performed this because we have directly compared the two projection subpopulations and, separately, identified lick modulation beyond that expected by chance, via our GLM (Figure 7). We have now calculated the percentage of neurons that would be classified as lick-excited or lick-inhibited by chance alone. For this, we performed 100x random circular shifting of the lick trace by between 500 to 5,000 frames, and found that our data points fall well beyond the 99th percentile (see distribution of randomly shifted data below and in modified **Figure 5e,f**). The average proportion of lick-excited neurons was only 1.5% for both projection subpopulations (compared to 4% / 7% in our true data), and markedly higher for lick-inhibited neurons, with 12% / 10% (compared to 20% / 17% in our true data). Given that the absence of licking is more prevalent than the presence, it is not surprising that the chance levels for a classification as lick-inhibited neuron would be higher.

6b. I think an appropriate way to identify neurons that truly encode licking would be to first confirm what the chance detection level is using ~2 GFP control mice (i.e. what fraction of GFP ROIs are “significantly” lick-excited/inhibited) and determine what the magnitude of the lick-induced artefact is in the green channel in the authors’ dataset. Next, classify neurons in the GCaMP dataset that have significant lick-bout-triggered responses (using a

non-parametric test of post vs. pre) AND have responses that exceed the threshold defined by the lick-bout-triggered GFP magnitude. This may limit the significantly excited cells to those with large and reliable responses like the 6 sd example in Fig. 5e.

We have measured the level of lick-excited neurons classified by chance to be ~1.5% (see above). We have then indeed used a non-parametric test (Wilcoxon's Signed Rank) to determine significant lick-bout-triggered responses, and found proportions significantly higher than this (4% / 7%).

6c. I recommend then presenting these highly excited (or reliably suppressed) neurons using their maximum-normalized activity rather than a z-score – this will avoid the negative-going signal for the “lick-inhibited” neurons and make it clear that these neurons are just not firing at lick times, as shown in Fig. 5e cell 11, as opposed to decreasing their activity “below baseline” (line 253). Deconvolved activity cannot fall below baseline since baseline of deconvolved events should be zero.

On a single-neuron level, of course, there is no deconvolved activity below baseline. We have not claimed this. On a population-average level, however, there exists a baseline defined by the average population activity that can be lower than average during certain time periods such as the one described. In line 253, we are referring to the population average displayed in Figure 5b. To ensure readers understand this difference, we have added the following sentence to our results: “We termed these neurons “lick-inhibited” or “lick-excited”, respectively, while noting that “inhibited” neurons are simply showing less average calcium activity immediately after lick onset compared to before, considering that deconvolved calcium activity cannot become negative.”

7. The GLM does indeed complement your other analyses, but I would still be curious to know how much licking modulation it detects from a GFP control.

8. See response #4 about xy vs. z motion.

9. I appreciate the added video. A few of these cells, such as cell 10, do seem to have very strong lick-correlated transients; I would be curious whether these cells produce some of the largest amplitude signal, exceeding any possible artefact contribution.

We are glad the reviewer appreciates the video provided, and would like to again point out the clearly visible stability along xyz (potential z-motion would be visible as global intensity changes in red/green channels). In the latest version of our manuscript, we now also provide this video as a supplementary video for our readers. In an effort to provide full transparency and provide the reviewer with an intuition of the amplitudes involved, we have attached the average lick-related signals of all cells seen in the video (**Figure R4**), both from GCaMP and mCherry, both z-scored deconvolved signal (a), as well as raw z-scored $\Delta F/F_0$ (b), and the non-normalised $\Delta F/F_0$ (c). The latter serves to demonstrate how small any potential deflections in signal of the static mCherry channel are, directly compared to the signals from GCaMP, given how z-scoring can potentially inflate noise.

Figure R4: Averaged signal of 12 sample cells shown in the video provided, around the time of appetitive lick onset. a) Deconvolved signal, z-scored. b) Raw $\Delta F/F_0$ signal, z-scored. c) Raw $\Delta F/F_0$ signal (F_0 defined as 8th percentile). Red traces refer to calcium signals from dHPC→NAC neurons, green traces to calcium signals from dHPC– neurons, grey traces to respective static red channel (mCherry) signals. Note that (a-b) show the same Y scale throughout, while (c) shows Y scales adjusted for each cell.

To illustrate how this activity compares to the rest of lick-excited neurons, we also provide the deconvolved z-scored traces of the neurons in the video (coloured) compared to all lick-excited neurons (grey) in **Figure R5**:

Figure R5: Averaged deconvolved and z-scored signal of lick-excited neurons around the time of appetitive lick onset. Left panels show dHPC→NAC neurons, right panels show dHPC– neurons. Top panels show deconvolved calcium activity, bottom panels show deconvolved static mCherry signal. Coloured traces represent those neurons taken from the video provided.

10. Finally, re: appetitive vs. consummatory licking – I agree with the authors that it is important to separate these as they have done. It is also important to consider that they are associated with different movement dynamics; appetitive licking typically occurs while the animal is slowing down but still walking (see provided figure), while consummatory licking tends to occur once the animal is closer to fully stopped. This observed difference in movement dynamics seems to concur with the videos provided by the authors. When I have isolated consummatory licking in my own data, I have likewise observed a decrease in static GFP fluorescence in the ROI population average as the authors observe with GCaMP, and this change is distinct from what is seen during appetitive licking (see provided figure).

We would like to point out that, upon consummatory licking in our data, we see a decrease in calcium activity, while the signal of the static channel shows an *increase*, similar to the small increase observed with appetitive licking (see **Figure R1e,f**). This shows that both appetitive and consummatory licking result in small positive inflections in our static channel that are associated with either positive or negative calcium responses, depending on the type of action involved. If our calcium signals were the mere result of such movement artefacts, we would expect similar dynamics between the dynamic and static channel throughout our recordings. We would also like to note, again, that the dynamics observed in the static channel are minuscule in comparison to our calcium signals, as evidenced by the $\Delta F/F_0$ values (**Figure R1f**).

10a. As a separate example of how GFP artefacts can be elicited by certain locomotor dynamics, please see Fig. S2D of Yogesh & Keller 2023, eLife (<https://doi.org/10.7554/eLife.89986.1>)

The authors find <1% $\Delta F/F$ changes in GFP linked to locomotion, not licking. They also find a similar magnitude of GFP fluorescence change in response to visual gratings that are not directly related to any movement, suggesting that this magnitude of responses may not (only) be attributable to movement artefacts.

10b. In conclusion, I strongly recommend that the authors collect a small amount of data from ~2 mice expressing static GFP to verify their findings. These mice would not have to perform the entire task – examining appetitive vs. consummatory licking in the first several days of training would probably be sufficient, as long as the behavior is similar to that of the GCaMP animals. If there is a substantial lick-induced signal in the GFP controls, lick coding in the GCaMP dataset should only be considered significant for neurons that exceed a threshold defined by the GFP signal. If this reduces the fraction of lick-excited neurons such that there is no longer a statistically significant difference between the dHPC-to-NAC and dHPC- populations, the authors should report this. If either population includes neurons with robust appetitive lick coding beyond the artefact, this would itself be an interesting and novel finding, going beyond what has been reported in the literature previously.

Given our new highly extensive and in-depth analysis of our existing dataset, we believe additional experiments would provide limited gain. Even for a set of experiments of N=2, animals would need to be injected, craniotomised, habituated, trained, recorded from, and analysed, requiring several more months of work. Due to infrastructure constraints, the result would be data collected from a different setup (our microscope had to be recently disassembled and re-assembled in a new institute). As the reviewer themselves notes, these differences may lead to control data that may not be directly comparable to the data we previously collected. Hence, the best control is simultaneous imaging of a static channel (see also Engelhard et al., 2019, Nature; Rose et al., 2016, Science; Driscoll et al., 2017, Cell), which we have provided (and now expanded on) and which, through multiple angles of analysis, suggests our lick modulation is not the result of mere motion artefacts.

Minor:

1. *Re: original point #2 regarding individual animals: I thank the authors for illustrating data points for each animal. In the newly added Supplemental Fig. S5, however, I only see 5 points plotted in (a), although I thought 6 mice were imaged. Mouse 209 seems to be missing? In addition, there are 5 points for dHPC- and 3 for dHPC-to-NAc in (b-f). Were some mice excluded for having zero place cells in the dHPC-to-NAc population as suggested in (a)? Were these same mice excluded from all subsequent analyses? If so, the n of mice that were actually included in each analysis should be reported in each figure legend.*

We thank the reviewer for pointing out this potential point of confusion. Indeed, Figure S5a only shows 5 data points because we detected no significant place cells in mouse 209. S5b-f shows only 3 points for the dHPC[→]NAc population because we could not identify dHPC[→]NAc place cells in these. We have now included these N in the figure legend of Figure 2: “Data in g-k: n = 5/3 mice with place cells”. These animals have not been excluded from further analysis, because the absence of place cells was no exclusion criterion for our analysis of velocity and lick coding; all other imaging data thus stem from n = 6 mice.

1a. *Relatedly, the linear mixed effects model reported in the rebuttal is helpful – those results should ideally also be reported in the supplemental. It suggests that spatial information and sparsity are the most robustly enhanced properties of the dHPC-to-NAc cells.*

We have now included the results of the LMMs in new **Supplementary Table 4** reporting on effect sizes, beta coefficients, standard errors, z-values, p-values, and confidence intervals for both fixed effects and random effects for all five spatial tuning characteristics.

2. *Clarifications about new inhibition results:*

2a. *In the response to reviewer 1, the authors note: “Interestingly, we found no effect of our optogenetic inhibition during reward consumption”. Where is this shown? Supplemental Fig. 2a only shows number of rewarded laps, not the effect of inhibition on lick rate in the reward zone, as the statement implies. This seems an important result to include for the readers.*

We thank the reviewer for pointing this out. We have now included average lick traces and a quantification of consummatory licking in new **Figure S2e,f**. The data show there is no significant effect of optogenetic inhibition on consummatory licking. Interestingly, and in line with our previous discussion, there is rather a trend toward an increase in consummatory licking.

2b. *What distance is defined as “shortly before the reward zone” for the quantification in Fig. 1h? Similarly in Supplemental Fig. S2G, how is “near reward” defined? Is this the same distance as used for “place fields near reward zone”?*

“Near reward zone start” in Figures 1h-j and S2g is defined as 20 cm immediately preceding the start of the reward zone, as also indicated by the black line above the traces in Figure 1h. “Place fields near reward zone” is defined as place cells with a place field in the 30 cm reward zone or the 30 cm preceding anticipation zone, for a total of 60 cm, as indicated in Figure 3e. We have now clarified this in the figure legend text and the methods.

2c. The authors interpret the inhibition experiment as “suggesting a lack of spatial precision to guide appetitive licking” (line 106). However, the reported effect is a reduction in peak licking rate, correct? This could alternatively be interpreted as lower confidence in the spatial location or lower willingness to expend effort, whereas an extension of the “ramp” of anticipatory licking to earlier positions on the belt would more clearly imply lower spatial precision. Perhaps the authors could clarify what they mean by “lack of spatial precision”, and/or try to quantify the shape of the ramp – one possibility would be to use a KS test on the distribution of pre-reward licking positions in controls vs. ArchT animals.

We thank the reviewer for pointing out this ambiguity in our phrasing. Indeed, using a 2-sample KS test on the average distribution of appetitive licking per position between ArchT and EYFP results in a significant difference between the two groups ($ks = 0.271$, $p = 0.0257$) (new **Fig. S2b**). At the same time, we agree with the reviewer that our data of reduced appetitive licking near reward zone start in Figure 1 h-i could also be interpreted as a decrease in confidence. Given, however, that the overall licking and appetitive licking of animals was not different between the two groups on the day of stimulation ($t(\text{overall}) = 0.552$, $p = 0.596$; $t(\text{appetitive}) = 0.545$, $p = 0.601$) (new **Fig. S2c,d**), we do not believe there is a decreased willingness to expend effort. Based on the reviewer’s suggestion, we have added this additional data into our supplementary figure and have adjusted our discussion accordingly, see lines 349-352: “This decrease indicates either a lack of spatial precision, as suggested by the flattened shape of licking across spatial positions, or a decrease in confidence to lick near the learned reward location, as evidenced by the lick decrease observed. Given that overall (appetitive) lick rates were not affected, the animals’ willingness to expend effort seems not to have been impacted by this inhibition.”

3. Please consider increasing the font sizes in Supplemental Figs. S5 and S8.

We have now increased the font sizes in these supplementary figures, as suggested. We have also split our previous Figure S8 into S8 and S9 to help with readability.

REVIEWERS' COMMENTS

Reviewer #3 (Remarks to the Author):

I sincerely thank the authors for continuing to thoroughly address my concerns about the lick coding. The reviewer figures and additional supplemental material they've provided have finally made it clear how much larger the appetitive-lick associated calcium transients are in a subpopulation of cells compared to any possible movement artefact in their data (a comparison that I agree can be difficult to evaluate from z-scored data). I think it is a service to the field that they have added text to directly address this issue in the manuscript. I have no further concerns and congratulate them on this exciting and impactful work.

REVIEWERS' COMMENTS

Reviewer #3 (Remarks to the Author):

I sincerely thank the authors for continuing to thoroughly address my concerns about the lick coding. The reviewer figures and additional supplemental material they've provided have finally made it clear how much larger the appetitive-lick associated calcium transients are in a subpopulation of cells compared to any possible movement artefact in their data (a comparison that I agree can be difficult to evaluate from z-scored data). I think it is a service to the field that they have added text to directly address this issue in the manuscript. I have no further concerns and congratulate them on this exciting and impactful work.

We thank the reviewer for their constructive criticism and their help in improving the clarity of our manuscript.